



# The ELK global emission inventory for the transport sectors

Mattia Righi[1], Simone Ehrenberger[2], Sabine Brinkop[1], Johannes Hendricks[1], Jens Hellekes[3],
Paweł Banyś[4], Isheeka Dasgupta[2], Patrick Draheim[5], Annika Fitz[6], Manuel Löber[7], Thomas Pregger[5],
Yvonne Scholz[4], Angelika Schulz[8], Birgit Suhr[9], Nina Thomsen[8], Christian Martin Weder[10],
Peter Berster[10], Maximilian Clococeanu[10], Marc Gelhausen[10], Alexander Lau[10], Florian Linke[10],
Sigrun Matthes[1], and Zarah Lea Zengerling[10]

[1]Deutsches Zentrum für Luft- und Raumfahrt (DLR), Institut für Physik der Atmosphäre, Oberpfaffenhofen, Germany
[2]Deutsches Zentrum für Luft- und Raumfahrt (DLR), Institut für Fahrzeugkonzepte, Stuttgart, Germany
[3]Deutsches Zentrum für Luft- und Raumfahrt (DLR), Institut für Methodik der Fernerkundung, Oberpfaffenhofen, Germany
[4]Deutsches Zentrum für Luft- und Raumfahrt (DLR), Institut für Kommunikation und Navigation, Neustrelitz, Germany
[5]Deutsches Zentrum für Luft- und Raumfahrt (DLR), Institut für Vernetzte Energiesysteme, Stuttgart, Germany
[6]Deutsches Zentrum für Luft- und Raumfahrt (DLR), Institut für Maritime Energiesysteme, Geesthacht, Germany
[7]Deutsches Zentrum für Luft- und Raumfahrt (DLR), Institut für Verbrennungstechnik, Stuttgart, Germany
[8]Deutsches Zentrum für Luft- und Raumfahrt (DLR), Institut für Verkehrsforschung, Berlin, Germany
[9]Deutsches Zentrum für Luft- und Raumfahrt (DLR), Institut für Raumfahrtsysteme, Bremen, Germany
[10]Deutsches Zentrum für Luft- und Raumfahrt (DLR), Institut für Luftverkehr, Hamburg & Köln, Germany

**Correspondence:** Mattia Righi (mattia.righi@dlr.de)

**Abstract.** The transport sectors, comprising land transport, shipping and aviation, are major contributors to climate change and have a detrimental impact on air quality, with adverse consequences for human health. The emissions from transport, already contributing 23% of total anthropogenic $CO_2$ emissions in 2019, are projected to continuously grow in the future, challenging the achievement of climate protection and pollution reduction targets. A major goal of transport research on climate and air quality is the accurate assessment of its impacts, which requires detailed emission data to drive atmospheric models and calculate projections for future scenarios. This paper presents the ELK global emission inventory for the transport sectors. The inventory is developed using a consistent bottom-up approach fed with a wide range of input data to model the transport fleets of land transport, shipping and aviation. It provides several major improvements over existing datasets, such as the explicit resolution of the emissions at the subsector level, the consideration of transport-specific quantities and emission species, and the quantification of the transport-related emissions from the energy sectors. The emission data is complemented by a quantitative uncertainty score, based on a detailed expert-judgement analysis along the modelling chain, from the activity data to the emission factors. The emission data is validated by comparing it with other, well-established global inventories, and biases are discussed and, where possible, explained in terms of the different assumptions and features of the underlying emission models. The ELK dataset is released under an open-source licence to encourage their use in the atmospheric modelling community.

## 1 Introduction

The emissions from the transport sectors contribute significantly to climate change. According to the sixth assessment report of the Intergovernmental Panel on Climate Change (IPCC; Jaramillo et al., 2022), transport was responsible for 8.7 Pg of





$CO_2$-equivalent emissions in 2019 and shared 23% of global energy-related $CO_2$ emissions. Land-based transport emissions represent the largest transport source (77%), while shipping and aviation account for 11% and 12%, respectively. A major

concern of the transport emissions are their large growth rates: according to Lamb et al. (2021), the greenhouse gas (GHG) emissions of the transport sectors grew at a global rate of about 2% per year in the last three decades, with considerable differences between developed and developing countries: the $CO_2$-equivalent emissions of transport in East Asia, for instance, grew by a factor of 6 between 1990 and 2018, but only by 20-30% in Europe and North America during the same period. Emissions scenarios project a continuous increase in the future, in particular for the shipping and aviation sectors which are

hard to defossilise (Feng et al., 2020; Lund et al., 2020). In addition to $CO_2$ emissions, the combustion process of fossil fuels, still driving the vast majority of the fleet, leads to the formation of several short-lived climate forcers (SLCFs; Szopa et al., 2021), including $NO_x$ (=$NO+NO_2$), CO, non-methane volatile organic compounds (NMVOC), $SO_2$, and aerosol particles (such as black and organic carbon). These compounds can have significant climate effects (Righi et al., 2023; Mertens et al., 2024) and, at the same time, be harmful for air quality (Fiore et al., 2012). The introduction of policy measures and the evaluation of

their effectiveness is therefore challenging, because the impacts on both climate and air quality, as well as their trade-offs, need to be considered. Furthermore, some measures are applied on national or regional scale, especially for land-based transport, while sectors like shipping and aviation are regulated at the international level.

To address these scientific and policy-making challenges in the transport research, geographically resolved inventories for the emissions of all relevant compounds are essential. Given the global nature of the emissions and of the resulting climate

impacts, a global data coverage is a key requirement, although regional inventories at higher resolution are also necessary for air quality studies. These datasets are the starting point for quantifying the effect of transport on climate and air quality, for developing scenarios of future emissions, and for assessing policy- and technology-based mitigation strategies to protect climate and improve air quality. State-of-the-art global inventories of anthropogenic emissions commonly used in climate science (e. g. Hoesly et al., 2018; Crippa et al., 2024; Soulie et al., 2024) include data for the transport sectors, often aggregated

at the sector level, i.e. land-based transport (road and rail), shipping (international and domestic) and aviation. Being general-purpose inventories of anthropogenic emissions, however, these datasets do not usually provide information on transport-specific quantities, which are relevant to address the challenges of transport research outlined above. Such quantities include, for instance, water vapour emissions from aircraft (required for aviation contrails modelling; Burkhardt and Kärcher, 2011; Bickel et al., 2025) or highly resolved sectoral emissions (e. g., different vehicle or aircraft types). Most of the available

inventories also do not report the transport-related share of emissions from other sectors, like the emissions from oil refineries and their share driving the transport fleet, as these are usually integrated in the emissions from the energy and/or the industry sector. This information is of key importance for a comprehensive assessment of the transport impacts and will be even more so in the future, with the expected shift of the road vehicle fleet towards alternative energy sources, with no or reduced direct (tailpipe) emissions from vehicles (Ghosh, 2020).

The goal of the project ELK (*EmissionsLandKarte*, en.: Emissions Map) of the German Aerospace Center (DLR) is to develop a consistent, complete, comparable and transparent global emission inventory of greenhouse gases and short-lived climate forcers (SLCFs) for the transport sectors and their relevant subsectors, while also accounting for the indirect emissions





of transport in the energy sector. The ELK inventory described in this paper aims at providing the following improvements and added values over the existing datasets:

1. A consistent quantification of the emissions from all transport sectors with a bottom-up approach, using state-of-the-art emission models and a wide range of input data from different sources to drive these models.

2. The explicit resolution of the emission data at the subsector level, considering the emissions in 7 subsectors for the land transport emissions (cars, heavy-freight trucks, light commercial vehicles, buses, 2-wheelers, passenger rail, and freight rail) and in 4 subsectors for aviation (wide-body, single-aisle, regional, cargo). Shipping emissions are resolved between

international (ocean-going) shipping and domestic navigation.

3. The inclusion of transport-specific quantities and emissions species, such as flight distance, propulsion efficiency and water vapour emissions from the aviation sector (necessary for contrail modelling) and non-exhaust emissions from the land transport sector.

4. An estimate of the indirect emissions of transport resulting from the energy sector via oil refineries.

5. An advanced assessment of the uncertainties along the whole modelling chain, providing a quantitative uncertainty score at the country level (for land transport) and in the IPCC regions (for the other sectors), thus informing data users about the quality of the data.

The ELK inventory considers present-day conditions, providing emission data for the year 2019 (hereafter, the reference year). The emission model chain is structured in a way that it can also be used to project future emissions according to given

scenarios or including new fuel types and technologies that might be used for an energy transition in transport. For example, considering new fuel types, this means to allow to modify the emission species and their emission factor, and also to include additional species such as hydrogen ($H_2$).

The ELK inventory is generated using common data standards for gridded emissions, to facilitate their use in climate and air quality models. The datasets are made available as CF-compliant NetCDF files, including standard metadata and units

defined according to SI-standard. Furthermore, a common grid with a resolution of $0.1° \times 0.1°$ is considered for all sectors, so that spatial aggregations across the sectors can be calculated consistently without the need of regridding. The temporal resolution of the data is monthly, although hourly data is provided for some specific cases. Aviation data is provided on a three-dimensional grid, further including 48 vertical layers from the surface to an altitude of 47000 ft (∼14.3 km) above the mean sea level, with a vertical resolution of 1000 ft, representing the main flight levels of the commercial fleet. To reduce

the significantly larger amount of data required by a three-dimensional grid, aviation data is provided at a reduced horizontal resolution ($0.25° \times 0.25°$) and temporal resolution (annual and two seasonal averages over the November-March and April-October periods). If necessary, the ELK emission models are capable of modelling individual sectors at a higher resolution. To increase the internal consistency of the ELK dataset, the same underlying framework data (e. g., population, gross domestic product, trade flows, globally aggregated energy, and fuel consumption) and the same methodology is applied across the sectors

where possible.





This manuscript serves as the main reference for the ELK inventory. The methods for generating the emission data of each sector are described in Sect. 2. Section 3 describes the method applied for the quantitative assessment of the uncertainties in the emission data. The results are presented in Sect. 4 for selected species and compared with the transport emissions of other well-established global inventories (additional species are shown in the Supplement). The main conclusions of this work are summarised in Sect. 5.

## 2 Methods

### 2.1 Land transport

#### 2.1.1 Method overview

Land transport includes most of the everyday movement of people and goods and hence contributes significantly to transport-related emissions, with the largest share originating from road-based transport. However, due to its nature, which is characterised by individual movements on large networks with a large, heterogenous vehicle fleet, the creation of an emission inventory proves to be a challenging task. On a global level, the transport sector has been modelled by integrated assessment models (IAMs) and transport specific models. These models differ in their modelling framework, in the underlying country-specific data and considered emission factors (Yeh et al., 2017). A benchmark emission stock is Emissions Database for Global Atmospheric Research (EDGAR; Crippa et al., 2023) which have been developed over the past 20 years and which include the transport sector among others. The EDGAR group acknowledges the huge challenge of collating and harmonizing datasets of countries for road transport (Lekaki et al., 2024) and provides the data sources for emission factor databases, vehicle stock and assumptions for country specific unavailable data.

Given the focus on transport of the ELK inventory, the objective was to build on and improve previous methodologies to provide vehicle category specific datasets for both exhaust and non-exhaust emissions, together with uncertainty metrics and also to improve the methodology for spatial disaggregation of emissions. The vehicle categories considered in the ELK inventory are passenger cars, 2-wheelers, buses, light commercial vehicles (LCVs), heavy freight trucks (HFTs), passenger and freight rail. The inventory contains emissions of black carbon (BC), organic carbon (OC), $CH_4$, CO, $CO_2$ (both fossil-fuel-based emissions and total emissions including biofuels), hydrocarbons (HC), NMVOC, $NH_3$, $N_2O$, $NO_2$, $NO_x$, particulate matter ($PM_{10}$ and $PM_{2.5}$), particulate number (PN), and $SO_2$. Emissions from non-exhaust species include $PM_{10}$ and $PM_{2.5}$ from tyre wear and brake wear, and PN from brake wear.

Two methods are developed in the ELK project using the following approaches: i) a global inventory is generated based on a proxy data set and traffic counts for spatial disaggregation; and ii) a European inventory is generated based on a transport model. Fig. 1 shows a schematic overview of the methods, which differ primarily in the spatial distribution of the transport activity. The global approach calculates the total emissions per country and species based on activity and emission factors and then spatially disaggregates them on a grid using proxy data. For selected countries with good quality data for validation, a model to spatially disaggregate emissions based on traffic counts is applied. While this approach is suitable for the estimation

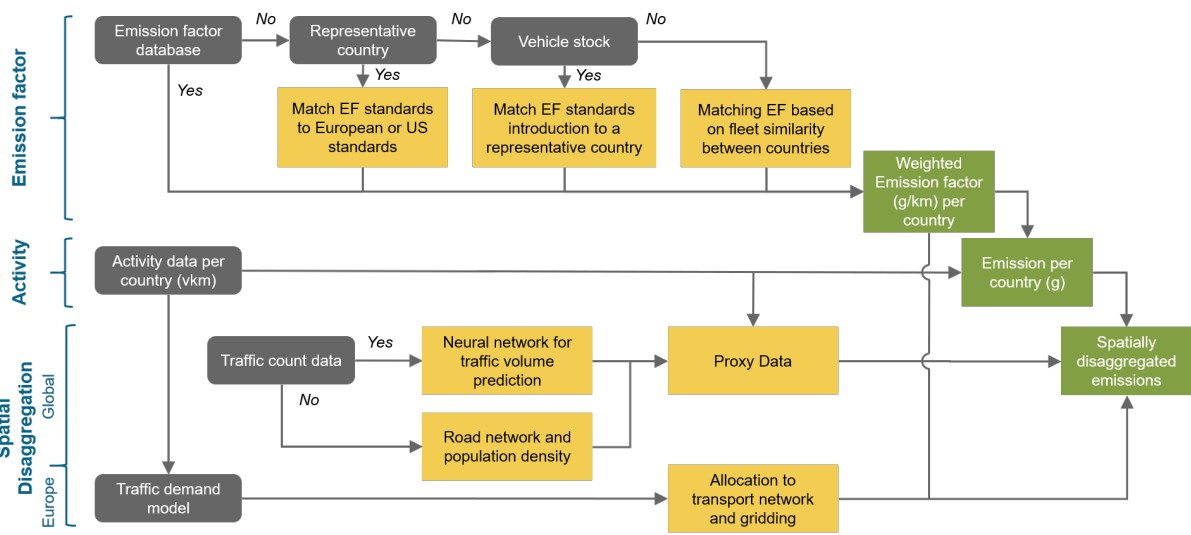

**Figure 1.** Schematic of the emission computation for a given vehicle category.

of emission sources on a larger scale with scarce calibration data sources, forecasting and the calculation of scenarios requires dedicated transport models describing travel behaviour (as demonstrated by Matthias et al., 2020). This is why, as a proof of concept for the correct allocation of emission sources, an additional emission inventory is created for Europe, where transport activity is first distributed on the road network, before aggregating the resulting emissions on a grid. Still, both approaches share the following major components: i) transport activity data by vehicle category on the country level, including passenger and freight transport; ii) emission factors on a country level; and iii) a spatial disaggregation model. These three components are described in the following.

### 2.1.2 Transport activity data

Emissions from land transport are defined by transport activity, or travel demand, which constitutes transport volume by people (person kilometres travelled) and goods (tonne kilometres travelled). Therefore, a global activity database for the reference year 2019 is created based on data for population and economic development.

For passenger transport, the calculation of transport volumes follows the methodology described by Thomsen and Schulz (2024). The first step is the determination of national vehicle fleets. Based on historical data, correlating motorisation to the gross domestic product per capita (GDPpc) growth for representative countries, the parameters of Gompertz functions for different world regions are estimated. The Gompertz functions are used due to their typical s-shaped curve, which allows to parametrise the point where a plateau for motorisation is reached. Applying these functions to the GDPpc in the reference year yields a motorisation rate (vehicles per capita), which is multiplied with the total population to generate the total vehicle fleet per country. Using reference values from literature for mean annual mileage per vehicle and world region, the total vehicle





kilometres travelled per country are generated. Applying occupancy rates then yields the transport volume for car transport. The transport volumes for other modes are then calculated based on these car transport volumes by applying representative modal splits.

Concerning freight transport, data for various countries of the world is available from sources like Eurostat or OECD. Since this data is regularly reported only for OECD countries, other countries are assigned to reference countries based on their similarities. Freight transport performance on the road and, where available, on the railways is then modelled for the reference year using regression models. Data up to 2013 is used as training data for the regression and the model is applied using population and GDP data for 2019. The comparison of the 2019 observations revealed some significant deviations between the modelled and observed data in both directions. In cases where observations are available, these are used as activity data, otherwise correction factors describing the relationship between modelled and observed data is derived. These factors are then used to adjust the transport performance in countries without observed values, using the correction factors of the representative country.

### 2.1.3  Emission factors

The methodology used to calculate emission factors of vehicles per country, vehicle category and species is shown in Fig. 1. Weighted emission factors represent the emission factor for a specific vehicle category and species, taking into account the distribution of drivetrain types, segment and vehicle ages within the fleet for the reference year. If country specific emission factor databases with integrated vehicle stock information was available (like HBEFA Infras, for Germany) then they are used directly for calculating the weighted emission factors. If not, vehicle stock data of a country combined with emission factor (EF) standards is used to calculate the weighted emission factors. If no vehicle stock data is available then assumptions from similar countries are used to for assigning emission factor values.

The main data source for the vehicle stock data for the year 2021 is acquired from S&P Global Mobility (2018), which covers 76 country fleets worldwide. A key feature of the dataset is that the number of registered vehicles in the stock data is provided differentiated by vehicle category, segment or weight class, drivetrain or fuel type, and vehicle age or year of initial registration. Since emission factor datasets have a similar or identical structures, the core idea was to link the vehicle stock datasets with emission factor datasets via country specific emission factor standards so that weighted, fleet-average emission factors can be calculated for these countries.

Emission factor databases with integrated vehicle stock of road vehicles are less widely available than vehicle stock data. Thus, the emission factors of a vehicle are determined by its emission standard, according to which it is type-tested and registered. Globally, there are essentially two major sets of rules for regulating pollutant emissions from road vehicles, differing by the year of their implementation in a given country/area. These are the European or UN-ECE based regulation with corresponding Euro 1-6 levels and the U.S. regulation according to EPA or CARB which applies to North, Central and partly South America. Thus, two basic emission factor datasets are prepared for the global emission factors database: one for Euro-based vehicles from HBEFA (Infras) and one for U.S.-based vehicles, from the California EMFAC model (CARB, 2021).





For the 6 European countries covered by HBEFA and for USA, the respective country-specific emission factor database or model-internal data included are considered for the calculation of the weighted emission factors. For all other countries for which S&P Global Mobility stock data is available, the emission factors are calculated on the basis of the global Euro- or U.S.-based data. Different emission classes or Euro levels are introduced and implemented at different years in respective countries. The goal was to assign a matching emission factor for each vehicle listed in the stock data by researching the applicable emission standard for each vehicle registration year in the period between 1971 and 2021. This mapping is implemented for the vehicle markets of representative countries for which information on emission standards are available (Argentina, Brazil, Canada, China, (rest of) Europe, India, Japan, Mexico, Russia, South Africa, South Korea, and U.S.). Assumptions of emission factor standards are made for the remaining countries which are available in the S&P Global Mobility dataset by assigning to one of the representative countries, for which similar emission standards apply demonstrably or by assumption (i.e., the same emission standard is assumed for the same registration year). The remaining countries of the world, for which no stock data is available, are assigned weighted emission factors from one of the 76 bottom-up calculated countries. Similarities in emissions regulations and geographical correlations are considered. As a result, calculated $NO_x$ emission factors are shown in Fig. 2a with vehicle-age-related contributions of passenger cars to their resulting NOx emission factor using the vehicle stock, assumed age-related mileage driven curves, emission standards and introduction year of the particular country. This is relevant for policies like vehicle age-related circulation bans and understanding stock turnover effects.

A similar approach is adopted to obtain emission factors of commercial vehicles, although these vehicles have an additional parameter related to their weight class, whose definition varies across datasets available. To finalize these definitions, several options are considered for the vehicle allocation in each case. Representative vehicles from HBEFA and EMFAC are selected which correspond as closely as possible to the S&P Global Mobility vehicle categories. For this purpose the weight classes with the given weight ranges are considered as closely as possible and global information on $CO_2$ tailpipe emissions of commercial vehicles is used for comparison. Fig. 2b shows the share of $NO_x$ emissions calculated by the above described methodology from passenger cars and commercial vehicles in representative countries. The relative contributions are influenced by the diesel share of passenger cars in the stock and their age which determines the emission standard they conform to and also vehicle activity. $NO_x$ emissions are heavily dominated by heavy commercial vehicles which further support the need of vehicle category specific inventories and policies.

It should be noted that in addition to a vehicle emission standards, there are other country-specific, in-vehicle and out-of-vehicle characteristics that can determine or influence its real-world emission level. These include vehicle size and engine size, traffic conditions, ambient temperatures, and fuel quality among others. In principle, this approach produces global emissions inventories for country fleets with annual resolution. Due to this rather low level of detail, it is assumed that, apart from individual exceptions, the influencing factors mentioned are negligible. The effects of these parameters and other stock data quality influencing errors are considered in the uncertainty analysis described in Sect. 3.1.2. As a note, the stock data year (S&P Global Mobility, 2018) is slightly inconsistent with the reference year for the ELK inventory, but the differences between 2019 and 2021 are expected to be minimal.

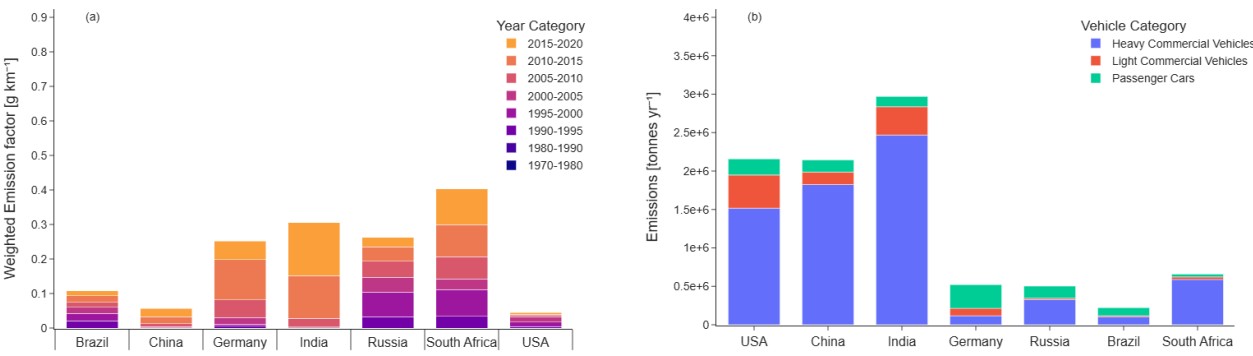

**Figure 2.** (a) Contribution to the $NO_x$ emission factor by different passenger car ages; (b) $NO_x$ emissions from vehicle categories in representative countries in 2019.

### 2.1.4 Spatial disaggregation

With emissions resulting from transport activity, which itself results from travel demand, applying a transport model for the spatial disaggregation is the logical approach. However, these models require a large amount of data for application as well as calibration and validation. Especially behavioural data, like National Household Travel Surveys (NHTS), is not widely available. Therefore, an approach using proxy data is used globally, while a simplified transport distribution model is applied for Europe. These methods are described in the following.

For the spatial disaggregation at the global level, existing transport inventories usually consider road type and density as well as population to disaggregate country-level emissions to its respective grid cells (Janssens-Maenhout et al., 2019). This has been found to lead to an underestimation in connecting roads and suburban areas and thus an overestimation in urban areas with high population densities (McDonald et al., 2014; Gately and Hutyra, 2017). An improved methodology is implemented in the ELK inventory to develop spatial proxies for passenger cars to disaggregate emissions in a country. A graph neural network is trained and tested to predict traffic counts in the U.S., Germany and UK, where data is openly available. The features used are population at different spatial scales (aggregated population at different distance buffers around a point), road density, population density, proximity to urban centres, and engineered in-betweenness features. In-betweenness features incorporate information of the surrounding cities and their contribution to traffic flow at a particular point. A marked improvement can be observed in traffic flow prediction in remote areas with high traffic flow by incorporating these features. This methodology can be easily implemented for other developing countries if traffic count data becomes available. For the other modes like trucks (road type 1 and 2; Meijer et al., 2018) and 2-wheelers (road type 3, 4 and 5 multiplied by population) and buses (road type 2, 3 and 4 multiplied by population), proxies of road density of certain types combined with population are used globally and need further research and improvement on their applicability and country-wise relevance.



The spatial disaggregation for Europe is based on the universal transport distribution model ULTImodel (Thomsen, 2023) and described in Thomsen and Seum (2021) and Thomsen and Schulz (2024). Here, the transport volumes of the considered

countries are distributed between cells using a gravity model and allocated to a higher-level road network with a network assignment. As reference cells, the NUTS 3 regions (Nomenclature of Territorial Units for Statistics; Eurostat, 2022) are used and the road network is generated from Open Street Map (https://www.openstreetmap.org/, last access: 10 June 2025). The distribution is based on the travel times and distances between cells, and their attractiveness as origins or destinations of trips is derived from their population and industrial sites. Transport volumes in the subordinate road network can be disaggregated

at cell level. The results of the model run are then intersected with the inventory grid and aggregated per pixel by route type, whereupon the emissions are calculated using weighted emission factors.

The resulting spatial distributions with the two methods are shown in Fig. S1 for $CO_2$, while the total emissions over the domain of the European inventory are compared in Table S1. Note that in the following only the global data will be discussed and validated, as this is the main focus of this work.

## 2.2  Shipping

For the shipping sector, the ELK inventory distinguishes between all ship movements in maritime environment (Sect. 2.2.1) and inland navigation (Sect. 2.2.2), for which different input data structures and modelling approaches are applied. For consistency with the IPCC definitions (IPCC, 2006), we use the terms international shipping (IPCC sector 1.A.3.d.i) and domestic navigation (IPCC sector 1.A.3.d.ii) to distinguish these two subsectors. Note, however, that domestic navigation in the ELK inventory

only includes inland waterways, while short-range coastal shipping is part of the international shipping sector, although this is not fully consistent with the sector definitions.

### 2.2.1  International shipping

Maritime emissions and their environmental effects have been an important topic of research worldwide. The first study aimed at calculating emissions originating from vessels which took into consideration operational activities of the fleet applied a top-

down approach (Corbett and Fischbeck, 1997). In that study, the authors based their analysis of air pollution on the research done by the International Maritime Organization (IMO). They concluded that the maritime sector was a significant source of air pollution on a global scale. Another interesting study of maritime emissions was completed by Eyring et al. (2005), who analysed five decades of civilian and military fleet movements and used them for global modelling of tropospheric chemistry. Various research activities were focused on predictions of future maritime emissions, too. An example analysis of the global

maritime emissions and marine fuel consumption including future scenarios for 2050 was carried out by Paxian et al. (2010), considering the opening of Arctic polar routes as the aftermath of projected sea ice decline.

After the introduction of the Automatic Identification System (AIS; IMO, 2004), it became easier to track vessel positions and to store their movements in big-data archives. One of the first attempts to utilise AIS data for computation of maritime emissions was undertaken by Jalkanen et al. (2009) using the Ship Traffic Emission Assessment Model (STEAM) and focusing

on the Baltic Sea. Since then, the method has been developed further (Johansson et al., 2017) and used by internationally

recognised research projects like the Copernicus Atmosphere Monitoring Service (CAMS; Soulie et al., 2024). The STEAM model takes into account the environmental factors which interact with vessel movements, such as currents, waves, winds and ice conditions. Such approach has a high demand for computational power and is sensitive to various uncertainties related to the estimation of dynamic environmental influences depending on time and location. Within the scope of the STEAM model,

it was assumed that considering the aforementioned factors could lead to an increase in the global annual fuel consumption estimates by as much as 5-15% (Johansson et al., 2017).

In 2020 the IMO published the fourth edition of their comprehensive assessment of maritime emissions between 2012 and 2018 (Faber et al., 2020). The work also presents a revised algorithm for calculating maritime emissions, built upon the STEAM model. The IMO researchers reduced the complexity of the original model and generalised dynamic environmental

factors in reference to aggregated classes of vessel type and size. Thanks to its lower requirement for computational power, the IMO algorithm is chosen for calculating maritime emissions for the ELK inventory, considering the global vessel traffic for the reference year 2019. A schematic overview of the applied method is depicted in Fig. 3.

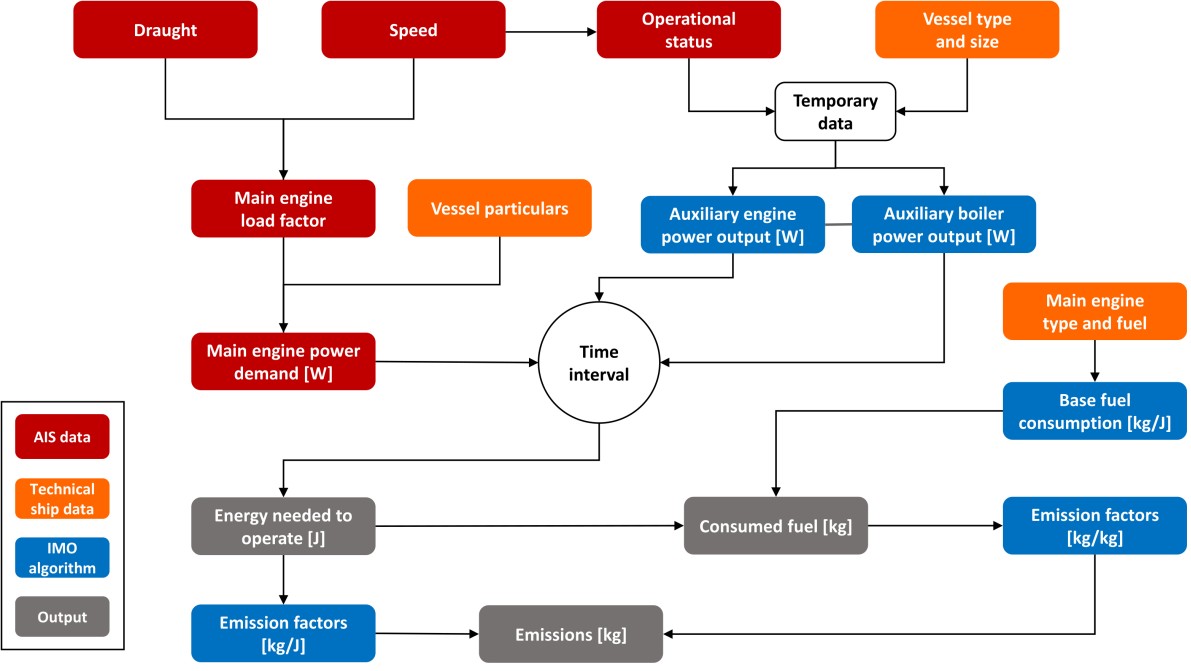

**Figure 3.** Schematic overview of the method applied for calculating the emissions from international shipping.

The relevant input data includes two datasets, describing vessel traffic and vessel particulars, respectively. The vessel traffic data originated from the AIS. The AIS data was purchased from FleetMon (https://www.fleetmon.net/, last access: 10 June

2025). It contains over 65 billion vessel movements registered around the world in 2019 by the shore-based and satellite-based AIS reception systems. In addition, AIS data includes basic technical details, such as for instance the hull size, the vessel type and the current draught. The following AIS messages are utilised for the emission computation (ITU, 2014, p. 105-106): class



A position report (IDs 1, 2 and 3), class B position report (ID 18), long-range position report (ID 27), and class A static and voyage related data (ID 5).

The vessel particulars describe technical details of 1,810,037 vessels. The dataset was purchased from Clarksons Research (https://www.clarksons.com/research/, last access: 10 June 2025) and includes both vessels in operation and historical craft. This data mixture is helpful for the gap-filling of technical data because it allows to search for fleet similarities among sister or near-sister vessels to complete missing technical details, especially in regard to engine and fuel parameters. The number of vessels which were engaged in voyages during 2019 and could be identified both in AIS data and the vessel particulars database amounts to 86,192. It is assumed that 50.4% of the fleet uses heavy fuel oil (HFO) and about 48.6% uses marine diesel oil (MDO). The remaining 1% of the vessels use either liquefied natural gas (LNG) or methanol. The vessel particulars dataset includes the following parameters: vessel identification, be it the Maritime Mobile Service Identity (MMSI), IMO number or call sign; construction year; vessel type and sub-type; reference power output of the main engine, main engine model, revolutions per minute (RPM) rating and engine design; fuel type; cargo capacity; maximum draught and maximum speed. With vessel identification data, especially MMSI, it is possible to merge the AIS data describing vessel movements with their technical parameters. The process chain for calculating the shipping emissions in the ELK inventory can be generalised as follows:

1. computation of the current power demand of the main engine;

2. calculation of the corresponding power demand of auxiliary engine and auxiliary boiler based on the operational phase of a voyage;

3. computation of energy necessary to move a vessel between two positions during a period;

4. query of the baseline fuel consumption parameters based on the main engine type, the fuel type and the year of construction;

5. calculation of the specific fuel consumption based on the current engine load;

6. assignment of the emission factors based on the engine and fuel parameters of a vessel;

7. computation of the emitted masses of the following species: $CO_2$, sulphur oxides ($SO_x$), $NO_x$, particulate matter ($PM_{10}$ and $PM_{2.5}$), CO, NMVOC, and BC.

It should be emphasised that there is currently no technical possibility to obtain the engine performance and fuel consumption data directly from AIS. Therefore, these parameters have to be averaged based on aggregated vessel types and sizes. The current load of the main engine $L(t)$ on board a vessel is calculated using the following equation (Faber et al., 2020):

$$L(t) = \left(\frac{D(t)}{D_{\text{ref}}}\right)^{2/3} \left(\frac{V(t)}{V_{\text{ref}}}\right)^3, \tag{1}$$





where $D(t)$ is the current draught of the vessel, $V(t)$ is the current speed of the vessel, $D_{\text{ref}}$ and $V_{\text{ref}}$ are the maximum draught and maximum speed, respectively, which are obtained from the vessel particulars dataset. The current power demand $W(t)$ of the main engine on board a vessel is calculated based on the so-called Admiralty formula (Faber et al., 2020):

$$W(t) = \frac{\delta_w}{\eta_w \, \eta_f} L(t), \tag{2}$$

where $\delta_w$ is a speed-power correction factor, $\eta_w$ is a weather correction factor, and $\eta_f$ is a fouling correction factor. The fouling is an overgrowth of invasive aquatic species on the submerged surface of a hull which can increase the friction of vessel movement through water (Townsin, 2003). The current speed of the vessel is calculated as a change in the position divided by the duration of a single movement. The change in the position is the distance along the geodetic line between two consecutive positions reported by the AIS transponder of the vessel. The duration of the movement is obtained from the absolute timestamps of the AIS position reports. It should be noted that the speed over ground (SOG) is an integral part of the dynamic AIS position report (ITU, 2014) and it can directly be used in the Admiralty formula. However, the design of AIS allows some data to be missing prior to a transmission of the AIS message. The AIS transponder, not to waste its reserved time slot for the upcoming AIS broadcast, automatically replaces unavailable parameters with default values, which explicitly indicate that certain sensor data on board a vessel are currently unknown. All parameters contained within a dynamic AIS position report except for MMSI are allowed to have an unknown value (ITU, 2014). Various analyses have shown that position reports containing unknown values of latitude and longitude have the lowest relative frequency of occurrence (Banyś et al., 2020). Therefore, the likelihood that a vessel's speed, derived from two consecutive positions, is available for emission calculation is quite high.

The weather and the fouling correction factor for different vessel types and sizes are taken from Faber et al. (2020). These factors account for the impact of meteorological conditions and hull fouling on a vessel's power demand. The speed-power correction factor is applied to cruise vessels irrespective of their size and to container carriers which have a capacity of at least 14500 twenty-foot equivalent units (TEU), again based on Faber et al. (2020). Analyses show that container vessels of this size are usually equipped with large engines but rarely proceed at their maximum speed, preferring rather an economical speed at lower engine load levels (Faber et al., 2020). The speed-power correction factors help to avoid an overestimation of the current power demand of those vessels.

The operational phase of a vessel is estimated based on her current speed and engine load as follows: *at berth* (when $V(t) \leq 1\,\text{kn}$), *anchored* (when $1\,\text{kn} < V(t) \leq 3\,\text{kn}$), *manoeuvring* (when $3\,\text{kn} < V(t) \leq 5\,\text{kn}$), and *voyage* (when $V(t) > 5\,\text{kn}$). The current power demand of auxiliary engines and auxiliary boilers fitted on board a vessel are categorised into different vessel types, sizes and operational phases. The following three conditions might occur:

– if the main engine power is not higher than 150 kW, then the current power demand of auxiliary engines and auxiliary boilers is set to zero,

– if the main engine power is between 150 kW and 500 kW, then the current power demand of auxiliary engines is set to 5% of the main engine power and the current power demand of auxiliary boilers is taken from the power demand parameter table,





– if the main engine power is higher than 500 kW, the power demand parameter table is applied to both auxiliary engines
       and auxiliary boilers.

The energy demand is computed as a product of the current power demand $W(t)$ of the main engine on board a vessel and
the time $\Delta t$ needed to move between two consecutive positions reported by the AIS transponder:

$$E(t) = W(t)\,\Delta t. \tag{3}$$

The baseline fuel consumption $F$ of a vessel is obtained from a parameter table which contains aggregated constant values in
relation to the main engine type, the fuel type and the year of construction of a vessel (Faber et al., 2020). To compute the
specific fuel consumption $S(t)$, a main engine load correction factor $C(t)$ based on the current engine load $L(t)$ is applied:

$$S(t) = C(t)\,F, \tag{4}$$

where the load correction factor $C(t)$ is provided by Faber et al. (2020) as:

$$C(t) = 0.455\,L(t)^2 - 0.71\,L(t) + 1.28. \tag{5}$$

The load correction factor $C(t)$ is a simplified attempt to indicate that every vessel has a certain optimal engine load level
at which the fuel consumption reaches its minimum (Al-falahi et al., 2018). It is assumed that the specific fuel consumption
shows a parabolic response to engine load levels. The fuel consumption curve is a specific property of each vessel. Here, we
assume that the fuel consumption of the auxiliary engines and the auxiliary boilers does not depend on the engine load. The
fuel consumption of the main engine and thus its emissions are set to zero if the main engine load $L(t)$ drops below 7%. This
kind of situation usually occurs at berth when the auxiliary engines and the auxiliary boilers are the only source of emissions.
The mass of fuel consumed during a single movement is then obtained as a product of the specific fuel consumption $S(t)$ and
the energy demand $E(t)$ as reported by AIS.

The final step of calculation of the emission fluxes followed one of the two alternative approaches: a computation based
on energy demand $E(t)$ is applied for $NO_x$, CO, $PM_{10}$, $PM_{2.5}$ and NMVOC, whereas a computation based on specific fuel
consumption $S(t)$ is applied for $CO_2$, $SO_x$ and BC. Depending on the applicable approach, either the energy demand or
the specific fuel consumption is multiplied by the emission factor of the species provided in the factor look-up tables (Faber
et al., 2020). The result is the mass of the species emitted by a given vessel during a single movement. The calculated masses of
species are stored in a temporary database format, enabling the accumulation of raw emission values based on grid coordinates,
time of occurrence, species, and vessel type. The raw emission data is then converted into the NetCDF files containing gridded
emissions at the resolution of $0.1° \times 0.1°$. The intermediate database storage of raw emission data allows the creation of
emission inventories for all vessel movements available in AIS data, as well as for selected fleets of specific vessel types.

### 2.2.2 Domestic navigation

In contrast to the international shipping sector, AIS data availability is more limited on inland waters. Fewer land-based an-
tennas along rivers compared to coasts, shading through the terrain and terrestrial installations, restricted data sharing and





regulatory differences between regions lead to a generally lower coverage. Therefore, bottom-up inventories using AIS data have been generated only on prominent waterway sections, which provide a higher density of AIS signals (e. g., Huang et al., 2022; Segers, 2021), but not on a larger scale. Top-down approaches use transportation statistics as an alternative to AIS data. Knörr et al. (2013) developed a top-down inland ship emission model utilising transport statistics of the German transporta-
tion emission model TREMOD (Allekotte et al., 2020). Based on goods statistics reported in ports and water locks by the Federal Statistical Office of Germany, the voyage and transport performance is assessed and differentiated by ship and cargo characteristics. Ship-type-specific energy demand and emission factors are developed to derive the total fuel consumption and emissions. The model, however, does not provide a spatial resolution of emissions in terms of an emission map. To address the problematic of low reliability in AIS for inland waters, Peng et al. (2024) proposed a combination of AIS and transportation
data. While AIS and ship technical data were used to reconstruct characteristic ship behaviour, statistics from the Ministry of Transport of the People's Republic of China were used to fully capture the activity level. Although this approach seems promising, the input data requirements are high and difficult to realise on a global scale. To the authors' knowledge, there is no spatially resolving global model of domestic navigation emissions available to date. Existing inland ship emission inventories are regionally limited and rely on region-specific data structures. For the global ELK inventory of domestic navigation a method
is presented limiting input data requirements to achieve a high regional flexibility while still addressing major differences in waterway systems.

For the ELK inventory the assessment of activity levels on the waterway systems consists of a primary approach, based on transportation data, and two fall back approaches, based on AIS data and on the assumption of homogeneously distributed activity, respectively. The primary approach for determining the activity levels is based on transportation data in tonne-kilometres
(tkm) with a high spatial resolution (state-to-state flows). This is the case for Europe, where transportation flows can be obtained from Eurostat (https://doi.org/10.2908/IWW_GO_ATYVEFL, last access: 10 June 2025) on the NUTS2 level (https://ec.europa.eu/Eurostat/web/nuts, last access: 10 June 2025), and for North America, where annual transportation reports between states were published by the Waterborne Commerce Statistics Center (2021). For China similar data is also collected by the Ministry of Transport of the People's Republic of China, as mentioned in Peng et al. (2024), yet they are
not publicly available. To achieve a spatial distribution, the transportation flows are modelled on a network graph of each inland waterway system with the Python library `networkx` (Hagberg et al., 2008). The path between two regions is identified utilising a Dijkstra shortest path algorithm, with the assumption that each ship performing the transportation task chooses the most direct route. Thereby all region-to-region transportation flows are distributed over the river network, thus determining the varying activity levels on each river section. If only total transportation volumes without region-to-region flows can be
obtained but there are AIS signals available, the first fallback approach is utilised. Although the AIS signal level is generally not sufficient to reconstruct the full activity level for inland ships, it is used to model the spatial distribution. This approach comes with the drawback that AIS-blind spots, for example where signals are fully shaded, are ignored in the activity distribution. This approach is applied for the China Yangtze River and Grand Canal region where the overall activity level could be estimated from values for 2018 published by Aritua et al. (2020). For the region of the Pearl River the overall activity level is
also obtained from Aritua et al. (2020), however the available AIS dataset does not contain any signals for this river. As the sec-





ond fallback approach the activity level is distributed homogeneously on the navigable sections of the river. Further waterway systems that are not considered in this dataset due to scarce data availability include the Amazon, Parana, Volga, Ganges, Nil, Congo, Mekong and Niger. Yet, according to the total transportation estimates of the World Wide Inland Navigation Network (Zentralkommission für die Rheinschifffahrt, 2012), these waterways contribute to less than 4% of total transported goods on
waterway systems.

To model the temporal distribution of the emissions, we follow the total waterway transportation volumes reported by the respective institutions. In the U.S. the Department of Transportation publishes monthly totals (https://data.bts.gov/stories/s/Monthly-Transportation-Statistics/m9eb-yevh/, last access: 10 June 2025), while for Europe quarterly data can be obtained from Eurostat (https://ec.europa.eu/Eurostat/databrowser/product/page/IWW_GO_QNAVE, last access: 10 June 2025). For China
no monthly waterway transport statistics are obtained, the temporal distribution therefore is applied homogeneously over the year.

In addition to the actual transportation tasks, ships also perform journeys without carrying any cargo and these are therefore not captured in the transportation statistics. Knörr et al. (2013) observed a difference in the percentage of empty trips between dry and liquid goods carriers by evaluating data from German water locks. Based on this data they concluded that dry cargo
ships perform 30% and liquid carriers 80% of their trips in unloaded conditions. This coincides with the average value of 47% for both, liquid and dry cargo, reported in the year 2023 for the Iffezheim water lock (CCNR, 2023). For the other regions no specific information on the percentage of empty trips could be found. To still capture the effect of empty trips, the values of 30% for dry goods and 80% for liquid goods are applied to other modelled regions as well, although this estimate might differ from the actual value due to the region-specific trading patterns.

To obtain the energy demand of domestic navigation, the average energy intensity of the transportation tasks is identified for different regions by a literature review. The energy intensity of the transportation task can be affected by multiple parameters, including ship characteristics, loading conditions, conditions of the waterway, water level, and the direction of flow (up- or downstream the river). Since the data availability is not given in all modelled regions, a simplified approach is chosen, utilising the average energy intensity typical for each region based on a literature review. Radmilovic and Dragovic (2007) conducted
a study on the energy intensity of various domestic transportation modes in Europe, concluding that inland waterway transport is the most energy-efficient, with a value of $0.423 \text{ MJ tkm}^{-1}$. This finding aligns with the energy intensity range of 0.38 to $0.52 \text{ MJ tkm}^{-1}$ reported by Knörr et al. (2013), who used a bottom-up calculation approach under average loading conditions for ships of different load classes. Consequently, we adopted the value of $0.423 \text{ MJ tkm}^{-1}$ as the reference for Europe. For the U.S., Bray et al. (2002) studied the energy intensity on different river sections with a top-down approach of transportation
statistics, lock data and vessel characteristics, and found an average value of $0.192 \text{ MJ tkm}^{-1}$. A later study by Belzer (2014) reported a lower value of $0.157 \text{ MJ tkm}^{-1}$, which is used in this study. According to these values the U.S. inland waterway system operates more than twice as efficient compared to the EU system. This may result from the wider and more navigable waterways and fewer locks. Furthermore the more widespread use of larger, non-propelled barges may enable more efficient transportation. For the remaining regions the energy intensity of the EU waterway system is applied as the more conservative
approach, assuming similar or worse conditions of waterways and low levels of large non-propelled barges.

Finally, the emission factors are determined based on the regional emission regulations and the age structure of the fleet extracted from Clarksons Research data. It is assumed that vessels built within an emission stage comply with the respective emission limits for $NO_x$, CO, HC and PM in each region. It is also assumed that the transportation tasks are distributed equally among the age distribution of the fleet resulting in the fleet emission factors shown in Table 1. The U.S. result in significant lower fleet emission values due to stricter emission limits of the U.S. Environmental Protection Agency (EPA) and a continuous replacement of vessels. In Europe, on the contrary, the aged fleet leads to higher average emission factors, despite the aligned emission limitations of the EU Non-Road Mobile Machinery (NRMM) regulations to the U.S. EPA regulations. China implements less strict emission limits compared to Europe and the U.S., starting with stage I only in 2015. Yet the overall younger age structure of the Chinese fleet leads to lower emission factors compared to Europe. In all regions the emission factors before any implementation of emission regulations are assumed to be 11 $g\,kWh^{-1}$ for $NO_x$, 5 $g\,kWh^{-1}$ for CO, 4.7 $g\,kWh^{-1}$ for HC and 0.4 $g\,kWh^{-1}$ for PM. Emissions of $CO_2$, $CH_4$, $N_2O$, NMVOC, BC and $SO_x$ are addressed independently of the emission stage but based on the general combustion characteristics of 4-stroke, high-speed diesel engines operating on distillate diesel oil, which represent the large majority of equipment used in domestic navigation. The values are harmonised with the respective equipment emission factors of the international shipping sector (Sect. 2.2.1). The same energy density of 42.7 $kJ\,kg_{fuel}^{-1}$ for distillate diesel oil (marine diesel oil) is applied to quantify the amount of burnt fuel for fuel-based emission factors.

**Table 1.** Average fleet emission factors of regulated pollutants from inland navigation in each modelled region. Units are $g\,kWh^{-1}$.

| Species | Unites States | Europe | China |
|---------|---------------|--------|-------|
| $NO_x$ | 5.62 | 8.5 | 8.27 |
| CO | 4.63 | 4.8 | 4.56 |
| HC | 2.78 | 3.87 | 3.20 |
| PM | 0.25 | 0.28 | 0.3 |

As noted in the analysis of available AIS data for domestic navigation, there is insufficient data in the investigated regions. Therefore, we conclude this section by illustrating a concept for using satellite data that could help identifying vessels on rivers and channels. This concept considers the use of freely available satellite data for scientific research, with a specific focus on optical and Synthetic Aperture Radar (SAR) data. To minimise the number of satellite scenes to be processed, a database is created which contains only those metadata of the radar and optical scenes that have an overlap with inland waterways. A second approach is to use existing ship detection algorithms to identify vessels over terrestrial areas. Ship detection algorithms mostly use land masks to determine whether a raster pixel is land or water. Also for this case, we identify and analyse various data sources in more detail with regard to their usability. As for the first approach, the detected ships are stored in a database and compared with the few existing AIS data to determine only the missing vessels in the AIS detections. These can then be passed on for validation purposes or to supplement missing vessel positions. Since the existing ship detection algorithms are developed for the open sea, the results must be evaluated to identify false or missing detections. When using the existing algo-



rithms, it should also be noted for which satellite data they were developed and which land masks can be used. This may limit the availability of usable, freely accessible satellite data, reducing the number of additional ship positions that can be identified.

One of the possible sources are radar data, which, compared to optical data, is almost independent of weather conditions and can record signals even during darkness. Radar data is generated by SAR sensors, which operate in the microwave frequency range of the electromagnetic spectrum (Skolnik, 2008). SAR sensors actively illuminate an area with microwaves and map it in two dimensions. In addition to the position, the ship width and length also play a role in determining the emissions in relation to her class or type, therefore the use of radar signatures of ships to derive ship length and width from the dimensions of the

signatures is widespread (Tings et al., 2016). The ship radar signatures in the SAR images usually have larger dimensions than the actual dimensions of the ship. The reason for such dimensional distortions is the higher normalized radar cross section (NRCS) at the boundary between the ship hull and the water surface as well as at the ship superstructure, which lead to different reflection effects (Crisp et al., 2004; Tings et al., 2016). The smearing of moving point targets also leads to a distortion of the ship dimensions (the so-called azimuth smearing effect; Skolnik, 2008; Tings et al., 2016). The overestimation of the

ship dimensions can be reduced by applying analytical and empirical methods (Tings et al., 2016). If AIS data is available for the radar scenes, these can be used to determine the ship dimensions, as they are more accurate than the derivation from the ship radar signatures. There are a large number of research teams working on ship detection based on radar data (Vachon et al., 1997). Since the existing ship detection algorithms are designed for the open sea, their applicability to inland navigation remains uncertain. Limitations include wind speeds, radar angle of incidence, satellite flight direction, ship speed, decay rates

of wakes, ship offset in azimuth direction, waves, width of the river compared to the open sea, other metal objects (bridges, bollards, harbour superstructures, quay walls, buildings, etc.) or the ship length. On the open sea, the detectable ship lengths are between 5 and 350 m.

## 2.3 Aviation

Air transport is essential for long-distance travel in a relatively short time and for the rapid transport of freight between

continents. Due to the increasing relevance of environmental effects and climate impacts of air traffic, that is responsible for 3.5% of global anthropogenic radiative forcing in 2018 (Lee et al., 2021), several studies have focussed on the generation of bottom-up quantification of aviation emissions. The underlying methodology of the ELK inventory for aviation emissions is an advancement of the approach by Weder et al. (2025b), where gridded aviation emission inventories for passenger flights for the years 2015, 2019 and 2020 were generated within the DLR project Transport and Climate (TraK) using Reduced

Emission Profiles (see Sect. 2.3.2), great circle routes and fuel-optimized cruise altitudes. The idea of emission quantification on a subsectoral level including cargo air traffic developed in this work has also been followed in Graver et al. (2020) for the years 2013, 2018 and 2019, but only for $CO_2$ and without modelling three-dimensional distributed data. Teoh et al. (2024) used detailed trajectories from Automatic Dependent Surveillance - Broadcast (ADS-B) data and EUROCONTROL Base of Aircraft Data model family 3 and 4 (BADA3 & BADA4) flight performance models to derive 3D annual emission inventories for 2019,

2020 and 2021 to evaluate the effects of COVID-19 pandemic on air traffic volume and emissions. Quadros et al. (2022) also investigated pandemic effects and covered the years 2017-2020 based on BADA3 aircraft performance and monthly averages



of ADS-B routing data. Eyers et al. (2005), Wilkerson et al. (2010) and Simone et al. (2013) produced annual bottom-up aviation emission inventories for 2002, 2004, 2005 and 2006. To model a detailed aviation emission inventory, a traffic demand and fleet scenario in the form of a flight schedule is required to construct the global network. 4D aircraft trajectory profiles, incorporating detailed information on fuel consumption and emission factors for each flight phase, are adjusted for wind affects and projected along actual flight paths to achieve a more realistic spatial distribution. The applied methodology is described in detail in the following subsections.

### 2.3.1 Air traffic schedule and fleet composition

In the ELK inventory, the underlying air traffic for the reference year 2019 is calculated with the model FORMO (Gelhausen et al., 2019) based on input from several databases like Airport Council International (ACI, 2019), Sabre Market Intelligence (Sabre, 2019), Official Airline Guide (Reed Travel Group, 2019), IATA Air Freight Bills (CASS; IATA, 2019) and Cirium Fleet Analyzer (Cirium, 2019). These databases provide information on the global fleet composition, scheduled commercial passenger flights, transported passengers and ticket fares. For passenger air traffic the model synchronizes the various databases: OAG, Sabre and Cirium to match passenger and flight volume by aircraft type and flight time for passenger transport. Cirium is used to determine the number of aircraft and aircraft type for the corresponding flight movement volume. This involves standardizing and cleaning data and removing data not needed. The data for a global air freight is largely based on aircraft movements derived from the ACI, OAG freight flights on individual routes and freight volumes by origin and destination airport pair from the CASS. Aircraft data from Cirium plays a major role in obtaining information on individual aircraft and airlines. In addition to cargo-only scheduled flights, a large share of air freight is transported as belly freight by passenger aircraft and by dedicated freight integrators like DHL, FedEx or UPS, which do not appear in official flight schedules from Sabre. The major challenge is that there is no standardized and complete database for air freight, so that at least some of the freight flows and related flights need to be modelled by comparing different databases, cleaning data, and modelling missing data. Cross-checking for plausibility and further quality checks are important for a consistent database of air freight. The result consists of global weekly flight plans providing information on origin and destination airport, local departure and arrival time, aircraft type, seat capacity as well as seat load factor (passenger transport only). The public domain airport database OpenFlights (openflights.org, last access: 10 June 2025) provides information on location and elevation of the origin and destination airports and the time zones, which are necessary to convert the local departure and arrival time into Coordinated Universal Time (UTC) units. For the ELK inventory, flight plans for both passenger and cargo air transport are created for the week 28/01/2019 to 03/02/2019, representative of the winter season, and for the week 22/07/2019 to 28/07/2019, representative of the summer season. These seasonal air traffic data enable an extrapolation to the entire year, by considering five months of winter season (November to March) and seven months of summer season (April to October). To gain a more specific view on the emission quantities by aircraft size, the global fleet of passenger aircraft is split into three subsectors based on the seat number and the maximum range, namely regional (seat number < 100 and maximum range < 1500 nautical miles (NM)), single-aisle ($100 \leq$ seat number < 220 and 1500 NM $\leq$ maximum range < 4000 NM) and wide-body aircraft (seat number $\geq$





and maximum range $\geq$ 4000 NM). If an aircraft exceeds either the maximum range or the seat number's threshold, the aircraft is categorized into the next larger category.

### 2.3.2 Aircraft performance and trajectory modelling

Flight trajectories are modelled with the Trajectory Calculation Module (TCM), an aircraft trajectory simulator developed at DLR Institute of Air Transport based on the Total Energy Model (Linke, 2016). For the atmospheric background data, Inter-
national Standard Atmosphere (ISA; ICAO, 1993) conditions are used. TCM performs a forward integration in high temporal resolution (typically 10 s) of the aircraft state along a four-dimensional flight trajectory defined by characteristic flight phases. Required aircraft characteristics (e. g., weight, geometry) and flight performance parameters are provided by BADA4 (version 4.2) from EUROCONTROL (Nuic et al., 2010). In order to cover the global fleet to the highest possible extent, we supplement BADA4 data by internally developed aircraft performance models of aircraft types recently put into service (e. g., Airbus
A321neo, A330-900neo, A220-300) or missing aircraft type in BADA4 of enhanced relevance for regional air traffic (Canadair Regional Jet 900), see Woehler et al. (2020). To facilitate the calculation of a large set of trajectories in an efficient manner, Reduced Emission Profiles (RedEmP) are derived (Linke, 2016) representing a pre-calculated dataset of relevant flight performance parameters along standardised flight trajectories, starting and ending at sea level and non-georeferenced, with a reduced number of datapoints. The RedEmP database contains trajectories for every available aircraft type with different flight lengths
in 100 NM steps, different altitude settings and take-off masses (characterised by load factors), as these factors significantly influence flight performance, fuel consumption and emission factors. Resulting trajectories are then reduced to flight phase characterising data points so that between 13 and 25 trajectory points can be used to efficiently describe every trajectory of a given flight plan. The stored parameters contain several state variables, such as altitude, true air speed, thrust, rate of climb and descent, and fuel flow along the entire trajectory. Flight phase dependent emission flows for various species are added to
the database of RedEmP in a post-processing step (see Sect. 2.3.3).
Compared to the previously developed RedEmPs (Linke, 2016; Weder et al., 2025b), here we extend the data set by considering different flight altitudes, namely fuel-optimized cruise altitudes, characterized by a minimum fuel flow and the operation of step climbs in the course of the cruise phase in case the next flight level is more fuel-efficient, and constant cruise altitudes (Zengerling et al., 2022). Furthermore, we improve the resolution of the assumed load factors by covering an interval width of
0.025 for load factors higher than 0.7. Finally, we extend the set of covered parameters by engine thrust and propulsion efficiency. Aircraft performance models from BADA4 and DLR cover more than 97% of the total annual available seat kilometers (ASK) of the underlying flight plans, thus ensuring a high coverage of the global fleet in the emission inventory. Nevertheless, BADA3 models (version 3.11), that are available for a plenty of turboprop and piston aircraft models, are additionally used to reduce remaining gaps in fleet coverage and to complete the regional and cargo fleet. BADA3 vertical profiles of aircraft
performance are forward integrated and vertically interpolated to generate simple trajectories with climb, cruise and descent phase with a payload-dependent rate of climb and cruise altitude.



### 2.3.3 Emission factors

$CO_2$ and water vapour ($H_2O$) emissions are scaled linearly to fuel consumption with constant emission factors of $3.159 \, \mathrm{kg \, kg_{fuel}^{-1}}$ and $1.237 \, \mathrm{kg \, kg_{fuel}^{-1}}$, respectively, assuming a stoichiometric combustion (Wilkerson et al., 2010). For the calculation of non-linear emissions of CO and HC, the Boeing Fuel Flow Correlation Method 2 (DuBois and Paynter, 2006) is applied, while $NO_x$ emissions are derived as $NO_2$ mass equivalent with the DLR methodology by Deidewig et al. (1996). NMVOC emissions are derived from HC emissions by applying a scaling factor of 1.0947 (Jelinek et al., 2004). For the application of fuel flow correlation methods, the required atmospheric background data is taken from ISA (ICAO, 1993), modified by a constant average relative humidity of 60% as used in Kim et al. (2005). Emission factors that have been quantified from test bench measurements for four thrust settings as defined in landing-takeoff-cycle (LTO; ICAO, 2008) are taken from ICAO Engine Emission Databank v29a (EEDB, https://www.easa.europa.eu/domains/environment/icao-aircraft-engine-emissions-databank, last access: 10 June 2025) for jet engines and from piston engine emission data from the Swiss Federal Office of Civil Aviation (FOCA, 2007). For turboprops no thrust-dependent emission factors are available, so constant values are used for $NO_x$, HC, CO and BC, as derived from fleet averages in Wilkerson et al. (2010). BC emissions are build on smoke numbers from EEDB and the methodology from Döpelheuer (2002). $SO_2$ emission quantities scale linearly with the fuel consumption, but the emission factor depends on the regional average fuel sulphur content as analysed in CRC (2012) for year 2010 and is set for each flight individually with regard to the continental area of the departure airport. The OC emission factor is set constant to $0.02 \, \mathrm{g \, kg_{fuel}^{-1}}$ (Stettler et al., 2011). Non-volatile particular matter number (nvPMn) and mass (nvPMm) emission factors for LTO cycle are taken from EEDB for jet engines and interpolated linearly depending on the relative thrust setting from the trajectory simulations. For engines without LTO particle emission data available, fleet average values of $0.080 \, \mathrm{g \, kg_{fuel}^{-1}}$ (Stettler et al., 2013) and $10^{15} \, \mathrm{\# \, kg_{fuel}^{-1}}$ (Schumann et al., 2015; Teoh et al., 2020) are used constantly for all flight phases.

In addition to GHG and SLCF emissions, propulsion efficiency $\eta$ is a relevant parameter for the contrail modelling and is therefore included in the ELK inventory. The propulsion efficiency depends mainly on the flight phase and can be expressed as a function of thrust setting, speed and fuel flow:

$$\eta = \frac{v_{\mathrm{TAS}} T}{\dot{m} Q} \tag{6}$$

with $v_{\mathrm{TAS}}$ as true air speed, $T$ as engine thrust, $\dot{m}$ as fuel flow rate and the specific combustion heat for aviation fuel $Q$ = 43 $\mathrm{MJ \, kg^{-1}}$ (Schumann, 1996). The resulting emission flows are additionally stored in the RedEmP database enabling a combined evaluation across a given flight plan.

### 2.3.4 Routing and flight path data

To enhance the realism of routing in the emission inventory, detailed waypoint and altitude profiles of approximately one million global flights operated in year 2019, mainly from week 05 and week 30, are quality-checked, filtered and stored in a database. The actual 3D flight paths are derived from ADS-B mode S data from OpenSky Network (Schäfer et al., 2014; Strohmeier et al., 2021), covering predominantly domestic flights over Europe, North America and some regions in Asia





(e. g., Japan, India and Australia) depending on the density of the ADS-B receiver network. Additionally, model 3 data from

EUROCONTROL Demand Data Repository 2 (DDR2) (Urjais, 2022) with flight path profiles of inner-European flights and
intercontinental flights from Europe to various worldwide destinations and vice versa as well as international flights crossing
European airspace contribute to the routing data. For the remaining flights in the schedules, where no flight path profile can be
identified in the datasets, the great circle route between the origin and destination airport is used.

### 2.3.5 Wind impact

To consider head and tail wind effects on flown ground distance, the statistical approach by Swaid et al. (2024) is used to
derive the average of the wind-corrected ground speed based on local annual wind statistics. For this purpose, global grids of
horizontal wind data at $0.25° \times 0.25°$ resolution on various pressure levels from the ERA-5 reanalysis data (Hersbach et al.,
2023) at 0-, 6-, 12- and 18-UTC for every day in the year 2019 are vertically interpolated to obtain a dataset for each flight level
from 0 to 44000 ft altitude (and the same value is applied for higher levels up to 47000 ft). The vertically interpolated wind
data is binned in each grid cell with a wind speed increment of 5 m s$^{-1}$ and 10° wind direction intervals and converted into
an annual wind rose statistic that consists of 1460 values per grid cell. For each origin-destination pair and aircraft type within
the weekly flight plans, the great circle route is divided into ten segments of equal length. Their mean headings, the aircraft
specific mean cruise $v_{TAS}$ at the mean fuel-optimized cruise altitude are then used to derive the resulting statistical distribution
of ground speed, incorporating the local annual wind statistics. To obtain a dimensionless wind impact scaling factor $\epsilon$, the
flight's mean cruise $v_{TAS}$ is divided by the average of the segment specific ground speed (GS) distribution:

$$\epsilon(i) = \frac{v_{\text{TAS}}(i)\,t(i)}{v_{\text{GS,ave}}(i)\,t(i)} = \frac{d_{\text{air}}(i)}{d_{\text{ground}}(i)} \tag{7}$$

with $v_{\text{TAS}}(i)$ the mean cruise true air speed in a trajectory segment $i$, $v_{\text{GS,ave}}(i)$ the mean ground speed depending on the wind
statistic along the trajectory segment $i$, $t$ the flight time of the trajectory segment $i$ and $d_{\text{air}}$ and $d_{\text{ground}}$ the air and ground
distance of the trajectory segment $i$, respectively. A value of $\epsilon > 1$ ($\epsilon < 1$) represents head (tail) wind situations. The mean
wind effect scaling factors $\epsilon$ are used to adjust the air distance for the trajectory segments of each route in the flight plan.

### 2.3.6 Model workflow

For each flight that is scheduled in the seasonal weekly flight plans, the flight route database is scanned, and up to eight
appropriate flight path datasets for the respective origin and destination pair are selected in a random-pick approach (see Sect.
2.3.4). Based on the wind-corrected distance, the load factor and aircraft type, the best-fitting RedEmP is picked or if not
available, a simple trajectory is generated out of the BADA3 model (see Sect. 2.3.2). The cruise altitude of the RedEmP is
assumed fuel-optimized including step climbs during longer flight distances for the first four selected flight path profiles and
for the remaining flight path profiles RedEmPs with constant discrete cruise flight levels are used, that have been identified as
the main cruise flight level of the route profile. For all available routes, the cruise phase of the selected RedEmP with a discrete
distance (100 NM steps) is extended or truncated to the actual flight length and the climb and descend phases are adjusted
to the elevation of the origin and destination airports. Subsequently, all available route profiles are weighted to represent one

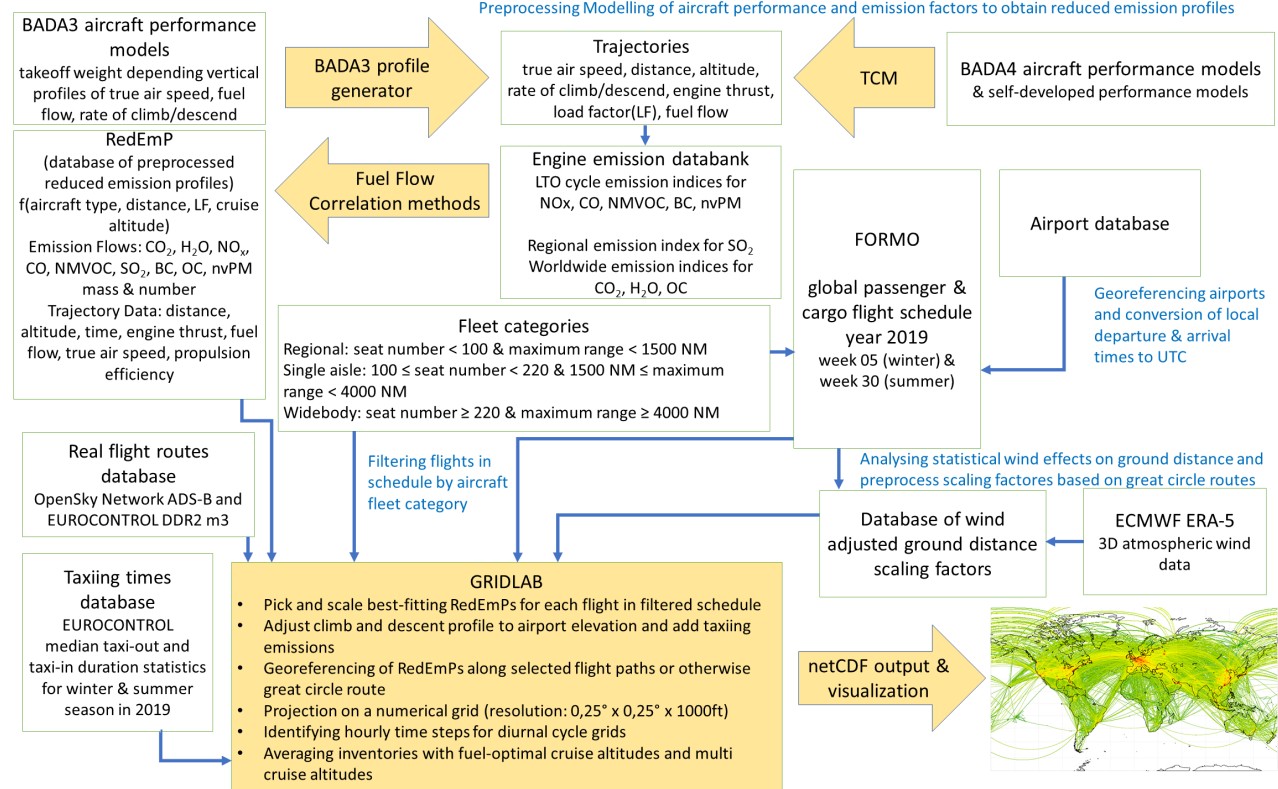

**Figure 4.** Workflow of the aviation emissions model, showing the required input data and a schematic depiction of the methodology.

single flight. The final emission inventory thus consists of 50% of trajectories with fuel-optimized cruise altitudes and 50% of various constant altitudes each. Emissions on ground caused by taxiing at the airports are derived with the measured emission factors and fuel flow in idle mode from the underlying emission factor databases (see Sect. 2.3.3) and multiplied with the airport-specific median taxiing times from the EUROCONTROL seasonal statistics for year 2019 (https://www.eurocontrol.in

t/publication/taxi-times-summer-2019, https://www.eurocontrol.int/publication/taxi-times-winter-2018-2019, last access: 10 June 2025). These times are available for more than 400 airports worldwide, otherwise intervals of 19 minutes for taxi-out and another 7 minutes for taxi-in as defined in LTO cycle (ICAO, 2008) are applied. For all flights, take-off emissions are represented by engines running 0.7 minutes in take-off mode, also following the LTO cycle. The resulting emission quantities are attached at the beginning and at the end of the emission profile at the airport location and allocated to the grid layer of the

respective airport elevation. Due to the unavailability of LTO cycle emission factors for turboprop engines, taxiing emissions cannot be taken into account for flights operated by turboprops. Finally, the modified emission profile is projected on the georeferenced route and converted to the 3D numerical grid with a resolution of $0.25° \times 0.25° \times 1000$ ft. The grid layers are arranged in a way so that each discrete flight level (FL) is located in the centre of each grid layer (e. g., grid cell layer from 36500 ft to 37500 ft for FL 370), starting from sea level at the lowest layer. The annual total emission inventory is derived by





an upscaling and weighted mean of the seasonal weekly inventories (see Sect. 2.3.1). A more detailed temporal discretisation of emission grids in up to hourly resolution is performed to show diurnal effects of air traffic volume on emission distribution. The entire model workflow including the required input datasets and the preprocessing steps is depicted in Fig. 4.

## 2.4    Energy for transport

This section describes the estimation of global emissions from the energy-related production of fuels for the transport sector,
that is, the indirect emissions from transport in the energy sector. According to the International Energy Agency (IEA), in the reference year 2019 about 91% of the fuels used in the transport sector were based on petroleum, while only 4% of the final consumption was natural gas, 3% biofuels and around 1% of the transport fuels were based on electricity (https://www.iea.org/data-and-statistics/charts/energy-consumption-in-transport-by-fuel-in-the-net-zero-scenario-2000-2030, last access: 10 June 2025). Thus, although the share of non-petroleum-based energy carriers and especially e-fuels is expected to
increase in the future, we focus here on petroleum-based fuels as the main energy carriers for global transport in 2019. Different studies indicate that refining represents the most important sector in terms of indirect emissions of petroleum-based transport fuels, although to varying degrees. According to Rahman et al. (2015), refining is responsible for 56% of GHG emissions for gasoline production in a North American refinery, while 25% and 15% were allocated to crude oil recovery and fugitive emissions, respectively. In a similar study for gasoline production in Chile, 87% of indirect GHG emissions from gasoline
production stem from refining activities, and 13% from oil extraction activities (Morales et al., 2015). That study also suggested that refining is the most important sector for indirect emissions when considering all other environmental impact categories (e. g., terrestrial acidification). However, not all emissions from refining can be allocated to the transport sector. Although the majority of refinery products is usually used for transportation purposes, there are also other uses of fuels (e. g., electricity generation) and other refinery products besides fuels (like lubricants and chemical base materials) which are dependent on
oil refining. Against this background, for the estimation of indirect emissions from the transport sector in the ELK emission inventory, we determine transport-induced global emissions from crude oil refining. Emissions from other upstream processes (e. g., fugitive emissions or crude oil recovery) were omitted because they are difficult to quantify and distribute geographically on a global scale. We first estimate total emissions from crude oil refining on refinery level for refineries worldwide. We then calculate the share of these emissions which can be attributed to the fuels used in the transport sector, deducting emissions for
refinery products which are consumed in other sectors. We estimate transport-induced emissions for 549 global refineries in 85 countries.

In the following, we first describe the methodology used to estimate the emissions for 2019. We then explain the approach of determining the share of transport-induced refinery products. Finally, we show how this share is applied to the total amount of refinery emissions to obtain only transport-induced refinery emissions. Further details on the methodology are provided in
the Supplement.





### 2.4.1 Refinery-level emission data

The majority of refinery emissions originates from combustion processes, according to Sun et al. (2019) who analysed U.S.-American refinery emissions in 2014. For the species considered in the ELK inventory, 66% to 87% of all emissions were caused from combustion with the exception of NMVOC, where this applies to only 9% of the emissions. As a result, fugitive emissions and emissions from other refining processes play a minor role. Estimating these emissions consistently on a global scale is both complex and highly uncertain due to variations in individual processes and plant configurations. Therefore, to ensure a more consistent and well-founded inventory, only emissions related to combustion are taken into account.

Depending on data quality and availability, four approaches are employed to estimate refinery-level combustion emissions on a global scale. First, whenever possible, publicly reported emission data on refinery-level for 2019 or a similar year are used, as they are considered to provide high-quality emission estimates. These are available for refineries in EU countries from the European Pollutant Release and Transfer Register (E-PRTR; https://www.eea.europa.eu/data-and-maps/data/member-states-reporting-art-7-under-the-european-pollutant-release-and-transfer-register-e-prtr-regulation-23/european-pollutant-release-and-transfer-register-e-prtr-data-base, last access: 10 June 2025), the National Emissions Inventory of the U.S. (https://www.epa.gov/air-emissions-inventories/2017-national-emissions-inventory-nei-data, last access: 10 June 2025) and the National Pollutant Release Inventory of Canada (Greenhouse gas reporting: facilities; https://www.canada.ca/en/services/environment/pollution-waste-management/national-pollutant-release-inventory.html, last access: 10 June 2025). If certain pollutants are not reported due to thresholds defined in the reporting guidelines, a gap-filling routine using species-to-species ratio factors is applied. Second, aggregated country-level data for the refinery sector (IPCC sector 1.A.1.b) from reported emission data of the United Nations Framework Convention on Climate Change (UNFCCC; https://unfccc.int/ghg-inventories-annex-i-parties/2023, last access: 10 June 2025) and the Convention on Long-Range Transboundary Air Pollution (CLRTAP; https://www.eea.europa.eu/en/datahub/datahubitem-view/5be6cebc-ed2b-4496-be59-93736fc4ad78, last access: 10 June 2025) are used for several countries. Third, if no other data is available, emissions are estimated based on refinery consumption data taken from the IEA World Energy Balances (https://www.iea.org/data-and-statistics/data-product/world-energy-balances, last access: 10 June 2025) and the default emission factors for the refinery sector from EMEP/EEA guidelines 2023 for energy industries (https://www.eea.europa.eu/publications/emep-eea-guidebook-2023/part-b-sectoral-guidance-chapters/1-energy/1-a-combustion/1-a-1-energy-industries-2023/view, last access: 10 June 2025). Finally, a separate approach is chosen for India, where emission factors are derived from country-specific emission limits for refineries and emissions calculated with consumption data from the IEA energy balance of India. If emissions are only available at the country level (cases 2 to 4), they are distributed to refineries based on their oil processing capacity, using refinery data from the Global Energy Observatory (https://globalenergyobservatory.org/list.php?db=Resources&type=Crude_Oil_Refineries, last access: 10 June 2025).

### 2.4.2 Transport-induced refinery products

Using the IEA extended energy balances, a method has been developed to determine how much of the refinery output of each country is used in the domestic transport sector for each country. Since refinery products are often exported (e. g., to





countries which do not have refineries, among others), fuel export data from Eurostat database for EU countries (https://ec
.europa.eu/Eurostat/web/international-trade-in-goods/database, last access: 10 June 2025) and UN Comtrade database
(https://comtradeplus.un.org/TradeFlow, last access: 10 June 2025) for the reference year 2019 are used in combination
with 2019 IEA energy balances to also consider fuels produced in domestic refineries which are consumed by the international
transport sectors. For each country, it is possible to estimate the share of domestically produced fuels from refineries which are
consumed for transportation purposes domestically or in transport sectors of other countries.

### 2.4.3 Merging refinery-level emission data and transport-induced refinery output

As the last step, the fuel-specific share of transport-induced refinery output has to be combined with the refinery-level emission
data. The refinery-level emissions are distributed to the refinery products to get the fuel-specific emissions of each refinery first.
This distribution is carried out for each refinery based on the energy content of the different refinery products generated in the
year 2019 in the country where the refinery is located. By multiplying the resulting fuel- and refinery-specific emissions with
the shares of transport-induced fuels, we obtain the transport-induced share of global refinery emissions for the year 2019.

## 3 Uncertainty assessment

The components and assumptions used in the compilation of the ELK emission inventory are uncertain to varying degrees.
Uncertainty estimates are a relevant tool for inventory development, as they identify where data and methodology need to
be improved. Furthermore, as emission levels are expected to decrease in the future, it is important to pay more attention to
estimating uncertainty to better understand which areas of emission calculation need to be enhanced to prevent inaccurate
data from misleading trends or influencing policy decisions in a suboptimal way. Therefore, the information on uncertainty
is not intended to dispute the validity of inventory estimates, but to help set priorities for improving accuracy in the future.
However, uncertainty reporting is challenging: a formal sensitivity analysis does not guarantee that all sources of uncertainty are
captured or quantified. Per IPCC guidelines, quantitative GHG uncertainty estimates focus only on parameter uncertainty and
do not consider structural uncertainty in models used for estimation or the potential uncertainty from incorrect data reporting
or missing emission sources (Smith et al., 2022). Correlation of errors across sectors or countries can increase uncertainty
estimates, but this is difficult to estimate and may not be considered in a formal uncertainty analysis (Solazzo et al., 2021).
Furthermore, some sources of uncertainty (sampling error, instrument accuracy) may generate well-defined estimates of the
range of potential error, while other sources may be much more difficult to characterise. Overall, the multidimensional nature
of the uncertainties (in terms of sector, species, time, and space) makes a quantification challenging.

For these reasons, we adopt an expert elicitation approach, combining a qualitative evaluation with a quantitative assessment
that summarizes the previously defined uncertainties. The elicitation was performed by initial proposals by the expert for the
specific data followed by a group discussion; in the end, the group came up with one agreed score. The experts consulted are
involved in non-exhaust measurements on chassis dynamometers for obtaining emission factors, in the development of the
models used, in the compilation of the ELK emission inventories, with the support of atmospheric modellers representing the





**Table 2.** Uncertainty factors characterising activity data and emission factors.

| Sector | Uncertainty factor | Explanation |
| --- | --- | --- |
| All | Delimitation | Does the delimitation to other (sub-)sectors and species lead to double counts or disregarded categories? |
| | Covariance | Are possible correlations between the activity level and the emission factors inadequately addressed (e. g., elevated emission in the start-up of the equipment)? |
| | Environmental conditions | Which environmental conditions influence the activity level? Have the emission factors been inadequately addressed? |
| | Input proxy data | What proxy data is used as input for deriving the activity level or the emission factors? |
| | Temporal variation | What are the specific uncertainties associated with the calculation of intra-annual values for both activity levels and emission factors? |
| | Spatial distribution | What are the uncertainties specifically arising from the approach for spatial disaggregation used (2 dimensions for all sectors except aviation, here 3 dimensions)? |
| Land transport | Type | What uncertainties are specific to freight and passenger transport? |
| | Mode of transport | What are the specific uncertainties associated with road and rail transport? |
| | Means of transport | What are the uncertainties specific to the means of transport (car, 2-wheeler, bus, rail vehicles, LCV, HFT)? |
| Shipping | Operational area | What are the uncertainties specific to domestic navigation and international shipping? |
| | Means of transport | What are the uncertainties specific for domestic navigation and international shipping vessel categories? How does this affect the magnitude of the uncertainty in the emission inventory? |
| Aviation | Type | What are the uncertainties specific to freight and passenger transport? |
| | Means of transport | What are the uncertainties specific for regional, single aisle, wide body, and cargo aircraft? |
| Energy | Type | What are the uncertainties specific to products from refineries? |




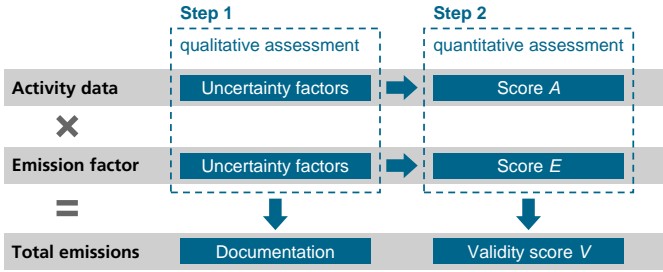

**Figure 5.** Flow chart for uncertainty assessment in the ELK emission inventories.

user's side. According to Hiraishi et al. (2000a, b), a key requirement in making estimates of uncertainty by expert judgements is that all the possible sources of uncertainty are considered. Our concept is inspired by Mastrandrea et al. (2010), who suggest using a calibrated language for key findings and providing comprehensive outlines describing the evaluation process. Therefore, the qualitative assessment is based on predefined uncertainty factors to ensure intersubjectivity and comparability across the
740 sectors analysed, and the results are provided as background information for later usage. For the quantitative assessment, the experts are guided by a list of criteria to be considered when assigning uncertainty scores. Since emission inventories are principally the product of activity data and emission factors, these two sources of uncertainty are reflected by the assessment procedure displayed in Fig. 5.

**Table 3.** Assessment criteria to be considered in the evaluation of uncertainty scores $A$ and $E$.

| Aspect | Explanation |
| --- | --- |
| Coherence | Are the data geographically coherent? |
| Consistency | Is there a high degree of consistency with other data sources? What are the areas of consensus and what are places where no consensus exists? |
| Data origin | Are these model results, (in-situ) measurements, or expert judgements? Do measurement or sampling errors play a substantial role? |
| Standardization | Is the method a de-facto standard in the community? |
| Traceability | Can the origin of the data be traced? |
| Up-to-dateness | Are the data up to date for the reference year or outdated? |

Both uncertainty scores $A$ (activity data) and $E$ (emission factors) are based on the qualitative evaluation results and are to
745 be spatially differentiated per area unit $i$. Since the uncertainty per emission factor $n$ can vary substantially, this is reflected by the resulting validity score $V_{ni}$. For the calculation of $V_{ni}$ the following equation is used:

$$V_{ni} = \frac{\alpha\,A_i + \beta\,E_{ni}}{\alpha + \beta} \tag{8}$$





**Table 4.** Values and interpretation of uncertainty scores.

| Score value | Qualifier | Interpretation |
|---|---|---|
| 1 | very robust | This uncertainty score is assigned to those areas that have values with the lowest uncertainty. This uncertainty score identifies areas of least concern regarding the validity of the resulting emissions. |
| 2 | robust | This uncertainty score is assigned to those areas that have values with a low uncertainty. This uncertainty score identifies areas of minor concern. |
| 3 | neither limited nor robust | This uncertainty score is assigned to those areas that have values with a medium uncertainty (e. g., a range can be given). This uncertainty score identifies areas of moderate concern. |
| 4 | limited | This uncertainty score is assigned to those areas that have values with a relatively high uncertainty. |
| 5 | very limited | This score is assigned to those areas that have values with the highest uncertainty (e. g., only an order of magnitude can be given). This uncertainty score identifies areas of major concern. |

The weighting factors $\alpha$ and $\beta$ can be chosen within a reasonable range to reflect the importance of emission factors and activity data in the emission calculation, but must be identical for each sector to ensure a comparable assessment. For the ELK emission inventory, the weighting factors are set to $\alpha = \beta = 1$. A validity score is only assigned to regions where the emissions by the respective sector are defined (e. g., no uncertainty scores are given for land transport over water bodies).

For step 1, the qualitative uncertainty assessment, predefined uncertainty factors support the holistic assessment of all sources of uncertainty. Table 2 exhibits the factors common to all sectors as well as the sector-specific ones. Each uncertainty factor is to be addressed for both activity data and all emission factors relevant for the corresponding sector. For step 2, the quantitative uncertainty assessment, the uncertainty scores for activity data and emission factors are to be differentiated in the spatial dimension. For all sectors except land transport, the IPCC AR6 scientific regions (Iturbide et al., 2020) are used (Fig. S2). For land transport, the amount and quality of input data differ significantly between countries; therefore, the uncertainty scores are obtained at the country level. Table 3 lists the assessment criteria which are intended to guide the expert judgement and make the ratings consistent between species as well as sectors.

The uncertainty is described by one of the qualifiers displayed in Table 4. The uncertainty scores show areas with relatively low ('very robust') to high ('very limited') uncertainties – the assignment of low/high uncertainty is specific for the inventory of a sector (land transport, shipping, aviation, energy for transport). If values are transferred from one $n$ to another due to the lack of reference data, the uncertainty score will be at least one unit higher than for the study region.

The qualitative assessment of uncertainties provides the rationale for assigning uncertainty scores for activity data and emission factors that are differentiated by geographic regions. To identify the major drivers of uncertainty in the resulting inventories, the effort devoted to characterising uncertainty factors should be proportional to their importance.



## 3.1 Land transport

In comparison to other modes of transport, land transport is less regulated and more influenced by human behaviour. The lower degree of organisation at the international level results in the absence of comprehensive timetable data of vehicle movements.
Consequently, the reconstruction of activity data from diverse sources of varying quality is necessary. This process inevitably introduces gaps in the data, particularly in land-based freight transport.

### 3.1.1 Activity data

The driving behaviour and conditions under which trips are made are only partially considered: while the discrepancy between emissions during a measurement cycle and in real world is known, it is only partially reflected by the applied emission factors
and thus in the inventory. For internal combustion engine (ICE) vehicles, the emission factors are increased in the cold start phase. Since the number of trips and their distribution over time are not reflected by the activity data, the total emissions are slightly underestimated. For temporal disaggregation of the annual volumes from both passenger and freight transport, the monthly time series from EDGAR per world region are used (Crippa et al., 2020). Since they do not capture the variations of individual countries, it is expected that the emissions for some periods are moderately underestimated, while in others they are
overestimated. As this only affects the monthly share, this has no impact on the annual emissions. There are indications that the extent of border control and customs affect the cross-country flow of passengers and goods. This is less relevant within the Schengen area; for other regions, it is expected that the emissions along the major border crossings are slightly overestimated.

Given the limited statistical data on passenger transport from a global perspective, the travel volumes for cars are derived from the motorisation rate based on GDPpC. This comes at the cost of not being able to study trip purposes and travel behaviour.
It is expected that both under- and overestimation of the activity occurs, albeit the magnitude cannot be assessed. It has a significant impact on the resulting inventories, given the bottom-up calculation approach. After the car traffic is calculated, the volumes for other means of transport are derived based on the average share per world region. Since car transport is the primary contributor to land-based emissions, the effect of over- and underestimations for the remaining subsectors on the resulting inventories is rather low. The new approach for spatial disaggregation, which is currently implemented only for
Germany, UK and the U.S. due to the availability of count station data, leads to a better understanding of the distribution of transport activity and emissions. For the rest of the world, an approach based on population density and the major road network is chosen, which causes underestimation along motorways in remote areas. At the same time, emissions in densely populated areas are overestimated, as alternative modes with lower emissions (such as public transport, cycling, and walking) have a higher share of usage. As a result, the emissions sum up to the target value for a country, but the spatial disaggregation
may be inaccurate by a factor of two. Only for countries with activity data for rail transport published, the associated emissions are calculated. As the mode share of this subsector is generally low, this causes a slight underestimation of the total railway emissions.

For about 50 countries the activity data for freight transport is known and easily accessible. For the remaining countries, a modelling approach based on GDPpC is chosen. However, for some countries structural differences in the relationship





between GDPpC and the freight transport performance are observed. Overall, this leads to both over- and underestimations
with a high impact on the resulting inventories since the activity data is the baseline for bottom-up calculations. For European
countries, only the transport performance of HFTs is reported, leaving the emissions of LCVs unconsidered in the resulting
inventories. The delimitation between vehicles used for passenger and freight transport is fuzzy when it comes to the usage of
pick-up trucks (especially in the U.S.) and similar vehicle classes (e. g., informal transport systems in African countries). This

is because of the lack of metadata for many countries with freight data reported. This may result in a slight shift in emissions
between the subsectors. In addition, an educated guess is required to combine the activity data of fuzzy subsectors with the
corresponding emission factors. The transport performance via railway is known only for a small number of countries. The
volume is inferred for remaining countries with a railway network but without detailed volumes published. As a tendency, this
causes overestimation with low impact given the small share of the mode.

### 3.1.2  Emission factors for exhaust emissions

The methodology for emission calculation relies on the vehicle stock data of 76 countries; for the remaining countries, assumptions are made regarding the vehicle stock and the vehicle age-related mileage. These assumptions may result in an inaccurate
representation of the circulation of older vehicles, specifically those manufactured to comply with Euro 2 emission standard or
earlier. As a result, the assumptions have a high impact on CO, HC and $NO_x$ emissions of gasoline vehicles in the reference

year. Furthermore, the proportion of diesel vehicles have a high impact on $NO_x$ and PM emissions. The resulting validity scores
account for the effect of non-available country-specific emission factors and stock data.

All emission factors are taken from HBEFA or EMFAC databases. Consequently, for countries not covered in these databases,
a number of factors that could potentially influence emissions are not captured due to the lack of information: the vehicle mass
and the related fuel consumption, the effects of driving shares on highways, rural and urban roads, the road conditions, and

age-based deterioration. For $SO_2$, in addition, the resulting validity score reflects the range and availability of data about fuel
quality and country-specific sulphur content.

As mentioned in Sec. 3.1.1, the emission calculation model does not model cold starts explicitly or their spatial distribution.
For instance, HBEFA emission factors were used at average temperatures. Since catalytic converters are not fully heated up and
the combustion process is incomplete in the cold start phase, CO, $CO_2$ and HC emissions are increased and thus underestimated

at lower temperatures. Thus, the effect of lower temperatures in countries due to for cold start is also accounted for in the
assignment of validity scores for each pollutant and drivetrain.

### 3.1.3  Emission factors for non-exhaust emissions

Determining emission factors for brake and tyre wear PM is a complex task as they are influenced by many parameters. For
tyre wear, apart from driving performance, vehicle weight is the most important factor, followed by the type of road surface.

Therefore, countries with a higher share of heavy vehicles and low road quality experience higher emission factors. Climatic
conditions are also important, as higher temperatures cause higher wear rates. Furthermore, in countries with cold climate,





studded tyres are often used (e. g., Scandinavia or Switzerland). The use of these tyres also increases the uncertainty of the PM emission factor.

For brake wear PM, vehicle weight and driving style are important factors which increase the emission rate. In general, brake wear emissions are expected to be higher in mountainous areas, as more frequent braking and higher brake pad temperatures also lead to higher wear rates.

There is no standardised measurement method for brake wear PN, and the available values from the literature cover a wide range from $10^9$ to $10^{13}$ particles/vehicle-km, mostly due to the large effect of brake temperature on PN emissions (Mathissen et al., 2023). Therefore, the emission factors used are subject to high uncertainty, while their effect on the resulting emissions is unclear. Less accurate emission factors are available for other vehicle types such as 2-wheelers and buses. Since in some world regions (especially some Asian countries) motorcycles account for a high percentage of road traffic, uncertainty on the overall emissions is increased.

Compared to brake wear PN emission factors, existing data about tyre wear PN is less reliable. This is partly because it is challenging to distinguish particles generated by the tyre from those produced by the pavement. While frictional brakes can be easily encapsulated, this is not possible for tyres. Existing literature suggests particle emissions range from $10^{10}$ to $10^{11}$ particles/vehicle-km (Löber et al., 2024; Dahl et al., 2006). Due to these high uncertainties, tyre wear PN is not included in ELK.

## 3.2 Shipping

Maritime shipping is inherently influenced by global economic fluctuations, which shape cargo flows worldwide. Nowadays, vessel movements can be tracked by analysing AIS data. AIS is limited by its design and provides only basic data related to the vessel identification, position and movement. Any access to additional information about cargo and vessel operations as well as internal voyage reports is highly restricted and cannot be used for emission estimation. The acquisition of AIS data depends not only on the technical capabilities of shore-based and satellite-based data reception networks, but also on legal restrictions. For example, the General Data Protection Regulation (GDPR) in the European Union aims to protect the privacy and personal data, which can impact AIS data availability. This may also apply to crew members working or living on board vessels. Moreover, access to AIS data may be blocked due to various aspects of national security like, for example, in case of the new Data Security Law (DSL) and Personal Information Protection Law (PIPL) in China.

Other details relevant for emission estimation such as the current engine use and fuel consumption are not included in the AIS messages. Therefore, accessing a vessel particulars dataset is essential. Such datasets provide shipyard-grade technical specifications for vessels and are compiled by maritime intelligence companies like Clarksons Research. The vessel particulars include detailed engine, fuel and propulsion parameters which are crucial for the emission calculation of a given vessel.

### 3.2.1 Activity data for international shipping

The vessels engaged in international voyage are selected based on the types of AIS messages broadcast by their AIS transponders. Special focus is placed on filtering out AIS message type 26 which is generally transmitted by AIS transponders on board





inland vessels and should not be used by vessels engaged in international voyage. To address problems related to interferences of AIS data which may lead to erroneous vessel positions, a land mask is applied globally to exclude unfeasible vessel movements which, as reported by faulty AIS transponders, are spotted on land. Despite careful examination of vessels belonging to the inland fleet, one cannot exclude a possibility of a data overlap: this is because many vessels operate on inland as well as international waterways, and their emissions might be included in both inland and international emission repositories.

The current engine load of a vessel is not reported within AIS data. Therefore it has to be estimated using the Admiralty formula in Eq. (2) which derives the current engine load by reckoning how close current speed and current draught of a vessel, obtained from AIS, are to her technical limits (Faber et al., 2020). The vessel particulars database does not contain technical records of all vessels visible in the AIS dataset used for emission calculation. Some of the records have missing data fields. To fill gaps in technical parameters, a statistical approach is applied by matching sister or near-sister vessels and leveraging

similarities in their technical details.

    Since the tracking of vessel movements, based on AIS, depends strongly on the quality of GPS positioning input used by AIS transponders onboard vessels as well as the completeness of data within AIS messages, it is necessary to avoid using parameters like, for example, AIS-derived speed over ground (SOG) which show a tendency to remain unknown or missing (Banyś et al., 2020). Therefore, speed over ground of a given vessel is calculated from two consecutive positions and timestamps reported by

AIS, disregarding the direct SOG readout provided internally by AIS itself. This approach ensures that latitude and longitude, which are the least frequently missing parameters within the dynamic AIS messages (Banyś et al., 2020), are used for the assessment of vessel movements.

    It might be worth mentioning that there are numerous AIS transponders on board vessels worldwide which are used without legal approval of the relevant maritime authority. The users of such AIS equipment configure MMSI numbers which end up in

the global AIS data archives as either a sequence of nine randomly chosen digits or, in the worst case, a copy of an existing legally approved MMSI of a legitimate vessel, be it purely accidental or on purpose. As described in Banyś et al. (2024), this situation may lead, to some extent, to an inaccurate assessment of vessel movements in the global scope and affect the estimation of emissions.

    It should also be emphasised that a stationary vessel may still use her main engine. There are navigational conditions for

stationary vessels, like for example anchoring in constrained tidal waters, which require the main engine to run most of the time to keep the vessel position under control. It is not possible to determine the status of the main engine operation with AIS data alone. Therefore, in numerous cases the emissions of stationary vessels, contrary to reason, may be underestimated.

    Another aspect of uncertainties related to vessel tracking includes the parameters of current draught and fuel consumption of the onboard machinery. The current draught is obtained from AIS message type 5, which is manually updated by the vessel's

crew. According to the standards of good seamanship, it should be kept up to date, but this is not always the case. An incorrect draught can lead to inaccurate engine load estimations, resulting in errors in emission calculations. Outdated draught, for example, can lead to an underestimation of emissions if a vessel is laden and reports a shallow draught via AIS, or it can be overestimated if the vessel proceeds under ballast and reports a deep draught.



Ideally, the estimation of maritime emissions would consider prevailing environmental conditions such as currents, wind, and wind-induced surface waves (sea state) measured by the significant wave height parameter. These factors can impact vessel movements in at least two ways. First, external conditions may affect the a vessel's current engine load. Second, vessels may intentionally deviate from planned routes to follow adverse weather avoidance procedures. The global distributions of climatological mean significant wave height indicate that major maritime routes, as seen in AIS data, cross the areas of prevailing rough sea conditions, especially in the North Atlantic (Timmermans et al., 2020). In cases of insufficient AIS coverage, especially in remote offshore areas, such route deviations cannot be fully reconstructed and it has to be assumed that vessels proceed along the shortest leg between two AIS-reported positions.

Global modelling of such ocean dynamics and weather patterns would certainly improve the quality of emission calculation but at the same time it would increase the computational complexity beyond the constrained project resources, too. Therefore, the environmental factors are generalised with respect to aggregated classes of vessel type and size. This simplification process might lead to an overall underestimation of emissions.

The tracking of global vessel movements is incomplete due to fragmented AIS data, particularly in remote areas, which is received from a constellation of low Earth orbit (LEO) satellites. Additionally, there are vessels worldwide that are not governed by the International Convention for the Safety of Life at Sea (SOLAS) with its carriage requirements for shipborne navigational systems and equipment (IMO, 2004). Such craft do not have to be equipped with AIS transponders and remain completely invisible for the emission calculation. Moreover, crews are allowed to disable AIS transponder on board their vessels, if their safety cannot be guaranteed, for example, in high-risk piracy areas where a vessel visibility via AIS would pose a hazard. Vessels may be engaged in illegal activities at sea, too. In such situation they often disable their AIS transponders, if they have any on board at all, or manipulate their AIS-reported positions to hide their movements (Paolo et al., 2024). There are attempts to track the movements of vessels which either disable their AIS transponders or do not carry one, for example, using satellite remote sensing (Milios et al., 2019). However, in the scope of project calculations such enhanced appraisal of vessel movements was not possible. In situations where a vessel disappears from the AIS traffic coverage for a longer period, her movements are estimated using a graph constructed from all available position reports stored in the AIS dataset used for emission calculation. Such approach allows for computation of the shortest nautically practicable route connecting two positions within globally navigable waters. Additional uncertainties of emission calculation may be introduced during a route reconstruction if vessels proceed through polar regions of the Northern Hemisphere. The sea ice concentration maps indicate that the changes in the ice coverage may be considered either as constraints of the present time or as new opportunities to overall navigational capabilities of vessels in the future (Rayner et al., 2003). However, there is no guarantee nor a possibility to verify that a vessel followed such a route, especially with fragmentary AIS coverage. Within the scope of the project's calculations it is generally assumed that vessels do not regularly choose the Northwest Passage between the Atlantic and Pacific oceans through the Arctic Ocean. Therefore, in cases of poor AIS coverage, emissions may be underestimated.

A similar tendency applies to vessels navigating within harbours, for example during berthing or unberthing. Vessel handling during manoeuvrers in constrained waters often involves a compulsory tug assistance with the vessel's main engine stopped. Such a situation may lead to an overestimation of emissions. Moreover, the main engine may temporarily run under full load



to adjust the speed or to improve the response to the helm. With almost no movement reported by the AIS transponder, a false
conclusion might be drawn that the vessel has no emissions from the main engine. Such situations, on the other hand, may lead
to an underestimation of emissions.

### 3.2.2   Emission factors for international shipping

The emission factors used for calculating emissions from the international fleet follow a generalized approach published by
the IMO (Smith et al., 2014; Faber et al., 2020). The emission factors depend, first of all, on the vessel typology because the
parameters are aggregated based on 18 major vessel types specified by the IMO algorithm (Faber et al., 2020). To apply the
emission factors properly, an accurate assignment of the IMO vessel types to the fleet types decoded from AIS data is necessary.
Unfortunately, there is no official typology of vessels. Various maritime classification societies have their own categorisation
schemes which are detailed up to a subcategory level of fleet types. This multi-level typology is then depicted in their own
vessel particulars databases. Mapping this multitude of vessel categories into the predefined set of vessel types used by the
IMO algorithm is challenging. This process introduces additional uncertainty into the emission calculations, as vessel type is
a crucial input parameter in the computation process.

The emission factors are defined either based on current energy demand of the main engine, which is related to current engine
load, or on amount of fuel consumed per unit of time by the main engine. Fuel data of a vessel is obtained from the Clarksons
Research vessel particulars dataset. Moreover, the IMO algorithm applies adjustments to emission factors for vessels located
within one of the following emission control areas (ECA), as of 2019: Baltic Sea, North Sea, North America, U.S. Caribbean
Sea. Before a vessel enters an ECA, she has to switch completely to an ECA-compliant type of fuel. This process follows
vessel-specific procedures through various stages of main engine maintenance and may take up to three days. The duration of
such ECA-related fuel transition events is considered during the emission computation. It should be noted that the emission
factors provided by the joint IMO researchers do not fully account for how the level of training and good seamanship of the
crews, responsible for maintenance of the onboard machinery, may influence the proper and environmentally friendly operation
of marine engines.

### 3.2.3   Activity data for domestic navigation

The inland shipping sector is excluded from the maritime shipping sector by the AIS message type 26, typical for domestically
operating vessels and applying land masks. Still double counts might be possible especially for coastal vessels, that are also
involved in transportation tasks on the waterways, leading to a slight overestimation in river mouths when both datasets are
combined.

The activity levels on the inland waterway systems are accessed through statistics of region-to-region transportation flows
in tkm (U.S., Europe) and the total transportation volume combined with AIS data (China). For the use of AIS data, a higher
uncertainty is introduced into the modelling region of China as it comes with the technology-specific restrictions as described
for the maritime sector and is additionally strongly affected with data sharing restrictions in China. This leads to blind spots,
where no AIS signals are received as well as overestimation and allocation of the transportation proportion of the blind spots





to areas with signals. Relying on the AIS data can lead to both under- and overestimation. Further waterway systems of the Amazon, Parana, Volga, Ganges, Nil, Congo, Mekong, and Niger are neglected in this dataset due to limited assessment and accessibility of transportation statistics. The overall proportion in transported goods on these waterways is below 4%

according to Zentralkommission für die Rheinschifffahrt (2012), therefore resulting in a slight tendency to underestimate the global activity level. Passenger transportation data, including distance and location details, is unavailable for the modelled regions and is therefore not included in the modelling approach. Since passenger transport on inland waterways represents only a small share, its exclusion results in only a slight underestimation of the overall activity level.

   The energy intensity for transportation is applied as an average value in MJ tkm$^{-1}$ and harmonized for all types of ships and

goods transported within a modelling region. This does not cover the variation of operating states and their specific emission behaviour, e. g., entering water locks, as well as navigating upstream or downstream the river. This can lead to over- as well as underestimation and shifting emissions on a modelled track. In contrast to the maritime sector, navigation on rivers is much less affected by storms as no severe wave forming is to be expected. Low water levels, such as those experienced in Europe in 2018, can reduce cargo capacity, necessitate additional trips to move the same amount of cargo, and consequently increase energy

intensity. However, for 2019 no low water levels have been reported. We distributed the transportation flows on the waterway network under the assumption that a ship would use the shortest route to fulfil a transportation task. In reality routes may differ due to specific trading patterns and stops along alternative paths, resulting in a possible underestimation. In summary, there is a tendency for the activity level assessment to underestimate the emission inventory for inland shipping.

### 3.2.4 Emission factors for domestic navigation

All modelled regions imposed emission regulations on the inland navigation sector for the emissions of NO$_x$, HC, CO and PM, with the strictest regulations and earliest implementation by the U.S. EPA Tiers, following by the European Non-road Machinery Regulations and the China Emission Stages. Emission regulations are imposed on ships at the time of their initial registration. Therefore the age structure of the fleet has a significant impact when developing fleet emission factors. We extracted the age structures of the database of Clarksons Research and developed an average fleet emission factor under the

assumption that the machinery complies with the relevant standards at the time of registry. Depending on the engine characteristics and maintenance by the crew, the emission limits might not be met in operation. Therefore the emission factors impose a tendency of underestimation.

### 3.3 Aviation

The accuracy of the modelled aviation emission inventory is influenced by a number of different aspects: both the availability

and quality of different input data as well as methodological uncertainties in connection with simplifications and assumptions. The quality of the input data varies regionally, e. g., due to spatially limited availability of routing data and taxiing time statistics. Also the quality of data on the worldwide aircraft fleet varies, e. g., due to the quality differences between the underlying aircraft performance models and the engine-specific availability of emission factors.





For these reasons, the quality assessment in aviation is carried out individually at the grid cell level and is based on the number of included trajectory segments that fulfil certain quality standards. In general, an uncertainty factor of 1 is is assigned to a grid cell when the relative frequency of included trajectory segments with reduced data quality exceeds a defined threshold, otherwise a value of zero is assigned. Multiple independent uncertainty factors of the included trajectory segments are weighted and finally summed up to the grid cell uncertainty score. A detailed discussion of the various uncertainties of the applied methodology is given in Weder et al. (2025b).

### 3.3.1 Activity data

The uncertainties of activity data depend on several criteria. First, the air traffic scenario for 2019 is based on various databases and modelling assumptions: for passenger air traffic, the demand and schedules, the airline network and fleet are considered to be of higher quality compared to the cargo subsector, where a higher degree of modelling is needed because of a lower availability of data. Furthermore, non-scheduled flights, helicopter flights, military air traffic, and general aviation such as private jets and business aviation are not included in the ELK emission inventory. However, the modelled subsectoral split of total $CO_2$ emissions is similar as derived by Graver et al. (2020). The most dominant uncertainty factor with regard to the gridded emission inventory is the quality of the applied aircraft performance model for trajectory modelling: these models affect the fuel flow rate, thrust, speed, as well as the climb and descend rate. BADA4 aircraft performance models cover the entire flight envelope and are more complex regarding flight physics than BADA3 models, which only cover the envelope at nominal conditions. Therefore, BADA4 models have an enhanced accuracy (Nuic et al., 2010; Poles et al., 2010). Due to the larger proportion of aircraft modelled using BADA3 models, the resulting emission inventories for regional and cargo subsectors have higher uncertainties. Besides the aircraft performance models, the applied Total Energy Model for trajectory simulation considering the aircraft as a point mass is another simplification. The simplification of the detailed trajectory to a RedEmP has only minor effects on the trajectory accuracy. Similarly, the impact on emissions due to deviations between the seat load factor from the flight plan and the discrete load factor from RedEmP are rather small (Weder et al., 2025b). For some flights with higher payloads or flight lengths close to the aircraft-specific and payload-dependent maximum range, the load factor has to be reduced or the initially assigned cruise altitude has to be changed to enable the trajectory simulation. This leads to a potential underestimation of the flight-specific emissions. If this occurs more frequently within a grid cell, the uncertainty score is increased.

Real flight path data is available for North America, Europe and inbound/outbound flights from the European airspace. Incorporating this data implicitly accounts for effects of air traffic management (ATM), enables a weighted multi-route pattern, and results in a more realistic routing. In contrast to this, missing detailed flight path data for the African and South American continent, necessitating the assumption of standardised great circle routes. In such cases, a horizontal route dispersion, detours and further routing inefficiencies are not considered and the vertical dispersion of flight levels is generic. Since idealised great circle routes lead to an underestimation of the emission quantities, an increased uncertainty score is attributed to the affected grid cells.





However, despite the fact that RedEmP are projected along a detailed horizontal flight path, in the vertical dimension the trajectories follow either a fuel-optimized or a constant cruise flight level over the entire cruise phase instead of depicting the exact ATM-affected altitude pattern of the route profile. This can lead to considerable deviations from real flight paths in individual cases. The permanent assumption of fuel-optimized cruise altitudes leads to a potential underestimation of emissions and constant cruise altitudes rather to an overestimation as the reality will probably be in between both modes due to ATM processes. Furthermore, RedEmP have only been calculated for the most common cruise altitudes of each aircraft type (29,000-41,000 ft for jets, 16,000-25,000 ft for turboprops); outliers of cruise flight levels in real route profiles cannot be taken into account. Therefore, for about 10% of the flight movements in the inventory, flight profiles are modelled with a cruise altitude deviating from the main cruise flight level as defined in the underlying flight path dataset – as a tendency, this leads to an underestimation of emission quantities in cases where a higher cruise altitude is underlain (and vice versa).

Another regional uncertainty factor is the availability of taxiing time statistics, which are available for most important airports in Europe and some busy airports on other continents. However, the median taxiing time does not represent daily conditions at the airport. For all other flights, standard taxi-in and taxi-out time from LTO cycle (ICAO, 2008) are used which do not consider individual airport size, capacity and runway layout. In contrast to most other emission inventories, ELK inventories consider emissions due to taxiing and use available data for local taxiing times to reduce uncertainties.

Air distance is corrected by wind effects, that are statistically modelled based on ERA-5 reanalysis data (Hersbach et al., 2023). For small aircraft with low cruise speeds, wind impact could not be derived. This results in a slight underestimation of the flown distance on average (Linke, 2016). Uncertainties of the wind speeds and their spatial distribution depend on the accuracy of the gridded 3D wind fields used. While ERA-5 data have the same horizontal resolution as the ELK aviation emission inventory of 0.25° × 0.25°, the vertical resolution of layers declines with increasing altitude due to the pressure levelling. Thus, the data resolution of the wind fields on common cruise flight levels is rather coarse. We follow the methodology of Swaid et al. (2024), who propose to statistically capture the wind effects on air distance for an annual period and on a fuel-optimized cruise flight level. Note that the annual mean wind effect does not represent the actual daily wind situation along the detailed 3D route profile. Further inaccuracies are introduced by the usage of the ISA for trajectory simulations instead of real atmospheric background data.

### 3.3.2 Emission factors

Uncertainties in the derivation of emission factors are mainly driven by the accuracy of the underlying input emission data and the robustness of the applied extrapolation and interpolation method. The uncertainty scores vary per emission species. The constant emission factors for $CO_2$ and $H_2O$ are considered to be reliable and robust. $SO_2$ emissions are directly affected by fuel sulphur content values and vary according to the used fuel sample both regionally and over time. To reduce uncertainties, in ELK we use available fuel sulphur content values quantified on continental level (CRC, 2012), which does not represent local fuel properties but should be more accurate compared to the assumption of a global average value. For state-of-the-art jets and piston engines, input emission factors for $NO_x$, HC/NMVOC, CO and particles (BC, nvPM) originate from test bench measurements for four engine thrust settings as defined in LTO cycle (ICAO, 2008). The emission factor measurements contain





small inaccuracies for jet engines depending on the used fuel specification (EASA, 2023). The uncertainties of the measured piston emission factors are higher (FOCA, 2007). For modelling turboprops, only constant values from overall fleet averages are available, which do not predominantly represent turboprop engines. This leads to increased inaccuracies and higher uncertainty scores, and mainly affects the results of regional and cargo subsectors where 51% and 35% of flight movements, respectively,
are operated by turboprop aircraft. The extrapolation of emissions from on-ground measurements for the LTO cycle to in-flight conditions is done using fuel flow correlations, methods whose inaccuracies are discussed in the literature (e. g., Schulte et al., 1997). For the emission factors of BC, nvPM mass and number, uncertainties occur due to the application of constant emission factors from fleet averages for all engines, where no measurements are available; this affects in particular older turbojets, some piston aircraft and all turboprops. For the affected grid cells, higher uncertainty scores are assigned.

## 3.4 Energy for transport

For the representation of energy-related indirect transport emissions from refining, a large share of the data rely on reported refinery-level or country-level emission data. Due to the complexity of estimating refinery emissions, reports are considered the most reliable data source for the emission inventories. Reported data is mostly available for countries in Europe, North America, Oceania and Russia which make up the majority of global $CO_2$ emissions (61%). For the remaining countries, emissions are
1080 calculated using refinery fuel consumption data and emission factors. Therefore, depending on the method for determining the emissions, the data carry different uncertainty factors which will be explained in the following.

### 3.4.1 Activity data

Uncertainties regarding refinery activity data arise from the lack of available information on the properties of individual refineries. Activities and product quantities of a refinery depend on various aspects, e. g., the crude oil used, the refinery complexity
or the preference of refinery operators. This aspect influences the uncertainty of the emission inventories in several ways. To determine the share of transport-induced refinery emissions from total refinery emissions, we use product shares based on country-level refinery product quantities from the IEA energy balances to distribute refinery emissions to single refinery products, although to a certain extent they might differ from one refinery to another within a country. Therefore, this aspects leads to moderate uncertainty in the spatial distribution of emissions in all countries. Furthermore, when emissions from refineries
are determined on country-level, we use the oil processing capacity of single refineries as a proxy despite the fact that other refinery-specific characteristics are also relevant, although to a less noticeable extent. For Europe, the U.S. and Canada we used reported total refinery-level emission data as a basis for our estimations. To approximate the share caused by combustion process activities from total refinery emissions, we applied species-specific factors from Sun et al. (2019). These shares represent an average of U.S.-American refineries and do not take into account individual emission characteristics of a refinery, e. g.,
due to the refinery setup, varying crude oils used or country-specific emission standards. We assume that the shares of combustion emissions in total refinery emissions of each species are well represented in its general tendency, thereby resulting in a moderate level of uncertainty. Since these shares apply to U.S. refineries, the uncertainty for refineries in the EU and Canada are expected to be slightly higher. To our best knowledge, no monthly profiles for refinery processes are available. Therefore,





we used the consumption profiles of land transport, the most consumption-intensive transport subsector, to allocate annual emissions to individual months. Since transport fuel consumption is usually relatively evenly distributed, the uncertainty can be considered low. Annual emission estimations are not affected by this uncertainty.

### 3.4.2 Emission factors

For a part of the globe, we used default emission factors for refinery combustion processes to determine refinery emissions. These emission factors do not represent the emission standards and limits given in individual countries and could both over- or underestimate real emissions. This applies to developing countries in South and Central America, Africa and Asia, representing about 34% of the global $CO_2$ emissions from refineries in the inventory. The uncertainty can be considered high in these countries for air pollutants and lower for $CO_2$, since $CO_2$ is usually released into the atmosphere uncontrolled. To estimate global emissions for the selected set of species, the emission quantities of some species are derived from those of another species, as they may not be equally available in all regions ("gap filling approach"). This is done using country-specific emission factor ratios between a known and unknown species. In Europe, especially CO and PM emissions are gap-filled in many cases from other species. Due to the individuality of emission factors of each refinery, this approximation leads to moderate uncertainties especially when species are derived from non-related other species (e. g., PM from $CO_2$) instead of related species (e. g., $PM_{2.5}$ from $PM_{10}$). Similarly, missing emission species in reported country-level data is gap-filled by approximating them using country-specific emission ratios. When reported emission data from refineries or countries are used, sometimes not all particulate matter categories targeted in the emission inventories are included in the emission datasets (e. g., $PM_{2.5}$ and BC in EU refineries and BC in Canadian refineries). To fill these gaps and provide an estimate of BC and $PM_{2.5}$ emissions in these cases, we approximated them from emissions of other particulate matter categories, based on the share of BC emission factors relative to $PM_{2.5}$ or $PM_{10}$. For EU refineries, also $PM_{2.5}$ emissions are derived from the reported $PM_{10}$ emission totals. The emission quantities of the derived species carry a medium level of uncertainty since related species (other particulate matter categories) are used for the approximation. For Canadian and U.S.-refineries, we used the reported VOC emission data to represent NMVOC emissions in the inventory. Since VOC emissions include methane emissions which are excluded in NMVOC emissions, we expect emissions in our inventory to be slightly overestimated in these regions. The uncertainty is relatively low, since methane is usually burned in the combustion process and the share of methane in VOC emissions is therefore low.

## 4 Results and comparison with other inventories

In this section, the ELK inventory is presented for the different sectors and compared with other well-established global inventories. For this comparison we consider CEDS (Community Emissions Data System; Hoesly et al., 2018) in the v2021-04-21 release (O'Rourke et al., 2021) and, for aviation, in the updated v2023-04-18 release (Prime et al., 2023), which includes the correction by Thor et al. (2023); CAMS-GLOB (Soulie et al., 2024), consisting of CAMS-GLOB-ANT_v5.3 (for land transport), CAMS-GLOB-SHIP_v3.2 (for shipping) and CAMS-GLOB-AIR_v2.1 (for aviation); and EDGAR8 (Crippa et al.,



2024) consisting of v8.0 (for GHGs) and v8.1 (for SLCFs). Note that these inventories are not completely independent of each other, as for example CAMS is partly based on an earlier version of EDGAR and includes data from the CEDS inventory as well. Nevertheless, they provide different emission totals, making it valuable to compare the ELK results with all of them. Additional datasets are considered for specific sectors and variables and are outlined in the respective subsections below. The comparison is generally conducted based on aggregated emissions, both at the global and the regional level. For the latter, we consider the IPCC AR6 scientific regions depicted in Fig. S2.

**Table 5.** Globally aggregated emissions of the four sectors considered in the ELK inventory. $CO_2$ emissions do not include the share of biofuels, which is accounted for in $CO_2$-total (for land transport only). The species nvPMm and nvPMn indicate non-volatile particle matter mass and number, respectively. Units are Tg(species), Tg($NO_2$) for $NO_x$ and particles for PN and nvPMn.

| Species | Land transport | Shipping | Aviation | Energy |
|---|---|---|---|---|
| BC | 0.46 | 0.29 | 0.009 | 0.002 |
| $CH_4$ | 0.86 | 0.02 | – | – |
| CO | 46.82 | 1.59 | 0.84 | 0.16 |
| $CO_2$ | 6638.60 | 822.17 | 851.04 | 533.03 |
| $CO_2$-total | 7382.04 | – | – | – |
| $H_2O$ | – | – | 333.25 | – |
| HC | 4.97 | – | – | – |
| $N_2O$ | 0.28 | 0.05 | – | – |
| $NH_3$ | 0.69 | – | – | – |
| NMVOC | 8.50 | 0.75 | 0.08 | 0.02 |
| $NO_2$ | 3.14 | – | – | – |
| $NO_x$ | 25.51 | 13.86 | 3.74 | 0.50 |
| nvPMm | – | – | 0.01 | – |
| nvPMn | – | – | $1.18 \times 10^{26}$ | – |
| OC | 0.32 | – | 0.005 | – |
| $PM_{10}$ | 0.94 | 1.48 | – | 0.03 |
| $PM_{10}$ brake wear | 0.22 | – | – | – |
| $PM_{10}$ tyre wear | 0.24 | – | – | – |
| $PM_{2.5}$ | 0.75 | – | – | 0.02 |
| $PM_{2.5}$ brake wear | 0.08 | – | – | – |
| $PM_{2.5}$ tyre wear | 0.16 | – | – | – |
| PN | $4.05 \times 10^{26}$ | – | – | – |
| PN brake wear | $8.30 \times 10^{24}$ | – | – | – |
| $SO_2$ | 1.01 | 10.62 | 0.20 | 0.53 |



## 4.1 Land transport

The global $CO_2$ emissions of land transport in the ELK inventory are shown in Fig. 6 for the sector as a whole and for each subsector. The emissions are particularly high in the Northern Hemisphere, especially over Europe, the eastern U.S., India and eastern China. Emission peaks are clearly visible in correspondence with the major population centres and along major roads and highways, both in the Northern and, to a lower extent, in the Southern Hemisphere. The global $CO_2$ emissions of land transport amount to 6639 Tg in the reference year 2019 (Table 5) and 7382 Tg when considering the contribution of biofuels ($CO_2$-total). The $CO_2$ land transport emissions are dominated by cars, sharing 45% of the total with a geographical pattern closely matching that of the total land transport emissions (Fig. 6b), and by heavy freight trucks with 33% of the total (see Table S4, for details). Compared to the emission distribution of cars, which are quite spread over large areas, LCVs and HFTs emissions are more concentrated along major roads and highways (Fig. 6c,d). The other subsectors are responsible for less than 10% of the total emissions, although they are important in specific regions, such as for instance 2-wheelers in India or bus transport in North America. Bus transport also shows high emissions spots in major urban areas. The rail transport contribution to land transport emissions is minor (about 1% for $CO_2$) and is mostly due to freight. Emission maps of other species are shown in the Supplement (Figs. S3-S22), with the totals and the shares of each subsector given in Table S4. The emission shares of the SLCFs show interesting differences with respect to $CO_2$: the largest share of BC, $NO_x$ and PN emissions, for example, is by heavy-freight trucks, due the predominance of diesel engines among these vehicles, while 2-wheelers dominate the emissions of CO, $CH_4$, HC and OC, due the widespread usage of two-stroke engines in this category.

The comparison of the ELK land transport emissions with other global inventories is summarised in the heat map in Fig. 7, showing the relative difference in the globally aggregated emissions between ELK and each of the three inventories considered for the comparison (CEDS, CAMS-GLOB and EDGAR8). This analysis reveals the excellent agreement of ELK with these inventories for $CO_2$, with deviations within a few percent. Breaking down this comparison in the different world regions (Fig. S26) reveals, however, that this agreement actually results from a compensation between higher ELK emissions in some regions (mostly South and East Asia) and lower ELK emissions in other regions (mostly Western and Central North America). These regional discrepancies are consistent across the three inventories used for the comparison, revealing a rather peculiar geographical distribution of the land transport $CO_2$ emissions in the ELK inventory.

The deviations between emission estimates in the inventories are larger for non-$CO_2$ species, not only with respect to ELK but also across the other inventories, as a result of the higher uncertainties in the emission factors of these species. The differences can be partly explained by the use of a bottom-up approach in the ELK inventory, which incorporates emission factors at the subsector level. The largest differences are observed with the CEDS inventory, showing deviations greater than 50% for BC, CO, $N_2O$, $NH_3$, NMVOC and $SO_2$. CAMS-GLOB and EDGAR8 are more aligned with the ELK aggregated emissions, although the latter shows a very large discrepancy for $SO_2$: the regional analysis (Fig. S34) indicates that this deviation can be attributed to a factor-5 difference between ELK and EDGAR8 in Southeast Asia, which accounted for 22% of the total $SO_2$ land transport emissions in ELK.



**Figure 6.** $CO_2$ emissions from land transport (a) and related subsectors (b-h), not considering the share of biofuels (see Fig. S6 for the total $CO_2$ emissions of land transport including biofuels).



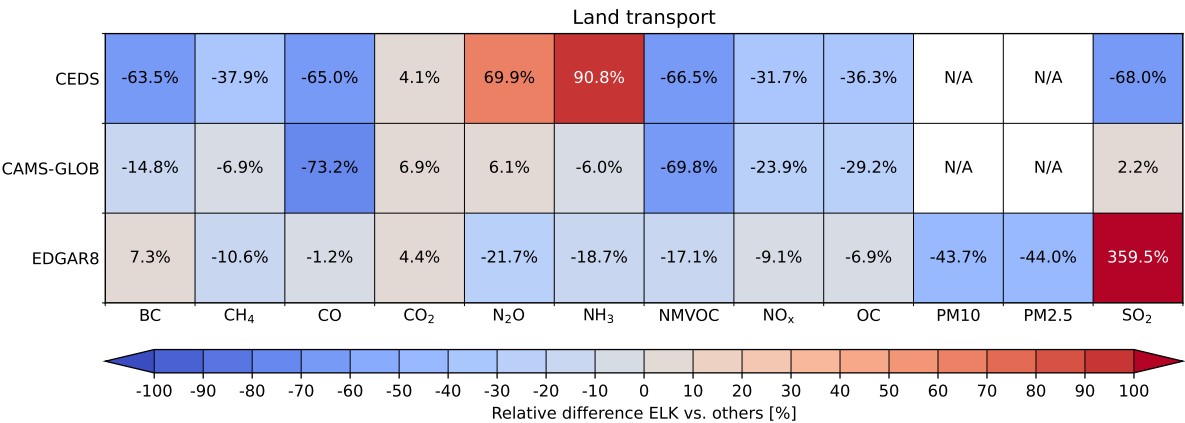

**Figure 7.** Relative difference of globally aggregated land transport emissions between ELK and the other global inventories: positive (negative) values indicate higher (lower) emissions in ELK. Note that the CEDS inventory includes inland navigation as part of the land transport sector.

Figure 8 shows the uncertainty scores for $CO_2$ land transport emissions assessed with the method outlined in Sect. 3, which for this sector is based on an analysis at the country level. Based on this analysis, the lowest uncertainty (score 1) is assigned to most of the developed countries, with a few exceptions, while other countries are assigned an uncertainty score of 2 or, in a few cases, an uncertainty score of 3 due to very limited knowledge of the corresponding activity data. The uncertainty scores for the other species mostly follow the same geographical pattern (see Fig. S35), although they are generally higher, in particular for BC, HC, NMHC and PN. Significantly higher uncertainty scores are assigned to the non-exhaust mass emissions from tyre and break wear, with a best uncertainty score of 2 assigned only to a few European countries and higher uncertainty scores up to 4 assigned to most of the other countries. These are even higher for non-exhaust number emissions, where only Europe and U.S. are assigned a best uncertainty score of 3.

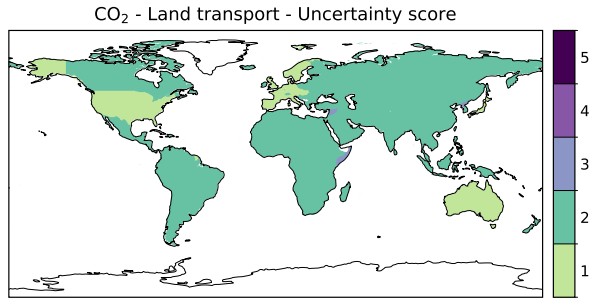

**Figure 8.** Uncertainty scores for the $CO_2$ land transport emissions, ranging from 1 (very low uncertainties) to 5 (very high uncertainties).




## 4.2 Shipping

In the ELK inventory shipping emissions are resolved into international (ocean-going) shipping and domestic navigation, although the latter is only modelled for Europe, U.S. and China (Fig. 9). The distribution of international shipping emissions follows the major shipping routes in the Northern Pacific, Northern and Southern Atlantic, with significant activity in the Indian Ocean and off the coasts of South and East Asia. A striking feature are the high emissions along the coastlines, especially in the North and Baltic Sea, in the Mediterranean and along China and South-East Asian countries. Two world-record-breaking traffic

zones are located in Europe, which is reflected in the emissions: the English Channel, the world's busiest natural waterway, and the Kiel Canal, the world's busiest artificial waterway. As for the land transport sector, there is a significant difference between Northern and Southern Hemisphere. The global $CO_2$ emissions from shipping amount to 822 Tg in the reference year 2019, with an overwhelming dominance (93%) of international shipping. This share remains above ∼85% for all species (Table S5), with the notable exception of CO, for which domestic navigation contributes 65%. This can be explained with the fact that

the inland vessel fleet contains many old ships and engines without emission standards. Furthermore, smaller engines are often operated with higher fuel-to-air mixtures, which can lead to incomplete combustion and increased emissions of CO. The geographical distribution of the emissions is similar for all species, although for $SO_2$ the ELK inventory considers the reduced fuel sulphur content in the Sulphur Emissions Control Areas (SECA), implemented from 2010 following the regulations of the IMO on shipping fuels to improve air quality in coastal regions. This is an important feature of the inventory, particularly in

the context of the recent debate on the impact of the IMO regulations on climate (Jordan and Henry, 2024; Gettelman et al., 2024; Quaglia and Visioni, 2024).

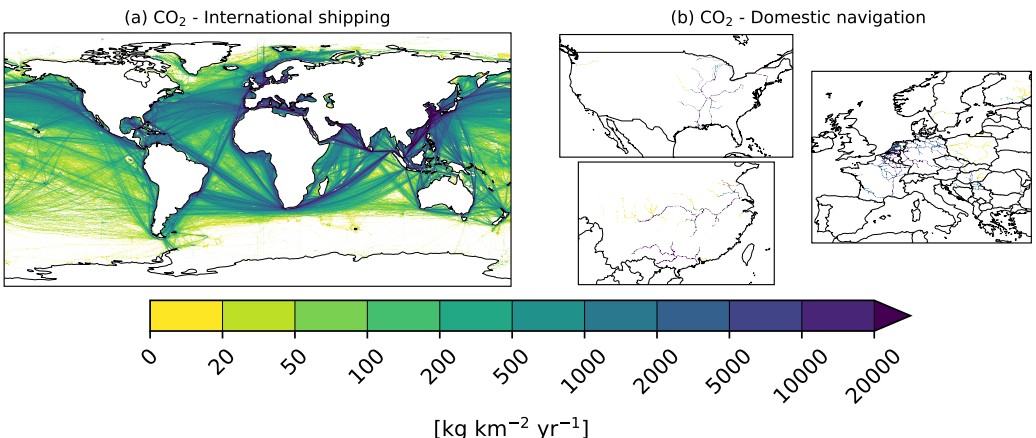

**Figure 9.** $CO_2$ emissions from shipping: (a) international shipping, (b) inland navigation.

The comparison with the other inventories shows a generally good agreement for the three species with the highest emissions, namely $CO_2$, $NO_x$ and $SO_2$ (Fig. 10). ELK global $CO_2$ emissions are 5-9% lower than in the other inventories. The agreement is also remarkably good for $SO_2$, with differences of only a few percent in comparison to CEDS and CAMS-GLOB, while ELK



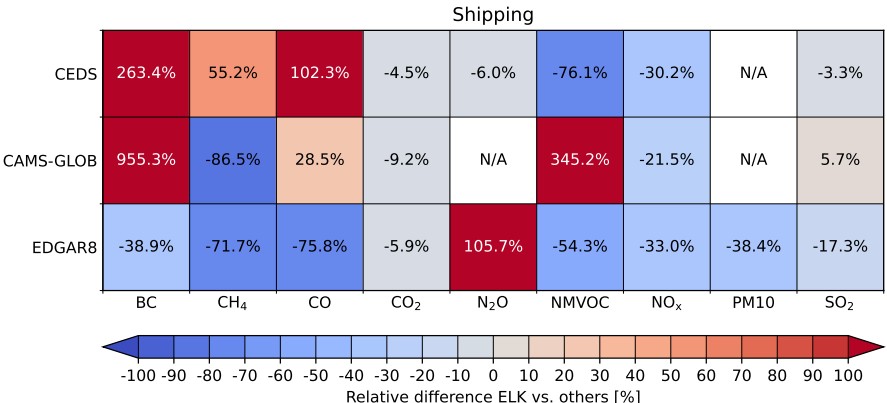

**Figure 10.** As in Fig. 7, but for shipping emissions. Note that the CEDS and EDGAR8 inventories do not include inland navigation in the shipping sector.

has 17% lower emissions than EDGAR8, since this inventory does not account for the IMO regulations in the SECAs, which obviously have an impact on the total $SO_2$ emissions. The $NO_x$ emissions in ELK are about 20-30% lower than in all other inventories, which could be due to incompleteness in the technical data: as the $NO_x$ emission factor strongly depends on the ship construction date and on the engine's RPM (Faber et al., 2020), incompleteness of this data could have a large impact on the resulting emissions. The deviations for the other species are considerably larger, especially for BC and NMVOC. However, there is considerable disagreement across the other inventories as well, indicating a high level of uncertainty in the emission factors for these species in the shipping sector. The emissions of these species are nevertheless low compared to three key species discussed above (see Table 5), which are also the most relevant in terms of climate impact (Righi et al., 2023; Mertens et al., 2024). The analysis of the deviations at the regional level shows a predominantly uniform distribution across the most important regions in CAMS-GLOB and EDGAR8 (Fig. S47), while large regional differences are found in the comparison with CEDS: in this case the ELK inventory shows much larger emissions in the Mediterranean, around the Arabian-Peninsula, and in East and Southeast Asia, and lower emissions in the North Pacific and North Atlantic ocean. This is somewhat surprising, since all inventories are supposed to be based on similar activity data from AIS and, at least for $CO_2$, they should use similar emission factors. One reason could be the allocation of the activity data to the different types of vessels and the corresponding assumptions drawn on their technical parameters, impacting fuel consumption and, in turn, $CO_2$ emissions. This could also be a reason for the higher global $CO_2$ emissions in ELK compared with the other inventories. The distribution of the regional deviations is similar for $NO_x$ and $SO_2$ (Figs. S50 and S52, respectively), again pointing to the activity data, in particular the assumptions about the ships technical features in each ship category, as the possible reason for these deviations.

The uncertainty in the activity data for the ELK inventory can be inferred from the uncertainty score for the $CO_2$ international shipping emissions shown in Fig. 11a. In our uncertainty assessment, we assigned score 1 for the $CO_2$ emission factor worldwide, such that the overall score for this species is only determined by the uncertainty in the activity data. For the latter, we consider the fact that our emission model accounts neither for wave heights, which are particularly significant at the



mid-latitudes, located between the Tropic of Cancer and the Arctic Circle as well as between the Tropic of Capricorn and the Antarctic Circle (Timmermans et al., 2020). We also assume that AIS data has a lower vessel coverage along the coastlines of certain regions due to unreported activities (Paolo et al., 2024). Combining all these factors results in an uncertainty score 2

over most of the globe, with a score 1 limited only to a few regions around Europe, U.S., South America, Africa and Australia. The scores are similar for $SO_2$ and considerably higher for other species (Fig. S53), consistent with the comparison with the other inventories discussed above.

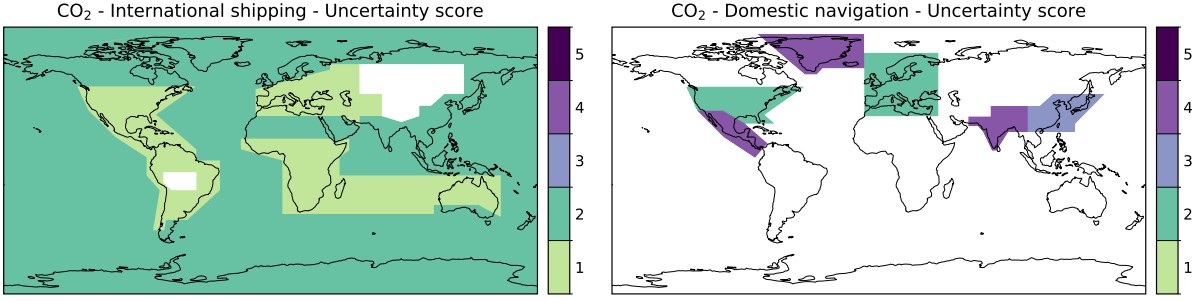

**Figure 11.** Uncertainty scores for the $CO_2$ shipping emissions, ranging from 1 (very low uncertainties) to 5 (very high uncertainties).

For $CO_2$ emissions from domestic navigation (Fig. 11b), we assess a score 2 for the U.S. and Europe and a score 3 to 4 for China. The uncertainty of this subsector primarily arises from the assessment of activity data based on the input transportation

statistics, which vary in spatial and temporal resolution. For China, we applied a combination of transportation statistics and AIS data, which is affected by the limited AIS availability and blind spots where no signals are received. This leads to a higher uncertainty score compared to Europe and the U.S.. The energy intensities of river transportation in the ELK inventory are set to a characteristic value within each modelling region, hence not capturing local phenomena such as increased energy intensity due to varying ship operating speeds, higher river flow velocities or location-specific fleet characteristics. The emission factors for

$NO_x$, CO, HC and PM are calculated based on the fleet age structure and region specific emission limits, under the assumption that these limits are met. In reality ships might not comply to the regulations, leading to a tendency of underestimation in our inventory. Other emission factors are aligned with those of the international shipping sector and are therefore subject to the same uncertainties.

### 4.3 Aviation

The ELK inventory for aviation considers four subsectors or aircraft types, namely wide-body, single-aisle, regional and cargo aircraft (Fig. 12). Wide-body aircraft dominate the long-distance intercontinental traffic between North America, Europe, and East Asia, predominantly in the Northern Hemisphere. These three regional hotspots are also evident in the single-aisle route patterns, which are mostly concentrated in these regions almost exclusively above the continents, with a few routes in the North Atlantic and in the Pacific connecting Hawaii with North America. Regional flights are frequent in the U.S. and Europe and

limited to a few routes in the other regions of the world, thus marking a difference between developed and developing countries

as well as in the population density. Cargo flights also have a long-distance pattern, with routes mainly connecting North America, Europe and China. The vertical stripes over the North Atlantic, especially visible in Fig. 12a,b, are a consequence of the reduced flight radar coverage over the North Atlantic and thus a reduced density of waypoints (at 10° longitude intervals) along the North Atlantic tracks. The various transatlantic routes are therefore mainly concentrated on the available waypoints

and are approximated by great circle segments between these points.

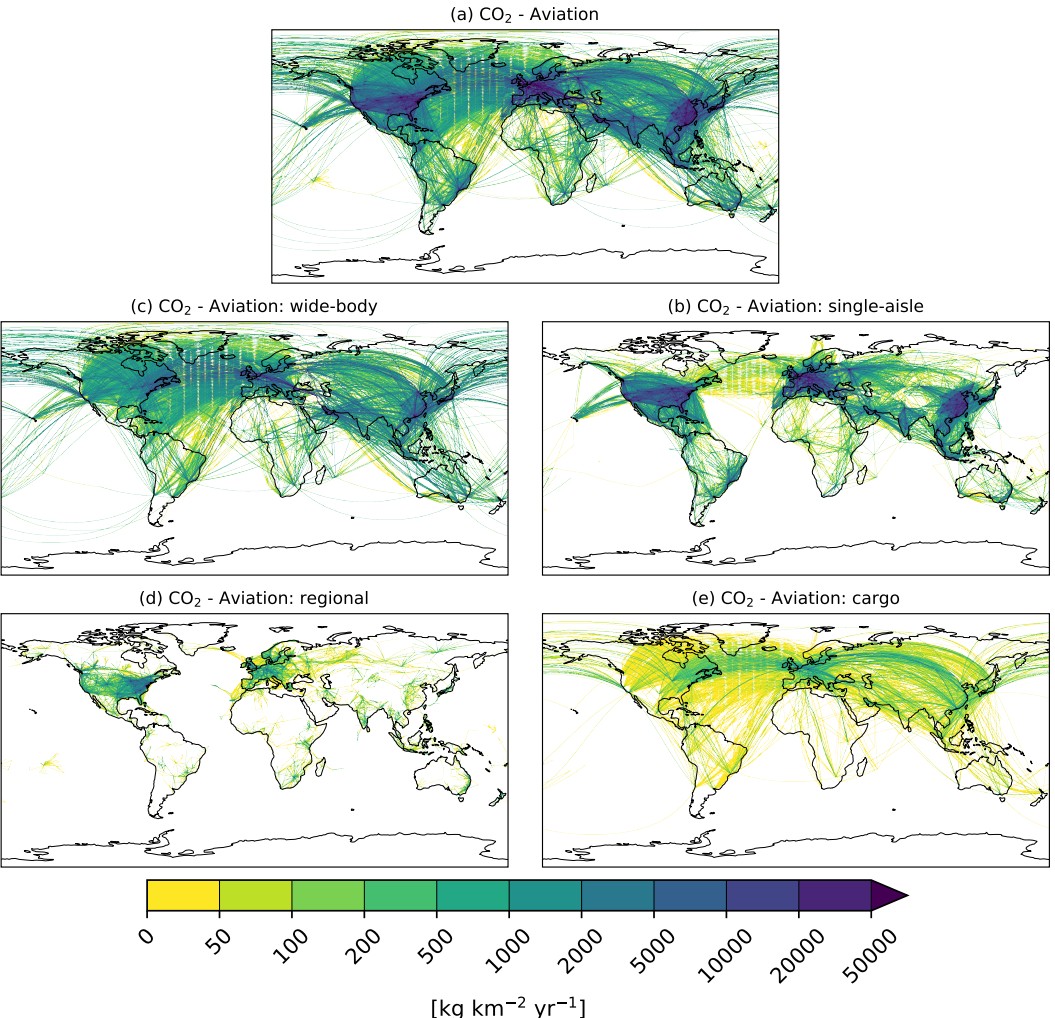

**Figure 12.** Vertically integrated $CO_2$ emissions from aviation (a) and related subsectors (b-e).

The zonal maps (Fig. 13) show a peak in the $CO_2$ emissions at the typical cruise altitude of 10-12 km at northern mid-latitudes for all subsectors, with the exception of regional aircraft, which tend to fly lower, with a peak at around 6 km. This is due to the often very short flight distances of regional aircraft, which prevent them from reaching higher cruise altitudes, as well as the fact that a significant number of regional routes are operated by turboprop and piston aircraft, which have lower cruising

altitudes for technical reasons. Wide-body aircraft tend to follow higher-latitude routes (50-60°N) compared to single-aisle aircraft (40°N). Due to limited range, transatlantic flights of single-aisle aircraft only commute between U.S. East Coast and Central Europe along the North Atlantic Track System, whereas wide-body flights also occur between the Middle East and the west coast of North American, resulting in more northward operations.

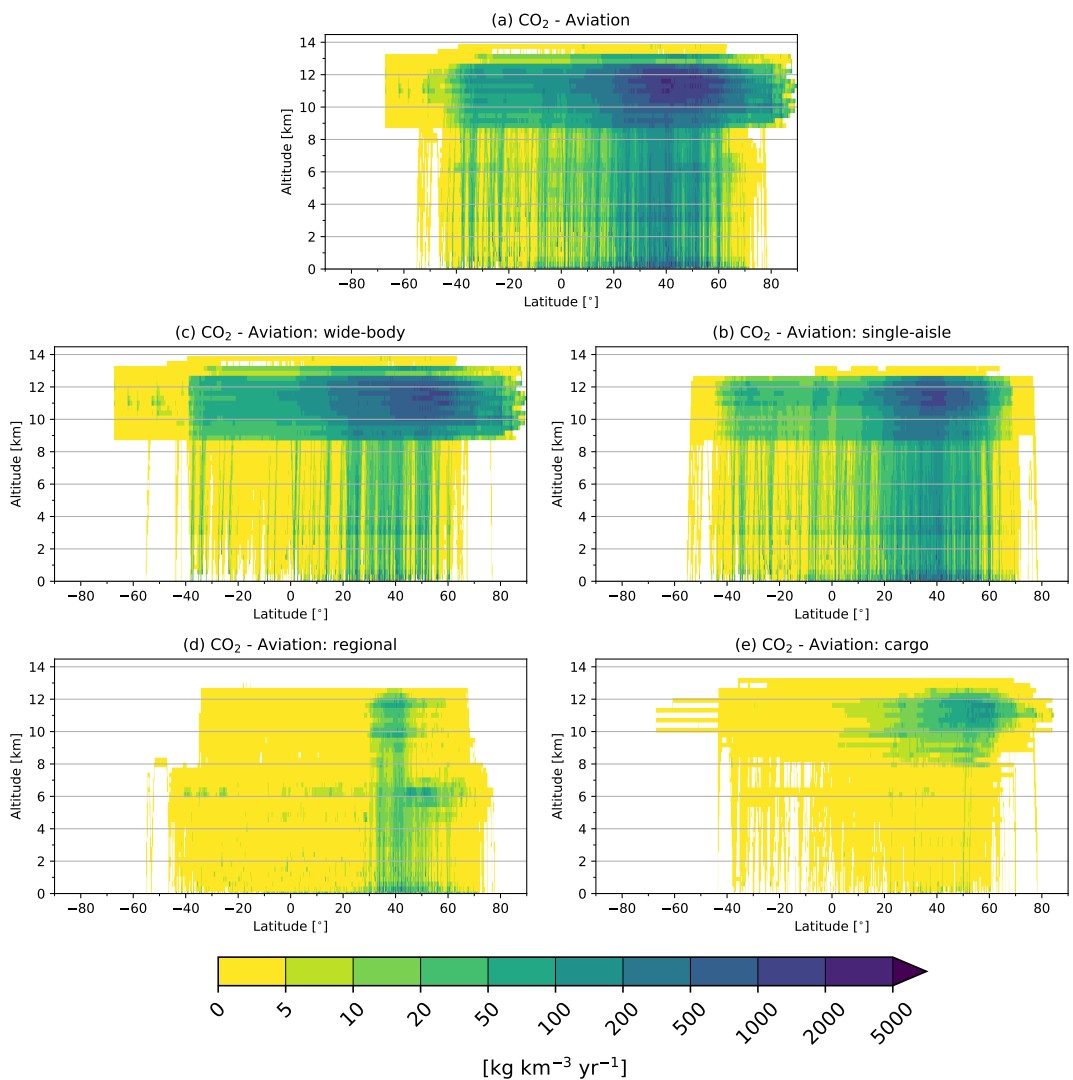

**Figure 13.** Zonally averaged $CO_2$ emissions from aviation (a) and related subsectors (b-e).

The global $CO_2$ emissions from aviation amount to 851 Tg (Table 5), almost completely due to wide-body (46%) and
single-aisle aircraft (48%), see Table S6. Interestingly, only 3.3 million wide-body aircraft flights (8.5% of the total) contribute to a similar share of emissions as the 27.8 million single-aisle aircraft flights. This is due to the higher distance-specific fuel





consumption of wide-body aircraft in consequence of their higher weights that require more powerful engines. The large shares of emission from wide-body and single-aisle aircraft are also evident for the other species, although for CO and NMVOC the share of single-aisle and regional aircraft is significantly larger, since about one third of NMVOC and CO emissions are caused

by taxiing activities on the ground when the engines are running in idle mode, as these activities have relatively higher impact for flights with shorter cruise phase. The ELK emission inventory for aviation also includes hourly data representative of the average diurnal variation of the emissions over the reference year 2019. This kind of data is useful for modelling of contrail cirrus and their impact on climate (e. g. Burkhardt and Kärcher, 2011; Bickel et al., 2025), especially concerning the emissions of $H_2O$ as well as flight distance and propulsion efficiency. The diurnal variation for $H_2O$ is shown in Fig. 14 for the four

aviation subsectors. It shows a main peak around 14 UTC, mainly driven by the daily maximum of single-aisle emissions. At this time of the day, domestic traffic is intense in the three busy regions of Europe, North America and Asia. On the U.S. East Coast, the morning increase in flight movements begins around this time. A second peak can be seen at 1 UTC and is due to the diurnal maximum of wide-body aircraft: this is the time of morning air traffic peak in East Asia and many North Atlantic crossings also take place at this time. However, the daily amplitude of wide-body $H_2O$ emissions is lower compared to the other

subsectors, because intercontinental and long-range flights take place at any time of the day. In contrast, the daily minimum of the overall aviation sector and of the single-aisle subsector occurs at 22 UTC, when nocturnal airport closures in Central European airports markedly reduce the European domestic air traffic. The contribution from regional traffic is high between 13 and 21 UTC, i.e. the morning time in the subsector's dominant area of North America. Cargo traffic is evenly distributed throughout the day, with slight peaks at night time in East Asia, morning in North America and evening in Europe.

The comparison with other inventories (Fig. 15) shows that aviation has the best agreement with the other inventories among the three transport sectors, with deviations within 50% and mostly within 20% for all species. The ELK inventory calculates lower aviation emissions than the other inventories in most cases, while it has generally larger emissions than the predecessor inventory TraK, developed in a previous DLR project (Weder et al., 2025b). This is due to the consideration of substantial parts of global cargo traffic volume, on the one hand, and due to various methodological improvements in ELK, on the other hand.

The latter includes the statistical dispersion of the flight-specific emissions along different flight altitudes, including those at which the aircraft does not operate optimally in terms of fuel, the prevailing routing along real flight paths instead of great circle routes, and the wind-induced scaling of the air distance. This leads to longer flight distances and higher fuel consumption and hence to increased emissions in ELK compared to TraK. However, both the ELK and the TraK inventories are developed using a bottom-up approach, where the transport demand and the quantity structures are modelled individually and in great detail at

the airport-pair level, so that, for example, a higher volume of air traffic is assumed for Asia due to a higher resolution of the traffic structure (which could explain the large differences of ELK against the other inventories in this region, see Fig. S80). The other inventories, on the contrary, mainly follow top-down approaches at a lower level of detail. Another reason for the higher ELK emissions in Asia could be related to the recent increase in activity in this region, which is not fully captured by the other inventories, since they possibly rely on less up-to-date activity data. The largest deviations are found for $SO_2$, with ELK

showing lower emissions than all others. The emissions of this species mainly depend on the assumed fuel sulphur content. For the ELK inventory, regionally varying sulphur concentrations in the fuel varying from 275 ppm to 635 ppm taken from

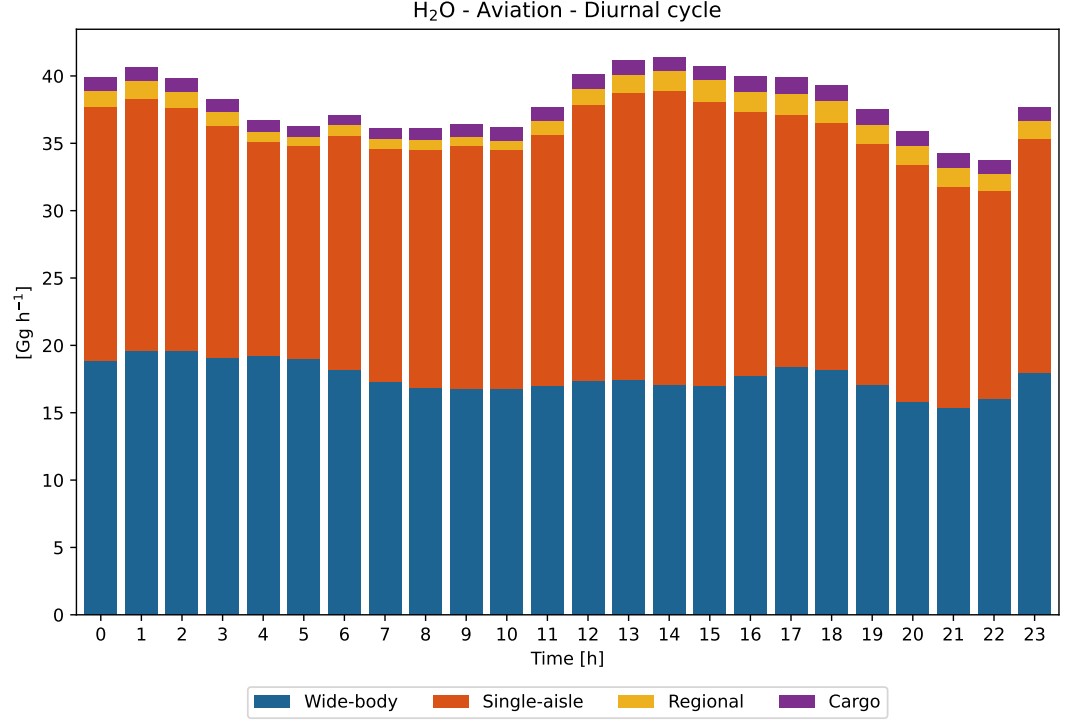

**Figure 14.** Aviation: Diurnal cycle of the annual total aviation $H_2O$ emissions by hour (UTC) and subsector for an average day.

CRC (2012) are used to calculate the $SO_2$ emission factor depending on the geographical location of the departure airport, while other inventories often assume a globally averaged constant fuel sulphur content (e. g. Wilkerson et al., 2010; Simone et al., 2013; Quadros et al., 2022; Teoh et al., 2024; Weder et al., 2025b). Large deviations are also found for BC: the ELK inventory considers flight phase-dependent information on engine thrust along the trajectory, which is included in the applied soot particle mass calculation in the form of relative thrust. The TraK inventory approximates the relative engine thrust using the ratio of the flight-phase-specific fuel flow to the maximum fuel flow at take-off, which is subject to uncertainties.

Unlike the other transport sectors discussed above, the uncertainty analysis for aviation is performed on a grid cell basis, thus providing geographically resolved uncertainty scores on the same grid as the emissions. The vertically aggregated uncertainty scores (Fig. 16) reveal a generally low uncertainty, with a score of 2 along most routes. The lowest uncertainty (score 1) is calculated over the North Atlantic, due to the high availability of real flight path data and the enhanced quality of performance models of operating aircraft for this region. In contrast, larger uncertainties are assigned over China, India, Mexico and Africa, where the coverage of flight path data and taxiing times statistics is limited. For particle emissions (BC and nvPM, Fig. S85), the uncertainties are generally higher compared to the other species, because engine-specific emission factors are not available for some aircraft and emission factors from fleet averages have to be applied instead.

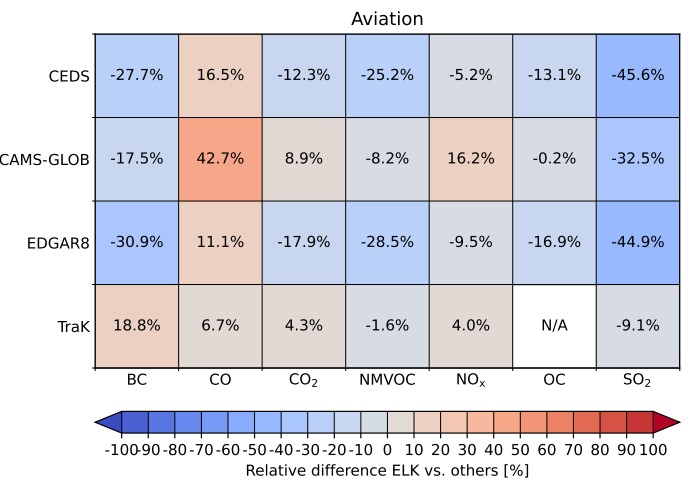

**Figure 15.** As in Fig. 7, but for aviation emissions.

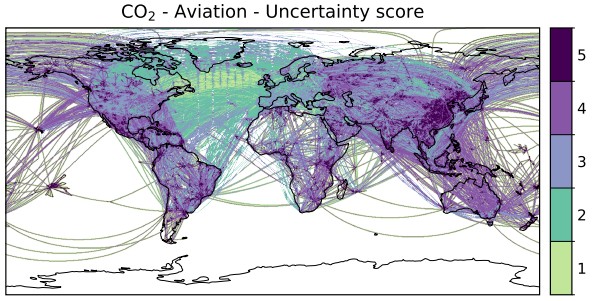

**Figure 16.** Uncertainty scores for the vertically integrated $CO_2$ aviation emissions, ranging from 1 (very low uncertainties) to 5 (very high uncertainties).

## 4.4 Energy for transport

The global $CO_2$ emissions from transport-related refining activities are shown in Fig. 17. Although the data is provided on the same grid as the land transport and shipping emissions, the figure shows the individual emission point sources to enhance their visibility on the map. The emissions are large in industrialized and highly populated regions, like U.S., Europe, Russia, Middle East, East China and Japan, while they are low in less developed and populated regions, e. g., Africa, South America and rural regions of Australia, Canada and Russia. Generally, the emission sources are concentrated along coastal areas (e. g., the North Sea coast of Belgium and the Netherlands) and big rivers (e. g., the Mississippi in the U.S.), as refineries are often built at strategic locations to facilitate the transportation of refinery products. The total $CO_2$ emissions from the energy-for-transport sector in 2019 amount to 533 Tg (Table 5), which adds to the direct $CO_2$ emissions of the three transport sectors (8695 Tg).



Hence, the energy-for-transport sector contributes an additional 6.4% of indirect $CO_2$ emissions, underscoring the importance of considering the emissions from this sector for a comprehensive assessment of the transport impacts on climate.

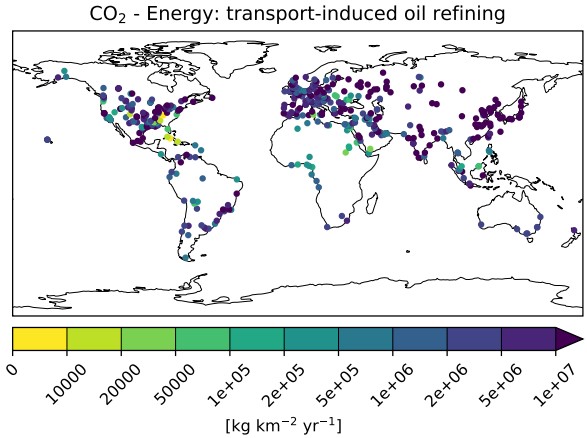

**Figure 17.** $CO_2$ emissions from the energy-for-transport sector.

Since we focus solely on the fraction of energy emissions related to transport, a comparison with other inventories is not appropriate, as they report total emissions from the energy sector rather than just the transport-related share. To validate the energy for transport emissions, other sources are considered instead, although a fully consistent comparison with the ELK

data is not feasible. One dataset that can be used for a plausibility check is the one by Ma et al. (2022), who calculated the overall $CO_2$ emissions from refineries in different countries. This is compared to the ELK energy emissions in Fig. 18a and shows generally higher values. This can partly be explained with the fact that Ma et al. accounted for the emissions from all refinery activities and not only the transport-related ones. In the ELK inventory, the transport-related emissions are on average responsible for about 50% of the total emissions. Ma et al. also included further sources of emissions, like processing,

flaring, and fugitive emissions, which are not considered in ELK and, according to Sun et al. (2019), are expected to contribute about 13% of total $CO_2$ emissions. Furthermore, Ma et al. comprises all GHG emissions, while the ELK values are for $CO_2$ only. Given these differences, it is not surprising that the ELK emissions in the energy-for-transport sector are lower, still this comparison is a useful quality check on the ELK energy emissions, ensuring that transport-related emissions from oil refining are below the overall emissions from the refinery sector. Fig. 18a also shows data for some countries from the UNFCCC

(https://unfccc.int/ghg-inventories-annex-i-parties/2023, last access: 10 June 2025), reporting total fuel combustion emissions from petroleum refining (IPCC sector 1.A.1.b). Also in this case, the ELK emissions, which consider only the transport-related fraction of the total, are below the UNFCCC figures. Note that the UNFCCC reported data for Russia and Japan are omitted in the analysis, since they are used as an input for the ELK inventory.

We further compare the energy demand estimated in the ELK land transport and aviation sectors against the IEA extended

energy balances (https://www.iea.org/data-and-statistics/data-product/world-energy-balances, last access: 10 June 2025) in Fig. 18b and c, respectively. The IEA fuel consumption is fed into the estimation of the share of fuels from refineries used in



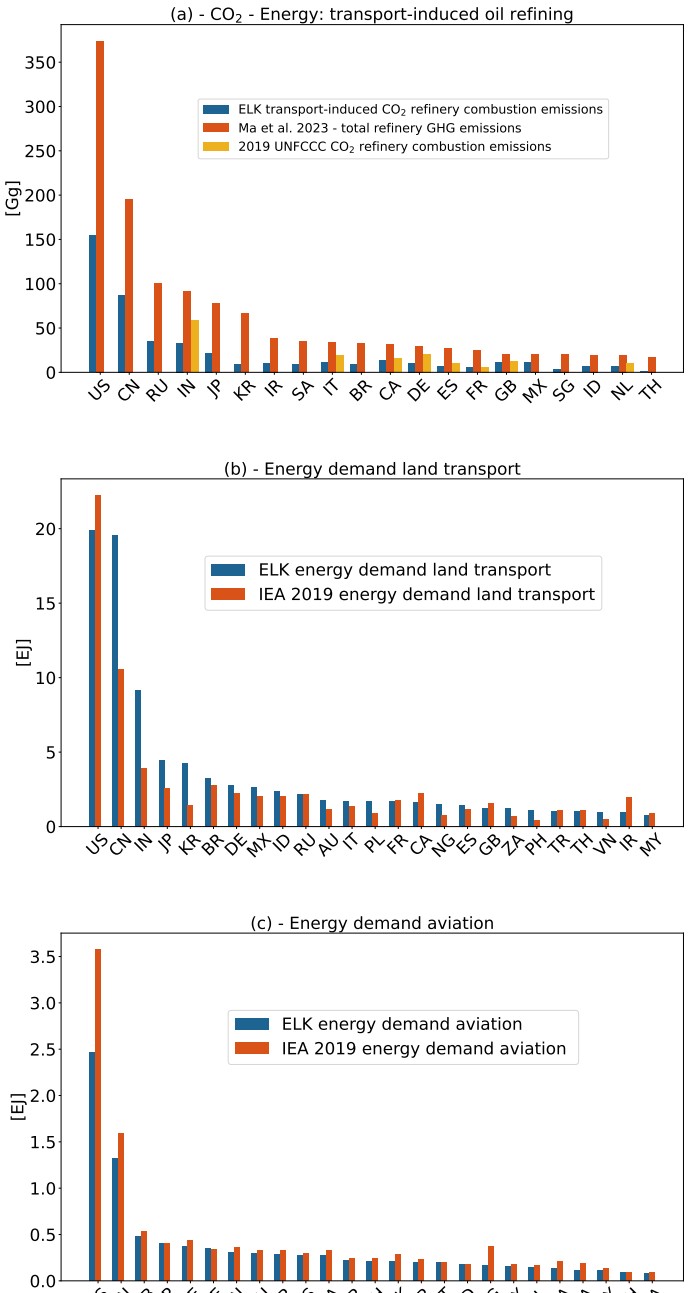

**Figure 18.** Comparison of (a) $CO_2$ emissions from energy for transport with GHG emissions in Ma et al. (2022) and reported data from the UNFCCC; energy demand of the land transport (b) and aviation (c) sectors in ELK compared with data from the IEA energy balances.



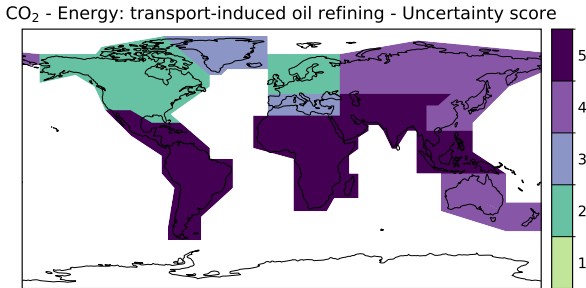

**Figure 19.** Uncertainty scores for the $CO_2$ energy for transport emissions, ranging from 1 (very low uncertainties) to 5 (very high uncertainties).

the transport sector against total refinery products. For aviation this is realized by assigning the consumption to the departure country. For the shipping sector an attribution of fuel consumption to specific countries is not possible. The ELK inventory shows a larger (smaller) fuel consumption for land transport in Asia (North America) than the IEA data. For aviation, the ELK fuel consumption is lower than the IEA data in China and North America.

The uncertainty analysis of the $CO_2$ emissions from the energy-for-transport sector results in a low uncertainty score 2 only for U.S., Canada, Central and Northern Europe where point source emission data is available for refinery activities. Larger uncertainties are assessed in the other regions, e. g., Russia and Oceania due to uncertainties regarding refinery locations, and are particularly high (score 5) in Central and South America, Africa, South and East Asia where a default approach is applied to estimate emissions.

## 5 Conclusions

This paper documents the ELK global emission inventory for the transport sectors, incorporating land transport, shipping, aviation, and energy-for-transport emissions from oil refineries. The methodology for emission calculation described here offers several advantages over existing global inventories, including a consistent bottom-up methodology applying, whenever possible, common underlying data across the transport sectors, the consideration of indirect transport emissions in the energy sector, and a higher sectoral resolution to resolve the emissions from specific vehicles or aircraft types, which opens interesting perspectives for assessing the effects of different transport categories, like passenger vs. freight. Furthermore, the ELK inventory provides more comprehensive information than commonly found in global inventories. This includes, for instance, aviation-specific quantities, that are particularly valuable for the aviation climate modelling community, and non-exhaust emissions of particulate matter by land transport, whose impact will gain importance with the transition to electric vehicles. The ELK inventory is complemented with a geographically resolved uncertainty score resulting from a thorough quantitative uncertainty assessment, to facilitate informed usage of the emission data.





In the ELK inventory land transport emissions are dominated by cars and heavy freight trucks, which are responsible of 45% and 33%, respectively, of the sector's $CO_2$ emissions globally. Other categories are relevant at the regional level, for instance 2-wheelers emissions in India and buses in North America, but the global contribution of each of them is below 10%. The emissions of the short-lived species reveal the importance of specific technologies for certain categories, like the widespread usage of diesel engines for trucks and of 2-stroke engines by 2-wheelers. The emissions from the shipping sector in the ELK inventory are dominated by the international shipping, which contributes more than 85% of the global emissions of all species, with the exception of CO, which to 65% is attributed to domestic navigation. This is due to the old age of the inland shipping fleet, characterized by engines with no emission standards and small engines operating at higher fuel-to-air mixtures leading to incomplete combustion and higher CO emissions. The ELK inventory also accounts for the IMO regulation on fuel sulphur content, resulting in lower $SO_2$ emissions in the SECA regions. The ELK aviation inventory shows that wide-body and single-aisle aircraft are responsible for 94% of the global $CO_2$ aviation emissions, although wide-body flights are only 8.5% of the total. Regional differences are also remarkable across the different aircraft categories in ELK, with wide-body and cargo flights dominating the long-distance routes, while single-aisle and regional aicraft fly shorter routes. Regional aircraft also tend to fly lower, with a peak around 6 km, in contrast to the typical flight altitude at 10-12 km of the other types. These considerations apply to all aviation emitted species, although exception exists, for instance for CO and NMVOC, which are caused to a relevant extent by taxiing activities, on which short-range flights have a larger impact. The energy-for-transport sector emissions, a key feature of the ELK inventory, contribute an additional 6.4% to the global transport $CO_2$ emissions, with the emissions concentrated in the developed countries, mostly around coastlines and big rivers, where refineries are located to facilitate the trade of refining products.

The ELK inventory was validated against established global emission inventories, yielding generally consistent results with minor discrepancies for $CO_2$ and larger biases for non-$CO_2$ species. The latter could largely be attributed to differences in assumptions and methodologies across these inventories, or to intrinsically large uncertainties for certain species, which are common across different inventories. Although the emission data presented here target the present-day (year 2019), the ELK emission models are designed to easily accommodate future projections under various scenarios and storyline assumptions.

Despite the added value of the ELK inventory demonstrated in this work, several improvements are possible and should be considered in future versions. Various components of the land transport emission calculation methodology and data sources could be enhanced. The calculation of the emission factors of $CO_2$ and $SO_2$ could be improved by considering country- and segment-specific fuel consumption data. As more databases become available, emission factors for other pollutants should be refined by incorporating additional country-based emission factors. Improved datasets would be particularly useful for 2-wheelers and for heavy and light commercial vehicles, especially in developing countries. The current spatial disaggregation model is limited to countries with available open data on nationwide traffic counts and applies only to passenger cars. Expanding this model to include heavy and light commercial vehicles would be a much required improvement. Furthermore, as more traffic count data become available in other regions, the model could be extended and validated to deliver improved spatial disaggregation of emissions.





In the shipping sector, using the best available AIS data source is essential for ensuring accurate emission calculations. Increasing the number of vessel movements in the input data will enhance the completeness of the dataset. Future expansion of the AIS data exchange system capabilities would allow to increase the number of satellite transmission channels, thereby improving coverage in remote areas. New research findings on alternative fuels and propulsion systems are needed to improve the accuracy of emission factors. Future changes to IMO regulations and improved emission inventories might require shipping companies to register emissions data onboard vessels and transmit it via binary messages, eliminating the need for manual estimates. No additional reception equipment updates would be required for this purpose, as the AIS binary transmission protocol can be received in the same way as it is currently the case for position data. The domestic navigation emissions in the ELK inventory are limited to Europe, U.S. and China. To extend the coverage to additional regions such as South America, India, and Southeast Asia, improved transport statistics would be crucial. However, these data is either not public available or difficult to access, and support from local authorities may be needed to obtain further information. Additionally, improvements to the domestic navigation dataset could benefit from further development of the SAR detection concept discussed in this work, along with an improved interpolation method for inland water vessels. This could complement the AIS data, especially in regions with limited coverage.

For the aviation sector, follow-up research should focus on novel aircraft concepts and engine technologies, such as the application of sustainable aviation fuels (SAF), electric and hydrogen-powered engines. This could be integrated into the aviation emission model, thus enabling the assessment of the environmental effects of future aviation scenarios. The underlying aviation emission model GRIDLAB should be extended to reliably simulate flights with short distances below 150 nautical miles. Depending on the available computational resources, the resolution of the aviation inventory could be further increased, enhancing its applicability in high-resolution atmospheric models.

With the expected increasing shift to electric vehicles, the assessment of transport-related emissions in the energy sector, specifically in the electricity generation, needs to be included in the energy emission inventory for the upcoming years. In addition to the emission from refineries, other upstream processes, like oil extraction, significantly contribute to the emissions and should be considered in future versions. Other sources, like fugitive and process-related emissions, could also be added with reasonable effort, although these are more difficult to determine. Some species, like $CH_4$, $NH_3$ and OC, were not included in the ELK energy inventory due to inconsistent data availability across different regions. Future investigations would be required to assess data availability and expand the inventory by additional species.

The method developed here for assessing uncertainty could be further refined by incorporating additional sources of uncertainties, addressing them at the subsector level, and by improving the robustness of the scores by applying different weights to the activity data and for the emission factors. As the uncertainty of the activity level and the emission factors might vary substantially within the relatively coarse spatial classification scheme adopted here, increasing the spatial resolution of the uncertainty scores would significantly increase their applicability.

The ELK inventory is released on a Creative Commons license and the scientific community is welcome to use this newly developed datasets for assessment studies and decision-making analyses, and to provide feedback for improvements on future versions of the dataset.





# 6 Data availability

The ELK emission inventory is available at https://doi.org/10.15489/d9dswthdix21 (Ehrenberger et al., 2025, for land transport), https://doi.org/10.15489/lhqawfes5755 (Banyś et al., 2025, for shipping), https://doi.org/10.15489/86s8uwpxik95 (Weder et al., 2025a, for aviation), and https://doi.org/10.15489/gixadaq6ds98 (Draheim et al., 2025, for energy-for-transport). Users are kindly requested to cite the present paper if using the data for scientific publications.

*Author contributions.* M. Righi led the ELK project, contributed to defining the requirements and structure of the emission inventories, performed the analysis and validation of the results, contributed to the concept for the uncertainty assessment and coordinated the writing of the manuscript with input by all co-authors. S. Ehrenberger created the emission inventory for global land transport, contributed to the uncertainty assessment and coordinated the generation of the emission datasets for all sectors. S. Brinkop contributed to the requirements definition, to the analysis and the validation of the emission inventories. J. Hendricks defined the requirements and structure of the emission inventories, and contributed to their analysis and validation. J. Hellekes developed the method for the uncertainty assessment and coordinated its implementation for all sectors. P. Banyś programmed the IMO algorithms for maritime emission calculation and carried out the emission calculation for international shipping. I. Dasgupta created the emission inventory for global land transport and contributed to the uncertainty assessment. P. Draheim created the emission inventory for the energy sector and contributed to the uncertainty assessment and the validation. A. Fitz created the emission inventory for domestic navigation and contributed to the uncertainty assessment. M. Löber contributed to the uncertainty assessment of non-exhaust emissions. T. Pregger co-developed the concept for the energy sector emission inventory. Y. Scholz co-developed the concept for the energy sector emission inventory, especially with respect to spatio-temporal data and data processing. A. Schulz co-led the ELK project and contributed to the emission inventory for European land transport. B. Suhr contributed to the emission inventory for domestic navigation. N. Thomsen created the emission inventory for European land transport, prepared the socioeconomic base data, and contributed to the uncertainty assessment. C. M. Weder extended the methodology of air traffic emission modelling, categorized the worldwide aircraft fleet into subsectors, quantified the aviation emissions, derived the methodology for the aviation uncertainty assessment, created the aviation emission inventory data and contributed to the validation. P. Berster modelled the air traffic demand and fleet composition for passenger and cargo transport and provided worldwide flight plans to the aviation emission modelling. M. Clococeanu validated and updated the database of emissions factors and improved the model capabilities and performance of the aviation emission model. M. Gelhausen modelled the air traffic demand and fleet composition for passenger and cargo transport and provided worldwide flight plans to the aviation emission modelling. A. Lau provided air traffic data and performed their post-processing and interpretation for the application in the aviation emission inventory. F. Linke originally developed the applied methodology of air traffic emission modelling and conceptualized the methodological improvements of the aviation emission methodology. S. Matthes provided data on regionally varying sulphur content in aviation fuel for the aviation emission modelling. Z. L. Zengerling performed comprehensive and detailed trajectory simulations for the aviation emission modelling.

*Competing interests.* The authors declare no competing interests.





*Acknowledgements.* This work was supported by the DLR impulse project ELK (*EmissionsLandKarte*) under the aeronautics, space, transport, and energy research programs of the DLR. We are grateful to Robert Sausen and Stefan Seum for their contribution in the early phases
of the project planning, to Thomas Haase for his constant support, to Alina Fiehn (DLR Institute of Atmospheric Physics) for her insightful comments on an earlier version of this manuscript, and to Volker Matthias (Hereon) for useful suggestions. We highly appreciate the support of Holger Pabst and Fabian Baier (DLR Institute of Air Transport), for the contribution to the underlying air traffic demand and fleet modelling; Felix Fritzsche, Simon Müller, Sebastian Wöhler and Sören Kolb-Geßmann (DLR Institute of System Architectures in Aeronautics) for the development and provision of additional aircraft performance models; André Twele, Lucas Angermann, Mathias Böck, Monika
Friedemann, Julian Meyer-Arnek, Jonas Müller and Martin Mühlbauer (German Remote Sensing Data Center) with the data processing and storage. We also kindly thank all data suppliers for providing the input data for aviation emission modelling (specifically ECMWF, EUROCONTROL, OpenSky Network). The ELK aviation emission inventory has been created with elements of Base of Aircraft Data (BADA) Family 4 Release 4.2 which has been made available by EUROCONTROL to DLR (German Aerospace Center). EUROCONTROL has all relevant rights to BADA. ©2021 The European Organisation for the Safety of Air Navigation (EUROCONTROL). All rights reserved.
EUROCONTROL shall not be liable for any direct, indirect, incidental or consequential damages arising out of or in connection with this product or document, including with respect to the use of BADA.

This paper is dedicated to the memory of Robert Sausen.



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
