# Peer review of "The ELK global emission inventory for the transport sectors"

_Earth System Science Data, 2025_

## Author Comment (AC1)

We grateful to the referees for their constructive comments, who helped us to revise and improve the manuscript. A detailed point-by-point reply can be found below: referees' comments in *italic blue*, replies in plain text, text passages quoted from the paper in red. The line numbers mentioned in our replies refer to the revised version of the manuscript.

**Referee #1**

*This manuscript presents a global emission inventory for the transport sectors of 2019, covering land transport, shipping, aviation, and transport-related emissions from the energy sector. Although this emission inventory is developed with detailed data and models, the spatiotemporal resolution is the same with some existing data products. Many studies on the source-specific transport emissions provided finer resolution data.*

We are not claiming that the added value of the ELK inventory lies in its spatial resolution. The strengths of our inventory are others, as clearly listed in the Introduction. As the main target application for our product are global modelling studies, the provided horizontal resolution of 0.1°×0.1° is more than enough to meet the needs of current global models, which have a typical resolution of 1° to 2°. Even high-resolution models, such as the one participating in the HighResMIP v1.0 project (Haarsma et al., 2016 https://doi.org/10.5194/gmd-9-4185-2016) require a 50 km resolution in the atmosphere (i.e. about 0.5°) and 0.25° in the ocean. The next generation of HighResMIP v2.0 will further increase this to 25 km, which is still within the scope of our data. Furthermore, as stated in the introduction (line 82): "if necessary, the ELK emission models are capable of modelling individual sectors at a higher resolution.".

*The manuscript is too long to read.*

The submitted version has 68 pages in discussion format, which should correspond to about 35 pages in final ESSD format. This length is probably above average, but there are examples of similarly long papers recently published on ESSD: Filippucci et al., 2025 (https://doi.org/10.5194/essd-17-5221-2025), Sullivan et al., 2025 (https://doi.org/10.5194/essd-17-5007-2025), Kanaya et al., 2025 (https://doi.org/10.5194/essd-17-4901-2025). We have chosen ESSD since it has no page- nor word-limit and we think it is important to document the emission models and the methods in detail.

Also note that we have structured the paper in a modular way, such that interested readers can focus only on specific parts: e.g., the aviation community may want to read just through Sects. 2.3, 3.3 and 4.3, while emission modelers interested in the methodical aspects may find all the details just by looking at Sect. 2. To facilitate the orientation of the reader in this modular structure, we have adjusted the section titles to be more explicit, e.g. "2.1.2 Activity data for land transport", "2.3.3 Emission factors for aviation", etc.

*The subsectors of the transport emissions are confusing. Does heavy-freight truck equal to heavy commercial vehicle? Does bus mean heavy-passenger truck? Please provide a definition of each subsector.*

As mentioned in Sect. 2.1.1 (line 106): "the vehicle categories considered in the ELK inventory are passenger cars, 2-wheelers, buses, light commercial vehicles (LCVs), heavy freight trucks (HFTs), passenger and freight rail.".

The use of HCVs or heavy commercial vehicles interchangeably with HFTs was indeed confusing and is now corrected in text and figures.

*For freight transport, data are only available for OECD countries. How did the author assign other countries to reference countries? Since OECD countries are mostly developed countries or large countries, how to deal with the system differences between OECD countries and other developing countries?*

The reference countries were assigned based on the 2010 to 2018 development of GDPpC, where OECD countries with the most similar values were assigned to non-OECD countries. After this assignment, the reference country's regression model parameters and an estimated elasticity were applied and travel volumes predicted. There is a certain level of uncertainty regarding non-OECD countries, as you have observed, which we addressed with our uncertainty assessment in Sect. 3.1.1 (line 828):

"For about 50 countries the activity data for freight transport is known and easily accessible. For the remaining countries, a modelling approach based on GDPpC is chosen. However, for some countries structural differences in the relationship between GDPpC and the freight transport performance are observed. Overall, this leads to both over- and underestimations with a high impact on the resulting inventories since the activity data is the baseline for bottom-up calculations."

*Regarding Fig. S1 and Table S1, if I understand correctly, only the spatial disaggregation employed the Europe model. Why the total emissions are different with the two disaggregation methods?*

Your understanding is correct. The difference is a result of using road-type differentiated emission factor values in Europe in comparison to a single emission factor for all roads per vehicle subsegment. The following line has been added to the supplement to clarify this: "Even though the methods only differentiate for the spatial disaggregation, road differentiated emission factors are used for Europe (as is available in HBEFA), and hence the total emission values are different between the global and European inventories."

*Only the U.S., EU and China were covered in the calculation of domestic navigation. This coverage gap makes the analyses on share of domestic navigation (three countries) in total navigation emissions unreasonable.*

The data availability in the inland waterways is much scarcer than for maritime transport, allowing only for a lower spatial coverage. Yet in terms of transport volume, we are confident to cover more than 90% of the goods transported on rivers worldwide, covering the largest and most relevant transport systems with a high spatial resolution for emissions. We updated our statistics in the paper on the remaining waterway systems to rely on more recent sources and reflected this in the discussion on the inland waterway activity levels in Section 3.2.3 (line 1015):

"Further waterway systems of the Amazon, Parana, Volga, India National Waterways, Nil, Congo, Mekong, and Niger are neglected in this dataset due to limited assessment and accessibility of transportation statistics. The overall proportion in transported goods on these waterways is below 10% comparing various data (Eurostat, 2022; Lewis et al., 2022; Press Information Bureau, Government of India, 2023; Jamali Jaghdani and Ketabchy, 2023), therefore resulting in a tendency to mildly underestimate the global activity level."

Please note, that even if the total volumes are known, estimating the trade routes and distance transported on complex river systems, e.g. the Amazon, is difficult. The activity level could only be estimated with very high uncertainty and we decided to not include it in the dataset.

*Some vertical emission gaps are shown on these maps, breaking the aviation rote. Please double check the data.*

We assume that the vertical separation in cruise flight levels follows worldwide the semicircular rule. If some flights operate only in one direction, just every 2nd cruise flight level show cruise emissions (see for example Fig. S56e at southern latitudes) and in between emissions occur only due to climb and descent operations. Above an altitude of 41000 ft, the minimum vertical separation is 2000 ft instead of 1000 ft. So, in FL420 and FL440 emissions only occur during e.g. step climbs or initial descend phase (e.g. Fig. S56c at 0°-30° latitudes)

*When comparing with other inventories, the author only listed the differences but did not provide explanations of these differences. With more than 100% differences, why did the author believe that the emission estimations of this study are more reliable than other inventories?*

We are not claiming that the ELK emissions estimates are more reliable than other inventories. Evaluating sector-resolved emission inventories at the global scale is not feasible, due to the lack of suitable experimental data. The goal of the comparison was to identify differences and explain them, when possible, in terms of the methodical differences between the inventories. More than 100% emissions are indeed found in some cases, but these often happens with species for which the spread across the inventories is also very large (BC from shipping is a good example of one of such cases). In most of the other cases the differences are smaller, while other inventories stand out as outliers (e.g. CEDS for $NH_3$ from land transport, EDGAR for $SO_2$ from land transport, GAIA for CO from aviation). This is not meant to be an evaluation, but a transparent documentation of the differences to support the data users when interpreting the results of their models.

*The emissions estimated in this study were compared with other Global all-source emission datasets and the author mentioned large variations among these inventories, thus some source-specific emission dataset may provide more reliable estimations, e.g. "The high-resolution global shipping emission inventory by the Shipping Emission Inventory Model (SEIM)" and "The high-resolution Global Aviation emissions Inventory based on ADS-B (GAIA) for 2019–2021". Please further compare the emission estimations in this study with them.*

Thank you for pointing us to these additional datasets, these have been included (Fig. 10, Fig. 15 and corresponding figures in the supplement). We have also added the recent GAIA aviation inventory by Teoh et al. (2024). The text has been updated accordingly.

*The SO2 emissions from navigation considered the IMO regulation on fuel sulphur content, leading to lower total emission comparing with EDGAR. First, a mapping showing the SO2 emissions distribution is helpful for readers to understand the effect of IMO regulations; Second, the total SO2 emissions are comparable with CEDS and CAMS, do these two inventories considered the IMO regulations?*

A map of $SO_2$ emissions is provided for the ELK inventory (Fig. S43). To do so for the other inventories is not the task of this paper. CEDS and CAMS consider indeed the IMO regulations of fuel sulphur content, while EDGAR8 does not (as noted in the text).

**Referee #2**

*1. (line 68): The label "present day" is used for 2019, which by now 6 years ago. Is there any indication of how much things might have changed since then? E.g. what has been the recent % change per year, very broadly speaking?*

As the ELK project started (2022), 2019 seemed the most obvious choice for a present-day case (2020 and 2021 were not suitable to the peculiar conditions following COVID-19). Recent trends in transport emissions can be inferred from recent updates of other well-established inventories, e.g. EDGAR8 (Crippa et al., 2024) or CAMS-GLOB (Soulie et al., 2024). Note, however, that the methods described in our manuscript are not bound to a specific year, but can be easily applied to generate emission inventories for any time horizon, including scenarios (assuming that boundary data are available).

*2. (section 2.1): the methods used for non-road (rail) transport are not really discussed here, both activity data and emission factors.*

Thank you for pointing this out. The following text has been added to Sect. 2.1.3 (line 195):

"For other transport modes, such as passenger and freight rail, two-wheelers, and buses, the technology shares (including train-type distributions) and corresponding energy efficiencies for major countries are taken from earlier work (Teske et al., 2019). The countries covered include Australia, Brazil, China, Germany, India, Japan, Russia, South Africa, and the United States. Emission factors for each technology in the reference year were derived from the most recent country-specific literature."

A full description of the literature used per remaining vehicle categories and major countries is beyond the scope of this paper given the multiple sources. Specifically, Fruhwirt et al. (2023, https://doi.org/10.1016/j.trd.2023.103858) was used as a reference for non-exhaust emissions for rail transport.

Regarding activity data for passenger transport, reference modal splits of the transport volume were applied for each country. This is mentioned in the Methods chapter (2.1.2, line 137): "The transport volumes for other modes are then calculated based on these car transport volumes by applying representative modal splits."

The modal splits were determined for countries with sufficient transport statistics, and then inferred for other world regions. This leads to an uncertainty score which is discussed in the Uncertainty section (Sect. 3.1.1, line 816): "After the car traffic is calculated, the volumes for other means of transport are derived based on the average share per world region. Since car transport is the primary contributor to land-based emissions, the effect of over- and underestimations for the remaining subsectors on the resulting inventories is rather low."

Regarding freight transport, the procedure for rail transport is similar, other means of transport are not relevant in this sector. The following is noted in the uncertainty assessment (Sect. 3.1.1, line 837): "The transport performance via railway is known only for a small number of countries. The volume is inferred for remaining countries with a railway network but without detailed volumes published. As a tendency, this causes overestimation with low impact given the small share of the mode."

*3. (section 2.1.3): it is not (explicitly) addressed how are electric vehicles taken into account. This can have a larger impact in certain regions and for certain subsectors (such as rail).*

For passenger cars and heavy-duty vehicles, the EV shares are part of the vehicle stock data along with other powertrains. With reference to the reply to the second point above, for buses, 2-wheelers and rail, previous studies conducted by DLR were the base of electrification estimates (Teske et al., 2019).

*4. (L200-203): are the stock numbers from 2018 instead of 2019? Where does the 2021 come from?*

Since vehicle stock data is expensive to purchase, the most recent data available at the beginning of the project (2022) was acquired for future use. Given that the stock data are available by year of registration (back to 1970 for some countries), survival curves for each vehicle category were used to build the stock for 2019. Given that the difference is only two years, major uncertainties are not expected from these assumptions.

*5. (L212-217): there are not enough details here for this method to be reproducible. Is there a separate publication that describes it? Especially if this is one of the improvements of ELK vs. the state of the art, it would be nice to mention things like what is the structure of the network, how was is trained, what is the source of the data, how well can it reproduce actual traffic data, etc.*

Yes, this is a valid point indeed. The spatial disaggregation methodology and prediction capabilities will be described in detail in a separate paper. For now, the following text has been added in Sect. 2.1.4 (line 216):

"An alternative methodology is implemented in the ELK inventory to develop spatial proxies for passenger cars to disaggregate emissions in a country. A graph neural network is trained and tested to predict traffic counts in countries where traffic count data is openly and widely available. The approach relies on passenger car traffic count data collected on major roads in the U.S., Germany and UK. For model training and validation, U.S. traffic count data from 2019 are used, comprising 5,648 data points, of which 2,400 located on interstate highways (Federal Highway Administration). For Germany, 1,170 cleaned traffic count data points from 2022 on major roads are utilized (Bundesanstalt für Straßenwesen, 2019). For the United Kingdom, traffic count data are obtained from a total of 13,900 data points (Department for Transport, 2014, 2019). From each of the three datasets, approximately 1,000 data points are selected to avoid introducing model bias."

*6. (L218-219): is this improvement shown in your results or from previous studies? The evidence for it should be referenced here.*

We agree with your comment. Since the improvement has not yet been proven in this paper, the word "improved" is changed to "alternative" at line 216 and the sentence at line 228 is rephrased to avoid any comparison without evidence.

*7. (L223-234): you describe this alternative method, show that it results in significantly different spatial distribution (Fig. S1), but then do not discuss what does that mean and this new method is simply discarded. "The spatial disaggregation for Europe is based on ... ULTImodel", except that is not was is used for the data being released or presented in the Results section, if I understood correctly.*

While the spatial distribution result is not significantly different between the approaches, it is visibly more defined with the more detailed transport model approach, mostly due to a better representation of main highways and unsettled areas. This additional method shows that it is possible to apply a more detailed transport distribution model with transport generation, trip distribution and assignment steps to demonstrate forecasting abilities, as mentioned in Section 2.1.1 (line 117):

"While this [the global] approach is suitable for the estimation of emission sources on a larger scale with scarce calibration data sources, forecasting and the calculation of scenarios requires dedicated transport models describing travel behaviour (as demonstrated by Matthias et al., 2020). This is why, as a proof of concept for the correct allocation of emission sources, an additional emission inventory is created for Europe, where transport activity is first distributed on the road network, before aggregating the resulting emissions on a grid."

We have added the following sentence to the spatial disaggregation section for clarification (line 234): "As mentioned above, a more detailed spatial distribution method was applied for Europe as proof of concept (see Supplement)."

Furthermore, we have added the following in the supplements describing Fig S1: "It is visible that the alternative method for Europe shows the emission sources in more detail, as the main road network as well as unsettled areas are better represented."

*8. (L279-280): what fraction of the fleet moving in 2019 (or fuel usage) does these 86,192 represent? I.e., what percentage is missing from the estimates due to unavailable identification.*

There are a total of 76,508,329,255 vessel positions stored in the 2019 AIS dataset. About 50.3% of them originate from the above-mentioned 86,192 vessels which could be fully identified for the purpose of emission calculation. The remaining 49.7% of vessel movements, according to the static and voyage related AIS data, come from small craft, be it pleasure craft, tenders or special-purpose vessels, which have a short range of operation and are not significant for the global trade and flow of cargo. In terms of number of vessels (MMSI) there are a total of 1,830,316 MMSI numbers but only about 4.7% (86,192) could be fully identified and included in the emission calculation.

This clarification has been added to Sect. 2.2.1 (line 299): "It is worth mentioning that a total number of MMSI spotted within the dynamic AIS position reports stored in the 2019 AIS dataset stands at 1,830,316. However, from as many as 790,653 (43.1%) vessels only one dynamic AIS position report was ever received and stored in the AIS dataset. This makes all those vessels, even if they existed in the vessel particulars dataset, unusable for the emission calculation because no movement can be reconstructed from their AIS data. Moreover, in terms of all vessel positions (76,508,329,255) stored in the AIS dataset of 2019 about 50.3% originate from the above-mentioned 86,192 vessels which could be fully identified for the purpose of emission calculation. The remaining 49.7% of vessel movements, according to their static and voyage related AIS data, come from small craft, be it pleasure boats, tenders or special-purpose vessels, which have a limited range of operation and are not significant for the global trade and flow of cargo."

*9. (L280): What is the source of this assumption?*

This assumption is based on a crosscheck of the vessel particulars dataset (Clarksons Research, database extracted on 16 May 2023) which provides fuel and engine characteristics of a given vessel.

*10. (L401-405): despite ELK being a global inventory, emissions from domestic navigation are only evaluated for select regions / river systems (understandably due to data limitations). Because of this, I think this caveat should be emphasized earlier in the text. The other mentioned systems are said to contribute less than 4% of tonnage, but that is from a webpage from 2012 which quotes statistics from even earlier. And crucially, is there any estimate of what is the share of transport from waterways besides those mentioned?*

We crosschecked the transportation volumes of other river systems with more recent data and found that there was in fact an increase in transport volumes of other systems, resulting in an

estimated total of 9.6% in transport volume of the excluded river systems. We corrected this in the text and updated to more recent sources in Sect. 3.2.3 (line 1000):

"Further waterway systems of the Amazon, Parana, Volga, India National Waterways, Nil, Congo, Mekong, and Niger are neglected in this dataset due to limited assessment and accessibility of transportation statistics. The overall proportion in transported goods on these waterways is below 10% comparing various data sources (Eurostat, 2022; Lewis et al., 2022; Press Information Bureau, Government of India, 2023; Jamali Jaghdani and Ketabchy, 2023), therefore resulting in a tendency to mildly underestimate the global activity level."

Yet the difficulty remains, e.g. for the Amazon and Parana river system, that these data represent only the total volume and not the distance transported in tkm. Estimating the distance transported on complex river systems is difficult, which is why the activity level could only be estimated with very high uncertainty and we decided to not include it in the data set. Besides the waterways mentioned we are not aware of any statistics or significant quantities for other waterway systems.

*11. (L434-435): I am not an expert on this field, but is this assumption that conditions of waterways in China are closer to Europe than the US trivial? Also, my impression is that for everything else ELK does not seem to go for "a conservative approach" and instead aims for a "best estimate", so this line seems a bit odd.*

One of the largest factors impacting the energy efficiency of the waterway transportation is the use of non-propelled barges and pushing tows. In the US waterway system extensive use is possible, because of fewer waterlocks and wider channels. Analysis of the Clarksons WFR register shows that in the US 87% of inland waterway vessels are push barges. In Europe only 16% and in China 9% of the vessels are barges, which likely is a result of higher manoeuvring requirements in narrower channels and more frequent waterlocks. Due to the similar proportion of self-propelled vessels and size structure between the EU and China, we are convinced that applying a similar energy intensity is appropriate. We improved the wording of the section as follows:

"According to these values, the U.S. inland waterway system operates more than twice as efficient compared to the EU system. This may be due to wider and more navigable waterways and fewer locks. Furthermore, the more widespread use of larger, non-propelled barges in push tows enables more efficient transportation. The analysis of the Clarksons Research data showed that while in the US 87% of the vessels listed are non-propelled barges, in the EU it is only 16% and 9% in China. Therefore, the energy intensity for China is assumed to be similar to that of the EU."

*12. (L452-482): This paragraph reads a bit weird because that you are speculating about the feasibility of applying a method that is not used here. Some of this would perhaps be better suited for a Discussion section rather than the Methods.*

We agree that this would fit better to a Discussion Section, but our manuscript does not have such a section, hence we decided to discuss this concept within the methods. This is also because we would like to retain the current modular structure of the paper, in which each sector is presented and discussed independently of the others (see also the reply to the second comment by Referee #1). Moving this paragraph to a dedicated Discussion section would reduce the readability.

Another option would be to move this to the Conclusions, in the paragraph about shipping. The conclusions section, however, would then become quite long and the discussion on shipping would be imbalanced with respect to the other sectors in this section.

*13. (section 2.3): just a comment that does not need addressing, but since you are going through the trouble of estimating transport-related emissions in the energy sector, I would like to suggest a similar improvement could be the estimation of emissions from airport activities (ground support equipment, electricity and HVAC for buildings, fuel storage, land traffic supporting the airport, etc.). There had been previous efforts in this regard, such as in Stettler et al. 2011.*

Thanks for this valuable hint. For a complete assessment of energy demand for transport sectors as air transport it would be necessary to consider ground infrastructure and there would be interactions and synergies to other transport modes. Considering airport energy demand and feeder traffic was not the scope of ELK and we decided to focus on the aircraft emissions due to the flight operations – airborne and taxiing.

*14. (L524-526): what is the rationale behind the 5/7 split in the seasons? Are there any indicators that these two weeks with these weights might yield a good approximation of the annual average?*

Scheduled air transport, which covers the vast majority of air traffic, is divided into five winter months and seven summer months. The seasons themselves are really homogeneous, i.e., repetitive from week to week. Therefore, with one week from each season, the whole year can be upscaled with close to 100% accuracy without producing a lot of unnecessary identical data. On the other hand, taking just one average week is a lot more imprecise compared to the 5/7 approach.

*15. (L527-531): the definition of aircraft subsectors are maybe a little too arbitrary. For example, a Boeing 757 would be classified as a wide-body, even though it is a "single-aisle" in the literal sense. The A321XLR (which was not flying in 2019) would likewise fall under an unnatural classification.*

Yes, you are right: both aircraft types are not so easy to classify. In our inventory, B757 and A321 are classified as single aisle category. We slightly adjusted the number of seats definition so that the sample composition will not change. There might still be some B757-300 and A321 with a seat configuration of more than 235 seats. But for the large number of aircraft configurations, it should be clear where the distinction between short-medium and long-range aircraft lies. B787, B767 and A330 might have seat configurations below 235 but their maximum range would be higher than 4500 NM, so they will be assigned to the larger category. E170 and E175 typically have less than 100 seats but a range above 2000 NM, so they will be assigned as single aisle category.

*16. (section 2.3.3, section 3.3.2): it is not specified, but I assume your activity data gives aircraft typecodes, which you then use to select emissions data from ICAO. As pointed out in Quadros et al. 2022, there is a large amount of uncertainty associated in the mapping of aircraft typecodes to specific engine models, with the emissions of some species varying by an order of magnitude depending on the choices made. So I think it is important to describe how was this issue was handled here and what are its implications with regards to the uncertainty of the emission estimates.*

We allocated the engine emission data from EEDB and FOCA for the engine as specified in the BADA3, BADA4 and DLR dataset to expect best possible accuracy. For some aircraft as e.g. A319, A320, A321, A380, B787-8, B787-9, more than one BADA4 model dataset with different engines is available. In those cases, we use the aircraft-engine combination with the highest market share in year 2019 based on Cirium fleet database for all flights of that aircraft type. So, we only used one engine type per aircraft type but those who are recommended by BADA. We added this explanation in Sect. 3.3.1 (line 1053).

*17. (L574-580): BC is estimated from smoke number, but is also a component of nvPM which is estimated from separate measurements. I am curious if there are instances when the mass of BC is greater than nvPMm. Is that something you checked?*

Thanks for the question. Interestingly, BC mass based on smoke numbers is higher than nvPM mass for the majority of flights in the inventory, and for all of them, engine-specific nvPM LTO emission rates from EEDB are available. Note that there is no fuel flow correlation method available for extrapolating nvPM mass emission rate into cruise phase and the applied linear interpolation between 30% and 85% thrust setting may lead to an underestimation of mission total nvPM mass. For most flights where nvPM mass is higher than BC, a constant default nvPM emission factor of 0.08 g/kg from Stettler et al. (2013) has been applied, due to missing engine-specific data. This may lead to an overestimation. Engine-specific nvPM LTO emission rates are available for 83% of flights in the ELK inventory, so for the remaining 17% default emission rate has been applied and those flights are mainly responsible for the excess of 12% of nvPM mass compared to BC.

*18. (L578-580): are the (LTO) emission rates of nvPM adjusted for cruise conditions in any way?*

There was no explicit altitude correction made, in a way like BFFM2 for NMVOC and CO and DLR method for $NO_x$, but we applied the simulated engine thrust in flight altitude for linear interpolation.

*19. (L597-598): great circle trajectories are known to underestimate emissions (Teoh et al. 2024, for example), I am surprised that you do not make any adjustments to the emissions in those cases. Considering you already have a database of flights with actual trajectories, would it be feasible to derive a factor to adjust flight lengths when you do not have trajectories?*

We are aware on the underestimation due to the shorter distance of great circle routes and reported that by increasing the uncertainty score (i.e., larger uncertainty) in grid cells where more than 5% of encircled trajectory segments consisted of great circle routes.

*20. (section 2.3.5): I am curious if you estimated what the overall difference in fuel usage (or emissions) is of considering wind. Since head and tail winds cancel each other out over all flights, the difference should mainly be from crosswind drift. Quadros et al. 2022 found a ~0.7% increase in fuel burn, but maybe this approach with wind rose distributions yields a larger impact.*

We did not calculate an emission inventory based on the ground distances and neglecting wind effects that would have been useful for comparison reasons, so we cannot quantify the fuel burn based on ground distances but we calculated the annual total ground distance. The worldwide annual total wind corrected air distance exceeds the ground distance by around 1.5%. Linke (2016) came to similar results of an increased distance based on a sample of 12,655 flights due to wind effects.

*21. (section 2.3.6): if you have "actual 3D flight paths" (L592), why use this approach of the average between optimal and constant cruise flight level? Could the actual cruise altitudes be used in these estimates?*
This calculation scheme to adjust the cruise altitude of a reduced emission profile (RedEmP) along a real flight path altitude profile was not available in ELK. We will consider this for future inventories and project cruise emissions along the real altitude profile.

*22. (L637-639): just something to consider in the future, but maybe for air quality research a higher vertical resolution close to the surface would be beneficial, as you could get more precision to split how much emissions are inside/outside the boundary layer.*

Thank you for this valuable suggestion, we will consider it in future versions of the inventory.

*23. (L665-666): how much these 549 represent of global emissions? Is there any metric available that could put this into perspective, i.e. % of refineries or capacity worldwide?*

Thank you for your comment. Indeed, the mentioned sentence is confusing. We estimated emissions for 85 countries and distributed the country-level emissions to 549 refineries. As described in the supplement, we did not consider 20 countries which showed refinery activities according to the IEA Extended Energy Balances. These countries have a combined share of an estimated 0.25% of total emissions. We changed the text accordingly to be clearer.

*24. (L652-655, L673-677): some 13-34% of refining emissions are excluded from the analysis, and refining excludes some 13-44% of all emissions related to fuel production. This limitation is something that I think should be highlighted more strongly. For example, when you say "quantification of the transport-related emissions from the energy sector" in the Abstract, it gives a misleading idea about the scope of this quantification.*

Thank you for this comment. We changed the text in order to specify that we merely cover major transport-related emissions from the energy sector in our inventories.

*25. (L719): I wish it was not the case, but at least for aviation I am aware that emissions are expect to increase significantly in the future.*

You are right, we have modified the statement accordingly: "as emission levels are expected to change in the future".

*26. (L10-11, L65-67, L87, L731-732, etc.): I appreciate the uncertainty assessment as a valuable contribution of this work and a novel feature of ELK, but I am not sure you can call it a "quantitative assessment" or "quantitative uncertainty score". It is a qualitative assessment which you happen to label with numbers 1-5. To me, a quantitative uncertainty analysis would involve estimating uncertainty in quantifiable terms, i.e. ±X kg of emissions for a confidence interval of Y%. The "Interpretation" column of Table 4 is almost completely subjective; you are assigning the numbers 1-5 using "lowest uncertainty" / "low uncertainty" / "medium uncertainty" / "relatively high uncertainty" / "highest uncertainty" as your criteria.*

We fully agree with the reviewer that our two-step approach (qualitative evaluation of uncertainty factors followed by an aggregation into scores) is indeed qualitative in nature and does not constitute a formal sensitivity analysis in the statistical or probabilistic sense.
To prevent any confusion, we have removed the distinction between "qualitative assessment" and "quantitative assessment" from the paper and replaced it with the terms "qualitative evaluation" / "characterisation" and "rating", respectively (see Abstract as well as Sect. 1, 3, and 5).

*27. (L780-782): could a reference be provided for these indications? Border control might slow down traffic, but I would expect major border crossings to have relatively higher traffic than a similarly sized road elsewhere.*

Thank you for this comment. We wanted to highlight that there are differences regarding border crossings in different world regions. Unfortunately, there is few data on this, which is why we conclude that international traffic might be overestimated outside of Europe by the described modelling framework. We adjusted the text as follows to make it clear that we could not include the impact of customs and border controls (line 809):

"The possible effect of border controls and customs on the cross-country flow of passengers and goods could not be quantified with the described modelling framework due to a lack of data. This is less relevant within the Schengen area; for other regions, it is expected that the emissions along the major border crossings are slightly overestimated."

*28. (L801-803): what does "leaving emissions of LCVs unconsidered" mean? You still have LCV emissions in Europe. Is it just that those emissions are calculated only from GDP without activity data?*

This is in regard to the officially reported inventories, where no LCVs are included. In contrast to that, we include them in our inventory, but there is a level of uncertainty which we wanted to address with this sentence. We understand that this sentence is confusing and have changed it into the following:

"For European countries, only the transport performance of HFTs is reported in official statistics. The volumes for LCVs and their resulting emissions are thus estimated based on typical shares of the vehicle classes."

*29. (L807-808): how is volume inferred?*

The rail freight volume is inferred similarly to the road freight volume by collecting reported data and assigning reference countries. As the section in the uncertainty chapter describes, there is very few information on rail freight performance globally. Thus, we derived countries with an existing rail network and no data and assumed rail modal shares based on reported countries. We supplemented the method (applying modal shares from reference countries) so that the section now looks as follows:

"The transport performance via railway is known only for a small number of countries. The volume is inferred for remaining countries with a railway network but without detailed volumes published by applying modal shares observed in reference countries. As a tendency, this causes overestimation with low impact given the small share of the mode."

*30. (L837-839, L845-847): tire wear PN is said to be in the $10^{10}$-$10^{11}$ per vehicle-km range while a wider range of $10^9$-$10^{13}$ is given for brake PN. So it is not clear why the former was excluded due to high uncertainties but not the latter.*

As mentioned in the text, brake wear particles can be better measured due to encapsulation whereas measuring techniques for tire wear have not reached the same level of maturity and hence have lower reliability. The following sentence has been added for further clarification:

"Despite higher ranges of emissions from brake wear, the testing methodology for brake wear is more reliable than tire wear and hence only brake wear PN emissions are included."

*31. (L1009-1010): the scope of the inventory should be clarified earlier in the Introduction or Methods. Non-scheduled flights and these other categories are also transportation, but it is only here in the section about uncertainty that it is clarified that the inventory concerns exclusively scheduled civil aviation flights.*

Thanks for this helpful hint. We added a sentence to explain the considered and excluded flights in Sect. 2.3.1.

*32. (L1068-1070): 51%/35% of movements, but presumably a much smaller percentage of CO2 emissions (and other species)?*

Yes, that's right. Turboprop aircraft contribute to 38%/7% of total regional/cargo $CO_2$ emissions. We added the numbers in the sentences.

*33. (L1070-1072): several in-flight measurement campaigns have been done since 1997, maybe reference to a newer discussion could be cited here.*

We added DuBois and Paynter (2006) to the references.

*34. (Table S4): what does tirewear from rail mean?*

For simplicity and consistency PM tirewear is used instead of terming it as PM emissions from wheel-rail contact.

*35. (L1157-1159, Figure S26): compared to other inventories, CO2 estimates for W.NA are a third lower, C.NA estimates are half, and E.NA are about the same. Since all three are dominated by emissions in the US, I would expect the methods and data sources to be consistent across them. What could be driving this behavior?*

The differences perhaps arise from the relative distribution of urban nodes and freight arteries in these regions and the relative performance of the spatial disaggregation methods in these area types per vehicle type. Higher percentage difference is also noticed in C.NA due to lack of information to calibrate the disaggregation model in low volume roads and settlements. Apart from the disaggregation model of passenger cars in ELK which uses traffic counts as ground truth data, disaggregation for all other modes is based on proxies in both ELK and EDGAR. Hence there are high uncertainties in such a comparison.

*36. (L1166-1169): "factor-5 difference" should be 115 times higher? Is there a reason why EDGAR8 has such low amounts of SO2?*

Thanks for spotting this typo. The difference is indeed a factor 116, we have corrected this.

ELK uses updated sources for sulphur content in land-transport fuels (UNEP, 2020), which could explain the differences of our total for South-East Asia (0.22 Tg) with respect to CEDS (0.52 Tg) and CAMS (0.36), but it cannot explain the huge difference to EDGAR (0.002 Tg). We could not find any hint in the respective publication about the possible reasons for this low value.

*37. (Figure 6): I understand Greenland, but is there a particular reason why there are no emissions in Montenegro and Kosovo?*

Due to a lack of reliable data for Montenegro and Kosovo for political reasons (inclusion in official statistics), activity was not calculated and hence emissions for the countries are not available in the ELK inventories.

*38. (Figure 6): related to the lack of description on the methods for estimating rail emissions, why are there so much emissions in railways that are electrified? For example, ~100% of rail in Switzerland.*

That makes sense, however given that we use the technology split data for railways for 9 major countries for railways as now clarified, exceptions like Switzerland have not been incorporated in the inventory. Despite having open data on percentage of electrified lines in many individual countries, the data could not be used due to lack of unified data source on percentage of total activity operating on these lines. Electrification alone can however be used as a proxy in future work and would result in lower uncertainty.

The following has been added to line 1437: "A limitation of the current approach for rail transport is its inability to reflect country-level exceptions, particularly those with highly electrified rail networks relative to their regional representation. This can be improved with the availability of harmonised global datasets and activity reporting."

*39. (L1208-1211, Figure S47): these differences are very large. What could be driving them? The uncertainty score is rated "2 (robust)" for N.Atlantic-Ocean, but CEDS has 4 times the amount of $CO_2$ there.*

The calculation of emissions within the scope of this work must rely on vessel movements obtained from AIS data. As a result, one could draw a conclusion that the more movements can be registered, the more complete a calculation can be. A comparison of emission results with various sources is difficult because one would have to assume that everybody has used the same AIS datasets of vessel movements.

*40. (L1266-1279, Figure 14): the diurnal cycle analysis is shown prominently as a unique feature of this inventory, but I am missing what the use is of looking at it globally in UTC. Would it not make more sense to think of the "diurnal cycle" in terms of local time? Sunlight and temperature cycles, which are relevant for contrail formation, are going to be shifted according to local time.*

From the view of atmospheric processes, you are right. The diurnal cycle shown in Fig. 14 gives an overview on the diurnal distribution of worldwide emissions by subsector and shows emission distribution from a global perspective. If we considered emissions at local time, it would be necessary to show individually data split by regions and seasons to isolate local effects, because solar intensity and temperature do not only depend on daytime, but also on latitude, season and climate zone. The netCDF data enable such analysis, if one combines the global emissions of a UTC time step with the Sun position. We could consider such a detailed diurnal analysis more in detail in a follow-up emission inventory.

*41. (Figure 2): the x labels are in different orders between (a) and (b).*

Done. Thank you for spotting.

*42. (L558): typo in "for a plenty".*

Fixed. Thanks for spotting.

*43. (L574): typo in "are build on".*

Fixed. Thanks for spotting.

*44. (L625): weighted = averaged with equal weights?*

Corrected. Thanks for spotting.

*45. (L784): "GDPpc" / "GDPpC" capitalization inconsistency.*

Fixed. Thanks for spotting.

*46. (L892): maybe I am confused, but it stands to reason that stationary vessels might have emissions underestimated.*

With this statement we wanted to emphasise that a calculation of emissions based entirely on vessel movements obtained from AIS data may inevitably lead to an oversimplification of reality if one considers that no vessel movement means no emissions from the main engine, leaving only an

assessment of emissions originating from power generators and heating systems on board. The are situations when the main engine is in operation while a vessel remains stationary. Since there is no indicator of main engine operation within AIS data, there is no possibility of properly assessing the emissions of the main engine under such navigational conditions.

*47. (L1027-1028): are flight path data available for Asia?*

We did not have any flight path data for domestic flights in Asia. There was inadequate coverage of ADS-B data from OpenSky Network in 2019 over Asia. We only applied EUROCONTROL model3 flight paths for flights from Asia to Europe and North America (if passing European airspace) and vice versa. But we considered the routing along great circle routes due to missing flight path data with increased uncertainty codes in the relevant regions.

*48. (Supplement): "tire" vs. "tyre" in the main text.*

Fixed. Thanks for spotting.

*49. (Table S4): formatting of PM10 subscript.*

Fixed. Thanks for spotting (not an easy catch).

*50. (L1337-1338): you could also leave Russia and Japan in the figure and add the note to it.*

Thank you for this good hint. We edited the plot accordingly.

*51. (Data availability): just a suggestion, and there are tradeoffs, but a lot of storage could be saved by using compression on the netCDF files. Relatedly, there are some species for rail transport where you have 300 MB files that are entirely zeros, which could be omitted entirely.*

We agree that NetCDF-compression could have been used to save storage. This, however, would require the use of additional libraries, which would make the data handling slightly more complicated on the user's side. Since with our inventories we aim at supporting not only expert users from the scientific community, but also other target groups, we have decided to go for a simple (albeit less efficient) solution.

Regarding the zero-filled species, we chose to have a consistent data structure throughout the subsectors and to explicitly provide all species, even when they are just zero-filled. We think this is better than omitting them, which could be misinterpreted as a missing information on those species.